# Dual donor-acceptor covalent organic frameworks for hydrogen peroxide photosynthesis

Chencheng Qin[1,7], Xiaodong Wu[2,7], Lin Tang [1,7], Xiaohong Chen[3], Miao Li[1], Yi Mou[1], Bo Su[4], Sibo Wang [4], Chengyang Feng [5], Jiawei Liu[6], Xingzhong Yuan[1] ✉, Yanli Zhao [6] ✉ & Hou Wang [1,6] ✉

Constructing photocatalytically active and stable covalent organic frameworks containing both oxidative and reductive reaction centers remain a challenge. In this study, benzotrithiophene-based covalent organic frameworks with spatially separated redox centers are rationally designed for the photocatalytic production of hydrogen peroxide from water and oxygen without sacrificial agents. The triazine-containing framework demonstrates high selectivity for $H_2O_2$ photogeneration, with a yield rate of 2111 μM h$^{-1}$ (21.11 μmol h$^{-1}$ and 1407 μmol g$^{-1}$ h$^{-1}$) and a solar-to-chemical conversion efficiency of 0.296%. Codirectional charge transfer and large energetic differences between linkages and linkers are verified in the double donor-acceptor structures of periodic frameworks. The active sites are mainly concentrated on the electron-acceptor fragments near the imine bond, which regulate the electron distribution of adjacent carbon atoms to optimally reduce the Gibbs free energy of $O_2^*$ and $OOH^*$ intermediates during the formation of $H_2O_2$.

A promising path towards a sustainable future is utilizing solar energy to create chemical fuels, such as hydrogen peroxide ($H_2O_2$), which can be stored and transported[1–3]. Currently, $H_2O_2$ is manufactured by an energy-intensive and indirect anthraquinone technique[4], which involves safety issues caused by mixed $H_2/O_2$ and produce toxic byproducts[5,6]. Although artificial synthesis of $H_2O_2$ using semiconductor-based photocatalysis is an alternative method, the low activity of photocatalysts in pure water and heavy dependence on various sacrificial agents need to be urgently solved[7,8]. In terms of sacrificial agents, the introduction of alcohol leads to additional consumption of valuable chemicals, and the resulting aldehydes disfavor the separation of $H_2O_2$. Continuous charging with oxygen in photocatalytic systems under dark reaction adds an additional source of

energy to the consumption. Much effort has been devoted to developing sacrificial agent-free, energy-saving, and direct processes $H_2O_2$ production.

In principle, the $H_2O_2$ photosynthesis pathway consists of two complementary half-reactions, that is, $O_2/H_2O_2$ (two electrons) and $H_2O/H_2O_2$ (two holes)[3,9]. Most oxidation centers (i.e., hole species) can only undergo four-electron oxidation of water to oxygen (4e$^-$ WOR) due to thermodynamically favorable reactions rather than direct two-electron oxidation of water to $H_2O_2$ (2e$^-$ WOR). The former has an atomic utilization efficiency of only 68%, while the latter reaches 100%[7]. If both halves occur at the same/adjacent sites, the efficiency of the reaction is extremely limited by charge recombination. Therefore, direct evolution of $H_2O_2$ through a two-electron reaction involving

[1]College of Environmental Science and Engineering and Key Laboratory of Environmental Biology and Pollution Control (Ministry of Education), Hunan University, Changsha 410082, China. [2]College of Materials Science and Engineering, Nanjing Tech University, Nanjing 210009, China. [3]School of Frontier Crossover Studies, Hunan University of Technology and Business, Changsha 410205, China. [4]State Key Laboratory of Photocatalysis on Energy and Environment, College of Chemistry, Fuzhou University, Fujian 350002, China. [5]Catalysis Centre, King Abdullah University of Science and Technology, Thuwal 23955-6900, Saudi Arabia. [6]School of Chemistry, Chemical Engineering and Biotechnology, Nanyang Technological University, 21 Nanyang Link, 637371 Singapore, Singapore. [7]These authors contributed equally: Chencheng Qin, Xiaodong Wu, Lin Tang. ✉e-mail: yxz@hnu.edu.cn; zhaoyanli@ntu.edu.cn; wangh@hnu.edu.cn

separate oxygen reduction and water oxidation by respective photo-induced electrons and holes provides an ideal routine. Functional motifs with accessible channels and spatially separated redox species in the photocatalytic system that can facilitate the mass transfer and selective formation of $H_2O_2$ intermediates in the two-electron pathway are highly desirable.

Covalent organic frameworks (COFs), a type of crystalline and nonmetallic polymers, have become feasible and promising platforms in the field of artificial photosynthesis due to their porous structure, photoelectric properties, and photochemical stability[10–12]. The structural tunability and regularity of COFs, which is easily realized by practical building blocks with abundant topologies and dimensionalities, endow broad optical absorption, favorable mass transfer and fast charge carrier mobility[13–15]. However, COFs are rarely used as photo-catalysts for $H_2O_2$ production. Voort et al. reported two COFs based on a (diarylamino)benzene linker for $H_2O_2$ production, and subsequently vinyl, fluorinated and Ti-based COFs were used to catalyze $H_2O_2$ production[16–20]. Most of these reports require sacrificial agents or oxygen bubbling during the reactions. Thiophene is a stable π-aromatic five-membered ring compound, and due to the aromatic rings, electroactive and solubilizing groups can be introduced at each alpha or beta site, with excellent conductivity and adjustable electron density[21,22]. The combination of COFs and thiophene motifs is a popular trend in various fields, including ion batteries, hydrogen production, carbon dioxide reduction and organic synthesis[23–25]. It was reported that the pentacyclic thiophene-S functional motif in JUC-527 and JUC-528 acted as the active center for the electronic oxygen reduction reaction (ORR), and more thiophene-S (one thiophene-S and bi-thiophene-S) structures enabled higher ORR efficiency[26]. However, utilizing thiophene-based COFs to produce $H_2O_2$ through molecular engineering and mechanistic investigation remain challenge. Functional thiophene motifs can not only be preassembled to easily construct donor-acceptor COFs, but also differ in the transport direction and dissipation of excited charge carriers in the presence of polarized

imine linkage. It also provides a paradigm and indicate how the push-pull effects between intramolecular motifs and linkage chemistry in polymers affords high catalytic efficiency. Considering the n-π* transition of the long pairs on sulfur and suitable molecular orbital occupancy, precisely placing independent oxidation and reduction centers in an ordered manner based on thiophene-COFs may be more accessible for photocatalytic $H_2O_2$ production.

In this work, we presented benzotrithiophene (Btt)-based COFs with an intrinsically adjustable charge distribution for nonsacrificial photocatalytic $H_2O_2$ synthesis in water and natural dissolved oxygen under visible light irradiation. Sulfur-rich Btt has a planar conjugated system of $C_{3h}$ symmetry, in which three thiophene rings are blended on the benzene central ring, tending to achieve favorable optical absorption, π-electron delocalization and high hole mobility[27]. Combining benzo[1,2-b:3,4-b':5,6-b"]trithiophene-2,5,8-tricarbaldehyde (Btt) and different monomers with various electron-acceptor capabilities, including 4,4,4-triaminotriphenylamine (Tpa), 1,3,5-tris(4-aminophenyl)benzene (Tapb), and 2,4,6-tris(4-aminophenyl)-1,3,5-triazine (Tapt), successfully afforded three kinds of imine-linked COFs, termed as TpaBtt, TapbBtt and TaptBtt (Fig. 1a). We unveiled the concept of a uniport "atom spot-molecular area" via a double donor-acceptor method in a periodic framework to directly clarify the differences in terms of the electronic band structure, charge transfer directionality, donor–acceptor energy difference, and $O_2$ adsorption. As a result, the optimal photosynthetic rate of $H_2O_2$ in TaptBtt reached to 2111 μM h$^{-1}$ (1407 μmol g$^{-1}$ h$^{-1}$), far higher than that of TpaBtt (252 μmol g$^{-1}$ h$^{-1}$) and TapbBtt (557 μmol g$^{-1}$ h$^{-1}$), and also exceeded all of the previously reported COFs. The efficiency of solar-to-chemical conversion was 0.297 % for $H_2O_2$ production, surpassing that of the natural synthetic plants (global average ≈ 0.10%)[28]. Collected $H_2O_2$ can be directly used to degrade sulfamethoxazole (SMT, an emerging pollutant in natural water bodies and medical wastewater) via the Fenton reaction. Mechanistic study and theoretical calculation revealed that the active sites of the three COFs (2e$^-$ ORR) were

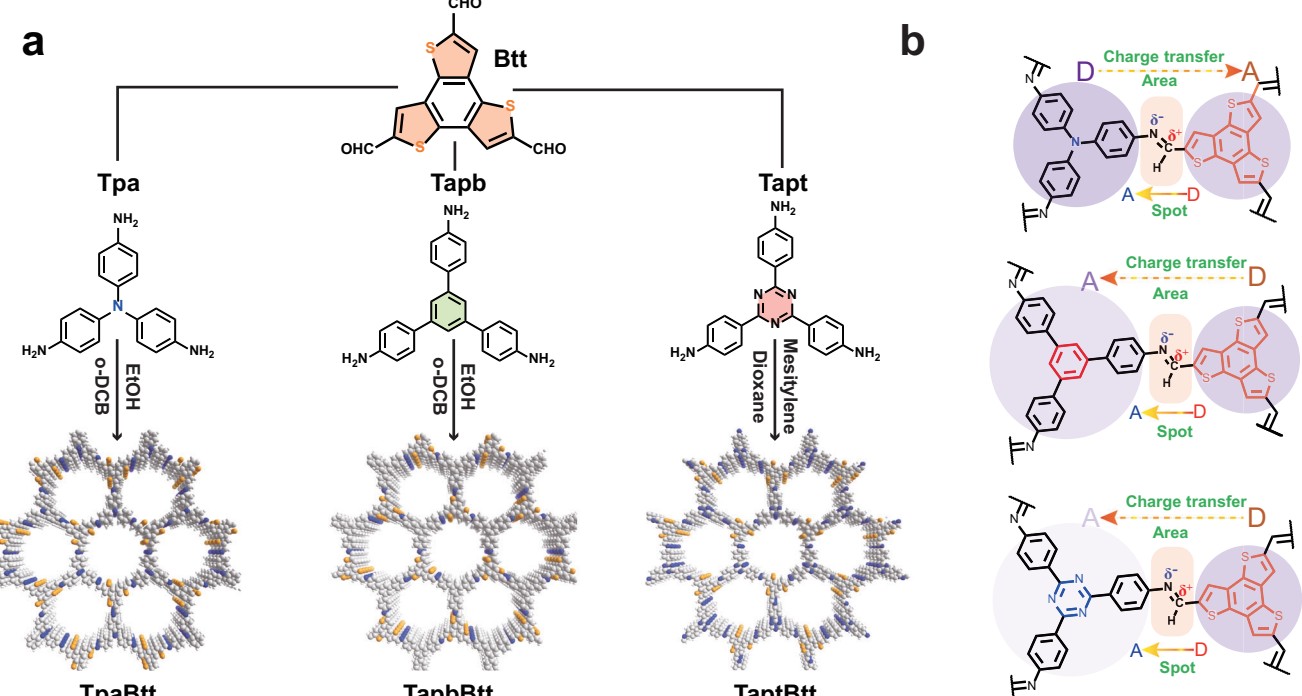

**Fig. 1 | Synthesis of Btt-based COFs with the directionality and energy difference of the charge transition. a** Synthesis of TpaBtt, TapbBtt and TaptBtt. **b** Directionality of electron transfer between functional motifs and imine linkage in TpaBtt, TapbBtt, and TaptBtt. The yellow dashed lines represent motifs and red dashed lines represent imine bonds. The shade of the color represents the difference in energy.

concentrated on the electron-acceptor fragments near the imine bond. The unique electronic structure of TaptBtt provides a sufficient driving force for the synchronous 2e⁻ WOR and 2e⁻ ORR, significantly lowering the Gibbs free energy of OH* and OOH* intermediates for $H_2O_2$ generation.

## Results

### Electronical and structural characterization of COFs

A mixture containing Btt and Tpa (or Tapb, Tapt) with a molar ratio of 1:1 was adopted via Schiff-base polycondensation under solvothermal conditions, producing TpaBtt, TapbBtt, and TaptBtt. The highest occupied molecular orbital (HOMO) and lowest unoccupied molecular orbitals (LUMO) for the three COFs are predicted and displayed in Supplementary Fig. 1. The HOMO and LUMO orbitals of TpaBtt are concentrated on the electron-donor Tpa unit (D) and the electron-acceptor Btt unit (A), respectively. However, the distribution of TapbBtt and TaptBtt in non-electron-occupied LUMO orbitals indicates that electrons in BTT are more likely to transfer to Tapb and Tapt upon excitation. This indicates that the intramolecular donor–acceptor (D-A) structure between the two functional motifs in COFs is profitably constructed. Based on the time-dependent density functional theory simulations, we calculated transition energies and probabilities of each excited state for all COFs. The strongest oscillator strength on fragments of TpaBtt, TapbBtt and TaptBtt indicated that the S1 state has mostly HOMO to LUMO transition (Supplementary Table 1). Energetic levels of several molecular orbital are equal to or lower than the HOMO, while almost all electrons are contributed by the LUMO in most of transition, indicating that the electronic configuration of LUMO nearly represents photogenerated electron composition.

Since the linkage of COFs serves as the connection and transport bridge, the D-A structure within linkage from the perspective of atomic scale should be considered, i.e., carbon of imine bonds as the D unit and nitrogen as the A unit[29,30]. In this work, we referred to the imine bonds as the "atomic spot" and the functional building blocks as the "molecular area". As shown in Fig. 1b, the electron-donor Tpa motif (D) is connected to the "anionic" nitrogen atom, and the electron-acceptor Btt motif (A) is coupled with the "cationic" carbon atom in the imine bond of TpaBtt. This makes the D-A direction of the imine linkage opposite to the direction of electron transfer between the molecular motifs. In contrast, TapbBtt and TaptBtt are in the same direction (D-A-type imine COFs), forming periodic and unhindered modes of charge transport. The energy necessary for the D-A-type imine COFs to twist the same angle in excited state is reduced, resulting in an increase in the energy difference of the groups connected near the imine bond. Hence, the imine linkage of Btt-based COFs is not only endowed with photoreactivity, but also synchronously regulates the charge transfer directionality and the intramolecular donor-acceptor energy difference in these COFs, further affecting the utilization of charge carriers. This combination of line-regions between linkages and linkers (termed as a concept of uniport "atom spot-molecular area") via a dual-donor-acceptor mechanism in a periodic framework is proposed to vary the photosynthesis performance and reaction pathways.

Molecular segments of the three COFs were used to verify that the push-pull interactions occur in the "atomic spot-molecular region". The results of photoelectrochemistry confirmed that the TaptBtt-fragment presents the upper charge separation capability (Supplementary Fig. 2). Furthermore, the fluorescence lifetimes of the TaptBtt-fragment (1.40 ns) were obviously higher than those of the TapbBtt-fragment (1.19 ns) and TpaBtt-fragment (0.74 ns), indicating that TaptBtt-fragment can more availably suppress e⁻-h⁺ recombination (Supplementary Fig. 3). To gain insight into the electron transfer direction for fragments, in-situ X-band electron paramagnetic resonance (EPR) experiments were then conducted (Supplementary Fig. 4). The relative intensity (0.31) of EPR signal for the TpaBtt-fragment in dark and light conditions is obviously higher than that of TapbBtt-

fragment (0.29) and TaptBtt-fragment (0.23). The lower relative strength for TaptBtt-fragment is probably related to the ability of ground state charge transfer from the acceptor unit to the donor unit[31–34], which is dominated by the integrated interaction of the directivity of the double D-A structure and the energy difference in these molecular fragments. Integrating photoelectrochemical measurements and photoluminescence lifetime with EPR results, TaptBtt-fragment exhibits more efficient charge transfer due to the favorable push-pull effects between the intramolecular motif and imine bond. Therefore, it can be concluded that the push–pull interaction of the "atomic spot-molecular area" occurs in aperiodic segments and COFs.

The crystalline construction of the COFs was confirmed by powder X-ray diffraction (PXRD), as shown in Fig. 2a. The Pawley refinement demonstrated the good fit of the eclipse stacking model (AA stacking) for three COFs (Supplementary Table 2)[35]. The optimized PXRD displayed the (100) reflection of TpaBtt, TapbBtt and TaptBtt at $2\theta = 5.43°$, 4.84°, and 4.76°, respectively (Supplementary Fig. 5). The intensity is substantially increased and becomes sharper for TpaBtt, TaptBtt and TapbBtt, with full width at half-maximum (FWHM) values of 0.51°, 0.39°, and 0.47°, respectively (Supplementary Fig. 6 and Supplementary Note 1). According to the Scherrer equation[36,37], the lowest FWHM means that TapbBtt has the maximum π-conjugated and ordered degree. Furthermore, the (001) plane of the three COFs at the peak position of ~26°, contributes to weak long-range order with an interlayer stacking of ~3.1 Å along the c direction. Fourier transform infrared spectroscopy (FTIR) showed stretching vibrations at 1579, 1617, and 1618 cm⁻¹ for TpaBtt, TapbBtt, and TaptBtt, respectively, which correspond to C=N bond that appeared (Supplementary Fig. 7), in contrast with the pure organic build blocks.

The chemical structures of the three COFs were further verified by solid-state ¹³C cross-polarization magic angle spinning nuclear magnetic resonance spectroscopy at the molecular level. As shown in Supplementary Fig. 8, the lower-field signal at ~157 ppm can be assigned to the carbon of imine bond for TpaBtt and TapbBtt[38]. The characteristic carbon signal of imine group for TaptBtt was observed at ~151 ppm, and the relatively high peak intensity is due to its coincidence with the position of benzene ring carbon bonded with nitrogen[39]. X-ray photoelectron spectroscopy (XPS) of the three COFs displayed C, N, O and S elements (Supplementary Figs. 9 and 10 and Supplementary Note 2). The C1s peak was divided into S-C=C (thiophene ring), C-C/C=C (sp² aromatic carbon) and C=N bonds (represented imine linkage)[40]. These results provide solid evidence that three kinds of COFs with favorable crystallinity and porosity were successfully prepared. From scanning electron microscopy images and transmission electron microscopy images (Fig. 2b, Supplementary Figs. 11–13, and Supplementary Note 3), TaptBtt presents larger raised burrs and thinner layers, and clear lattice fringes with a spacing distance of 0.31 nm were found at its (001) plane. However, TapbBtt is more likely to form dendritic aggregates, with tiny folds clearly visible on the surface of the branches. These tiny folds may be responsible for TapbBtt having a large surface area (1492.4 m² g⁻¹), compared with TpaBtt and TaptBtt (850.9 and 994.9 m² g⁻¹) in Supplementary Table 3. Additionally, nonlocal density function theory indicated that they have dominant pore widths at 1.20, 1.45, and 1.51 nm, respectively (Supplementary Fig. 14). Therefore, it was concluded that the larger raised burrs and larger pore size improve the exposure for $O_2$ in the photocatalytic reactions.

### Photocatalytic $H_2O_2$ production

For photocatalytic $H_2O_2$ synthesis, both pure water and air were utilized as the hydrogen and oxygen sources. Figure 2c shows the photocatalytic activity of TpaBtt, TapbBtt, and TaptBtt. A clear linear relation between $H_2O_2$ generation and irradiation time for TaptBtt was found compared with the other two COFs. After 90 min of light irradiation, the $H_2O_2$ concentration in the presence of TaptBtt reached

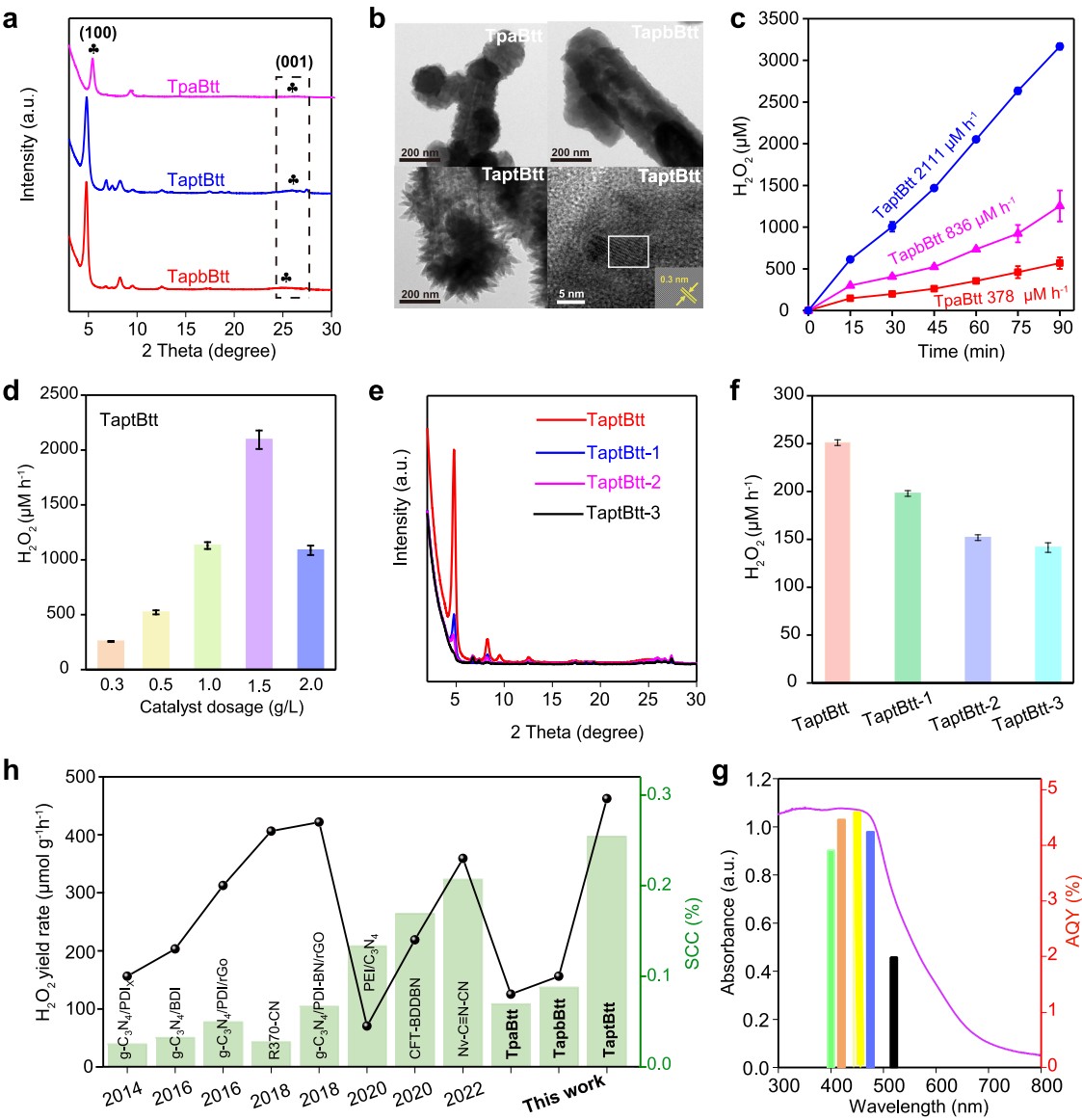

**Fig. 2 | Structural characterization and H$_2$O$_2$ evolution performance of Btt-based COFs.** **a** PXRD patterns of TpaBtt, TapbBtt, and TaptBtt. **b** TEM images of TpaBtt, TapbBtt and TaptBtt. **c** Photocatalytic activity of TpaBtt, TapbBtt and TaptBtt for H$_2$O$_2$ generation in pure water without any sacrificial agents. Conditions: water (10 mL), catalyst (15 mg), 300 W Xe lamp, $\lambda > 420$ nm. **d** Effects of TaptBtt dosage on H$_2$O$_2$ synthesis in water. **e** PXRD patterns of various degrees of crystallinity in TaptBtt. **f** Effects of TaptBtt crystallinity on H$_2$O$_2$ synthesis in water.

Conditions: water (50 mL), catalyst (15 mg), 300 W Xe lamp, $\lambda > 420$ nm. **g** Apparent quantum efficiency of TaptBtt. Conditions: water (60 mL), catalyst (75 mg), 300 W Xe lamp, $\lambda = 400, 420, 450, 475,$ and 520 nm. **h** Solar-to-chemical conversion efficiency of TaptBtt (conditions: water (60 mL), catalyst (75 mg)) under simulated AM 1.5 G sunlight irradiation and efficiency comparison of this work with other reported photocatalysts.

3167 μM (31.67 μmol), which was ~5.5 and 2.5 times higher than that of TpaBtt (568 μM and 5.68 μmol) and TapbBtt (1254 μM and 12.54 μmol), respectively. The observed trend was consistent with the intrinsic electronics of three COFs based on theoretically calculated results. The photosynthetic rate of H$_2$O$_2$ under different catalyst dosages was evaluated in Fig. 2d and Supplementary Fig. 15. An optimized photocatalyst concentration of 1.5 g L$^{-1}$ was obtained, corresponding to 378 μM h$^{-1}$ (3.78 μmol h$^{-1}$ and 252 μmol g$^{-1}$ h$^{-1}$) for TpaBtt, 836 μM h$^{-1}$ (8.36 μmol h$^{-1}$ and 557 μmol g$^{-1}$ h$^{-1}$) for TapbBtt and 2111 μM h$^{-1}$ (21.11 μmol h$^{-1}$ and 1407 μmol g$^{-1}$ h$^{-1}$) for TaptBtt. Excessive COFs inhibiting the ability to absorb light could cause the decrease in H$_2$O$_2$ concentration. The optimal performance of TaptBtt is much higher than that of TpaBtt and TapbBtt, and surpasses that of previously reported non-metal COFs and other nonmetal-heteroatom-doped C$_3$N$_4$-based photocatalysts (Supplementary Table 4)[41–47]. To demonstrate the influence of the ordered degree in pure COFs on the

production of H$_2$O$_2$, TaptBtt with different crystallinities was prepared by adjusting the ratios of mesitylene and 1,4-dioxane in the mixture, corresponding to 3:3, 2:4, 4:2, and 6:0 for TaptBtt, TaptBtt−1, TaptBtt-2, and TaptBtt-3, respectively. The crystallinity follows the order (Fig. 2e and Supplementary Fig. 16) of TaptBtt > TaptBtt−1 > TaptBtt-2 > TaptBtt-3. The crystallinity of TaptBtt is positively correlated with the yield of H$_2$O$_2$ (Fig. 2f), revealing that the long-ranged ordered structure of TaptBtt may assist H$_2$O$_2$ photosynthesis. A higher crystalline TapbBtt, due to its π columns, allows for quick exciton migration and hole transport along the π-conjugated direction, greatly retarding the backwards reverse recombination of charge. In contrast, the lower crystallinity in TaptBtt−1,2,3 cannot efficiently prevent backwards charge recombination, resulting in possible dissipation of the photoexcited states[48,49]. TaptBtt displayed an apparent quantum yield of 4.6 % at 450 nm, and a solar-to-chemical conversion efficiency of 0.296 % (Fig. 2g).

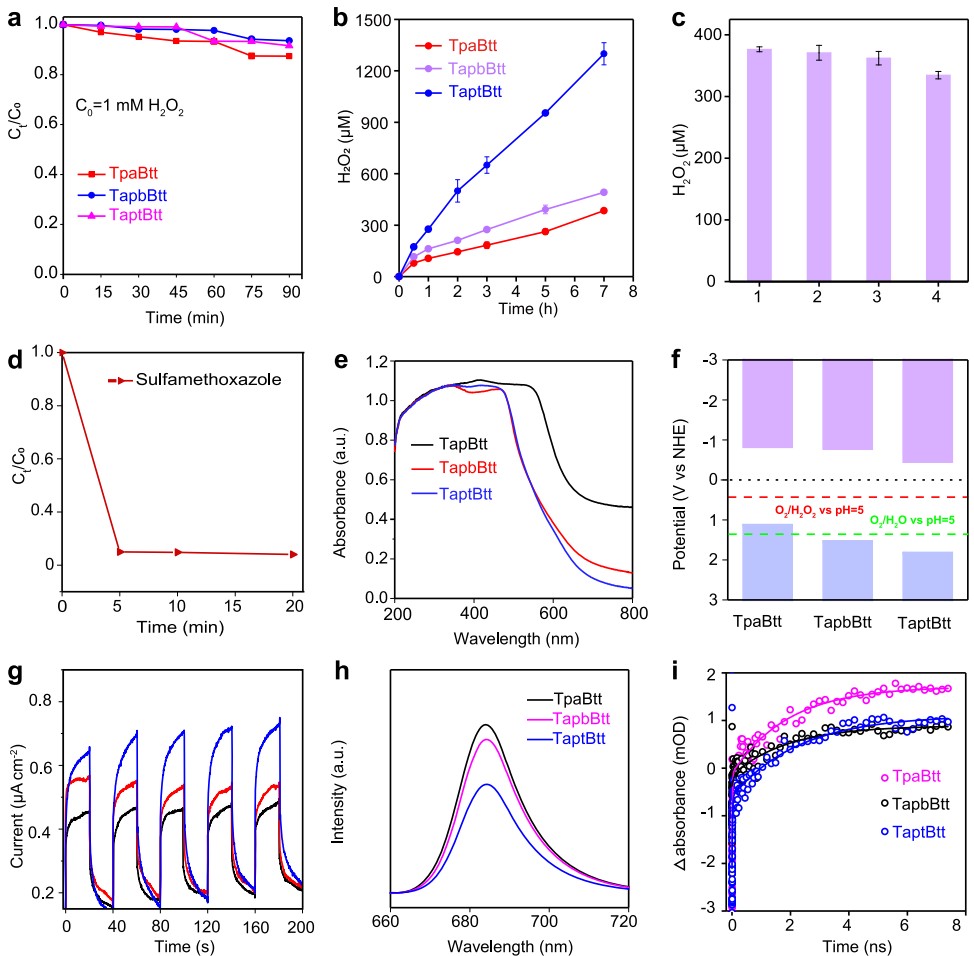

**Fig. 3 | Stability, implication and photoelectrochemical analysis of Btt-based COFs. a** Photocatalytic decomposition of $H_2O_2$ ($C_0 = 1$ mM) in pure water under an Ar atmosphere over TpaBtt, TapbBtt and TaptBtt. **b** Photocatalytic $H_2O_2$ production of TaptBtt for 7 h under simulated AM 1.5 G sunlight irradiation. Conditions: water (60 mL), catalyst (50 mg). **c** Stability measurement of TaptBtt for $H_2O_2$ generation in pure water. Conditions: water (50 mL), catalyst (15 mg), 300 W Xe lamp, $\lambda > 420$ nm. **d** Sulfamethoxazole decomposition directly using produced $H_2O_2$ solution via Fenton reaction. **e** UV–vis DRS of TpaBtt, TapbBtt, and TaptBtt. **f** Energy band values of these three COFs. the red and green lines represent for 2e⁻ ORR and 2e⁻ WOR, respectively. **g** Transient photocurrents of COFs under $\lambda > 420$ nm. **h** Photoluminescence spectra of three COFs. **i** Corresponding kinetics of characteristic fs-TA absorption bands observed at 540 nm for the spectra of TpaBtt, TapbBtt, and TaptBtt.

The final concentration of $H_2O_2$ depends on the dynamic equilibrium of generation-decomposition of $H_2O_2$ over the catalyst for a long time. The $H_2O_2$ concentration remained over 90% under continuous irradiation for 90 min (Fig. 3a). Compared with TpaBtt and TapbBtt, continuous and steady $H_2O_2$ evolution was observed over 7 h for TaptBtt, reaching to 80 µmol (Fig. 3b). To be practically useful, the long-term photostability of catalysts is essential. We therefore tested the photostability of TaptBtt using a continuous approach (96 h) in pure water. As shown in Supplementary Fig. 17, the photocatalytic $H_2O_2$ production rate of TaptBtt reaches to 580 µmol, which is higher than 330 µmol for SonoCOF-F2 under same conditions[50]. Although the rate of $H_2O_2$ formation began to slow after 48 h, the total amount continued to rise. The $H_2O_2$ production rate of TaptBtt was well preserved after four consecutive cycles (Fig. 3c), indicating enticing photocatalytic stability. Meanwhile, the crystallinity and chemical structure of three COFs were still maintained, as seen in the PXRD (Supplementary Fig. 18) and FTIR results (Supplementary Fig. 19). The presence of sacrificial reagents or buffers normally limits the direct utilization of $H_2O_2$ for environmental implications[51]. In this non-sacrificial system, the separated $H_2O_2$ was directly used to degrade sulfamethoxazole in wastewater. A fast decomposition with a removal efficiency of 72% was obtained within 5 min via the Fenton reaction (Fig. 3d). This result revealed that the photocatalytically produced $H_2O_2$ solution could be directly applied for environmental remediation.

## Thermodynamic and kinetic analysis of $H_2O_2$ origination

The precondition for photocatalytic $H_2O_2$ generation involves thermodynamics requirements. The absorption edges of TpaBtt, TapbBtt, and TaptBtt, shown in UV/visible diffuse reflectance spectrum (UV/Vis-DRS), correspond to 645 nm, 578 nm, and 590 nm, respectively (Fig. 3e). Therefore, it can be deduced that the wide difference in photocatalytic $H_2O_2$ production in the three COFs is not dominated by the crystallinity, surface area and visible-light absorption.

Subsequently, the band gap energies ($E_g$) of TpaBtt, TapbBtt, and TaptBtt, from the plots of $(ah\nu)^2$ vs. photon energy ($h\nu$), were counted as 1.95, 2.32, and 2.29 eV, severally. The valence band positions of TpaBtt, TapbBtt, and TaptBtt, calculated from the valence-XPS spectra, were 1.13, 1.54, and 1.83 eV, respectively (Supplementary Fig. 20). As shown in Fig. 3f, the band structure of TaptBtt was thermodynamically sufficient for the synchronous synthesis of $H_2O_2$ from $H_2O$ oxidation ($E_{H_2O_2/H_2O} = +1.76$ eV and $E_{O_2/H_2O} = 1.23$ eV vs. pH = 0) and $O_2$ reduction ($E_{O_2/H_2O_2} = +0.70$ eV vs. pH = 0)[52,53]. However, in the actual reaction the pH of our solution is ~5.0. Thus, according to the Nernst equation, the above values are offset to a certain extent ($E_{H_2O_2/H_2O} = +1.47$ eV; $E_{O_2/H_2O} = 0.94$ eV; $E_{O_2/H_2O_2} = +0.41$ eV vs. pH = 5). As opposed to TpaBtt

and TapbBtt, TaptBtt also demonstrated better wettability (Supplementary Fig. 21), which guarantees satisfactory dispersion in water for $H_2O_2$ photosynthesis.

The separation and recombination of photogenerated carriers is another important factor. According to Fig. 3g, TaptBtt exhibited a photocurrent density (i-T), that was markedly higher than that of TpaBtt and TapbBtt. TaptBtt had an upper charge separation capability and contained more available surface photogenerated carriers for solid−liquid interfacial reactions[54,55]. Electrochemical impedance spectroscopy (EIS) revealed that TaptBtt showed a lower charge transference resistance, which resulted in the faster transfer of electrons in the interface (Supplementary Fig. 22). The tendency of the photoelectrochemical results is positively correlated with that of the $H_2O_2$ yield. There is various accumulation of surface charge carriers on the surface of the catalyst for participation in the evolution of both intermediates and $H_2O_2$.

Furthermore, steady-state photoluminescence (PL) spectra exhibited obvious PL quenching of TaptBtt, as shown in Fig. 3h, compared with TpaBtt and TapbBtt. The fs-TA spectra of the three COFs (Supplementary Fig. 23) form a wide negative feature at 575 nm, assigned to ground state bleaching and stimulated emission, while the positive absorption band at 650 nm belongs to excited state absorption (ESA). The dynamics of excited state relaxation are mainly determined by the magnitude of intramolecular charge-transfer in molecules[56]. As a consequence, the peak shift at progressively increasing time delays could explain the charge-transfer character of these push-pull units in COFs[57]. As shown in Supplementary Fig. 23, compared to that of TpaBtt, the ESA and SE peaks for TaptBtt have obvious redshift amplitudes (black arrow). This results from the electron-deficient N-bridging of imines to the electron-acceptor unit, further demonstrating that COFs can achieve efficient photogenerated charge transfer using the push-pull mechanism from energy difference. Consequently, a global target analysis is used in three COFs, in which the initial Franck-Condon state splits into excited states and rapidly reaches the bound excitonic state (BE) and charge separation state (CS) through internal transformation. The exciton under the BE state is trapped, localizing on a single edge of the COFs, and the electron and hole under the CS state reside on separate motif edges either by intra- or interlayer charge transfer to increase their exciton radius, reduce their coulombic force, and prolong their persistence in the excited state[58]. The fitting dynamics show the variation of two decay time constants in Fig. 3i, where the short lifetime corresponds to the ascending component of TA and the intermediate recombination lifetime of the exciton trapped in the BE state, and the other component is the separation of the exciton into the SC state with a longer recombination lifetime[59–61]. The values of $\tau_1$ and $\tau_2$ are 13.8 ps and 1925.2 ps for TpaBtt, and 15.7 ps and 2283.8 ps for TaptBtt. The $\tau_2$ lifetime in TaptBtt is much longer-lived than that of TpaBtt, which is accountable for the greater charge separation capability. The results for PL and TA indicate that the channels of charge transfer between triazine motif and benzotrithiophene motif are accessible via the dual D-A structure of imine linkage in TaptBtt, leading to efficient suppression of photoexcited charge recombination.

It has recently been reported that the protonation of imine bonds in COFs could reverse the direction of charge transfer, improving their photocatalytic performance[30]. Inspired by this result, we think that the charge reversal caused by protonation is directly related to the energy difference of the line-region combination between linkages and linkers (Supplementary Fig. 24). Thus, all three COFs were protonated by ascorbic acid to afford TpaBtt-AC, TapbBtt-AC, and TaptBtt-AC, and their FITR spectra demonstrated that the protonation was successful (Supplementary Fig. 25). A new peak appeared at around 1800 cm$^{-1}$ (broad), assigned to the C=NH$^+$ bond[62]. Subsequently, the photocatalytic $H_2O_2$ evolution performance of all COFs was examined (Supplementary Fig. 26). A value was obtained for the $H_2O_2$

concentration by TpaBtt-AC (5.8 µmol h$^{-1}$), which was nearly three times higher than that of pristine TpaBtt, followed by that of TapbBtt-AC (7.2 µmol h$^{-1}$), and nearly 1.8 times higher than that of pristine TapbBtt. However, $H_2O_2$ production in TaptBtt was essentially unchanged, and even slightly decreased. We further probed the difference in the separation and recombination of carriers for these COFs by i-T and EIS. Obviously, TpaBtt-AC and TapbBtt-AC showed a higher current density than that of their pristine COFs (Supplementary Fig. 27a, b), while TaptBtt-AC exhibited the opposite trend (Supplementary Fig. 27c). A similar phenomenon was also observed in the EIS tests (Supplementary Fig. 27d–f). Meanwhile, the results of fs-TA spectra before and after protonation provide solid evidence (Supplementary Figs. 28 and 29 and Supplementary Note 4). Based on the above results, it was concluded that the performance of TpaBtt can be improved through protonation of imine bonds. Intrinsically, the protonation of imine bonds leads to the inversion of charge transfer orientation in an intramolecular way. In terms of codirectional charge transfer for TaptBtt, the larger energy difference in the line-region combination between linkages and linkers was difficult to overcome by protonation of the imine bond to reverse the charge transfer orientation. This result directly indicates that TaptBtt exhibits greater energy transfer between motifs than the other two COFs.

## Photocatalytic reaction pathways of $H_2O_2$ evolution

To verify that photoinduced holes and electrons participate in photocatalysis, the yields of $H_2O_2$ under various conciliatory conditions were determined. When the holes were trapped in the presence of $CH_3OH$ and air (Fig. 4a), only the $H_2O_2$ production for TaptBtt presented a downwards trend, while those for TpaBtt and TapbBtt showed an upward trend (Supplementary Fig. 30). This phenomenon indicates that holes generated from TpaBtt and TapbBtt may not be directly involved in the photocatalytic production of $H_2O_2$. When $O_2$ was replaced by Ar in the reaction system, the yield of $H_2O_2$ decreased significantly for the three COFs. Compared with the Ar-only condition, the yield of $H_2O_2$ increased in TaptBtt when the electron-trapping agent ($KBrO_3$) was added in the presence of Ar. However, the $H_2O_2$ concentration was almost undetectable for TpaBtt and TapbBtt under the same conditions. This result implies that a four-electron water oxidation process may have occurred in TpaBtt and TapbBtt, while TaptBtt could directly utilizes holes and electrons synchronously during the production of $H_2O_2$.

Rotating disk electrode and rotating ring-disk electrode (RRDE) measurements were used to examine the 4e$^-$ WOR and 2e$^-$ WOR routes of COFs. As shown in Fig. 4b and Supplementary Fig. 31, the average electron transmission numbers participating in the ORR were 1.62, 1.57, and 1.71 for TpaBtt, TapbBtt, and TaptBtt, respectively. Compared with TpaBtt and TapbBtt, the number of metastasizing electrons of TaptBtt approaches 2, indicating that $H_2O_2$ selectivity is higher under the same conditions. The results also reveal the advantage of the line-region combination between imine linkage and Tapt/Btt linker in TaptBtt for 2e$^-$ ORR pathway. During the RRDE test, the incremental disk currents (Supplementary Fig. 32a) with potentials higher than 1.4 V (solid lines) imply that water oxidation occurs at the rotating disk electrode for TpaBtt, TapbBtt and TaptBtt. No reduction current was detected for three COFs at the Pt ring electrode, demonstrating that TpaBtt, TapbBtt, and TaptBtt could not generate $O_2$ via water oxidation (4e$^-$ WOR process). TaptBtt might have the ability to directly exploit holes (Fig. 4a). When the potential provided at the ring electrode was altered to an oxidative potential of +0.6 V, a weak oxidation current can be detected for TaptBtt, due to the oxidation of $H_2O_2$ under the ring electrode (Supplementary Fig. 32b). As the RRDE results indicated that TpaBtt and TapbBtt cannot produce $O_2$, and the factors that cause the weak $H_2O_2$ production under the condition of Ar should be further explored.

Subsequently, $H_2^{18}O$ was used in photocatalytic tests to further identify the two/four-electron water oxidization. As shown in

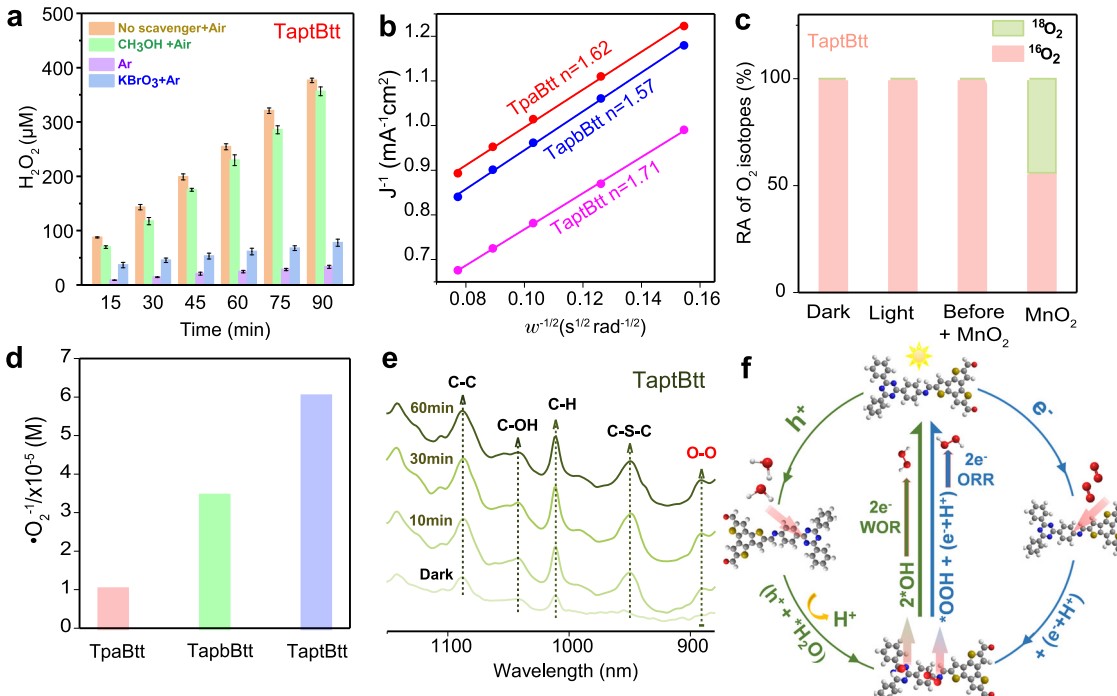

**Fig. 4 | Reaction pathways and mechanisms of H₂O₂ photosynthesis. a** Amount of H₂O₂ generated on TaptBtt in CH₃OH (10% V/V, as the hole trapping agent), H₂O₂ produced in AR and KBrO₃ (0.01 M). Conditions: water (50 mL), catalyst (15 mg), 300 W Xe lamp, λ > 420 nm. **b** Kouteckly-Levich plots obtained by RDE tests versus Ag/AgCl. **c** ·O₂⁻ yields of TpaBtt, TapbBtt and TaptBtt detected by NBT method under light conditions. **d** ¹⁸O₂ isotope experiment for TaptBtt. **e** In-situ DRIFT spectra of TaptBtt. **f** Mechanism of TaptBtt for photocatalytic H₂O₂ formation. The white, gray, blue, yellow and red spheres refer to hydrogen, carbon, nitrogen sulfur and oxygen, respectively.

Supplementary Fig. 33, none of the three COFs was detected for ¹⁸O₂ production in the first stage (including dark, light and before addition of MnO₂), while all did in the second stage-decomposition of photogenerated H₂O₂ by MnO₂ (Fig. 4c and Supplementary Fig. 34). However, we can clearly see that the ratio of two types of oxygen (¹⁸O₂ and ¹⁶O₂) is significantly different after H₂O₂ decomposition in the second step. The ratio of ¹⁸O₂ and ¹⁶O₂ is 1:4.8 (close to the four-electron water oxidation process, Eq. (7) for TpaBtt and TapbBtt, while the ratio is 1:1.2 (close to the two-electron water oxidation process, Eq. (3) for TaptBtt[51,63,64]. In addition, a difference trend was observed in the H₂O₂ concentration after a sacrificial agent was added, and H₂O₂ can be detected under an Ar atmosphere for TpaBtt, TapbBtt, and TaptBtt (Fig. 4a and Supplementary Fig. 30)[47,51]. For TpaBtt and TapbBtt, it was reasonably concluded that this four-electron process is involved in the synthesis of hydrogen peroxide. The oxygen produced by the four-electron water oxidation is tiny and may be adsorbed on the surface of COFs, and then directly used for the formation of H₂O₂. Thus, the four-electron process provides little contribution to H₂O₂ production for TpaBtt and TapbBtt. This also explains the absence of O₂ in the RRDE (Supplementary Fig. 32a) and oxygen-producing isotopes (Supplementary Fig. 33). Therefore, these results provide solid support that H₂O₂ photosynthesis undergoes 2e⁻ ORR and 4e⁻ ORR for TpaBtt and TapbBtt, while TaptBtt has 2e⁻ ORR and 2e⁻ WOR dual processes with higher atomic efficiency.

There are two pathways by which WOR and ORR generate H₂O₂ from water and air via 2e⁻ redox[63], corresponding to Eqs. (1–3) and Eqs. (4–6). Whether one-step (Eqs. (3) and (6)) or two-step (Eqs. (1, 2) and Eqs. (4, 5) occurs through the OH and ·O₂⁻ intermediates can be checked. Therefore, a 5,5-dimethyl-pyrroline N-oxide probe was utilized as a free radical spin-capturer to measure ·O₂⁻ and ·OH. The ·O₂⁻ signal could be detected in TpaBtt, TapbBtt and TaptBtt under light irradiation, but there was no ·OH signal (Supplementary Fig. 35). The results indicate that H₂O₂ is mainly produced via a 2e⁻ two-step routine mediated by ·O₂⁻ in the presence of three COFs, while TaptBtt involves an extra process of H₂O₂ formation through 2e⁻ one-step oxidation of water. The ·O₂⁻ was quantified via a recognizable reaction with nitro blue tetrazolium (NBT, Supplementary Fig. 36). The photocatalytic yield of ·O₂⁻ by TaptBtt was 6.02 × 10⁻⁵ M, markedly higher than that of TapbBtt (3.44 × 10⁻⁵ M) (Fig. 4d), suggesting that the larger energy difference of intramolecular donor–acceptor in TaptBtt promoted the generation of ·O₂⁻ intermediate. Capture experiments of the active species were carried out (Supplementary Fig. 37). The addition of benzoquinone (BQ, ·O₂⁻ scavenger) significantly inhibited the yield of H₂O₂ for TpaBtt, TapbBtt, and TaptBtt. When tert-butanol (·OH scavenger) was added to the system, H₂O₂ production tended to increase. The carbon-centered radical (R·) produced by tert-butanol reacts with dissolved oxygen to form RO₂·, which spontaneously binds to form tetroxide intermediates, and finally splits to form H₂O₂[65].

$$H_2O + h^+ \rightarrow \cdot OH + H^+ \tag{1}$$

$$\cdot OH + \cdot OH \rightarrow H_2O_2 \tag{2}$$

$$2H_2O + 2h^+ \rightarrow H_2O_2 + 2H^+ \tag{3}$$

$$O_2 + H^+ + e^- \rightarrow \cdot OOH \tag{4}$$

$$\cdot OOH + e^- + H^+ \rightarrow H_2O_2 \tag{5}$$

$$O_2 + 2e^- + 2H^+ \rightarrow H_2O_2 \tag{6}$$

$$2H_2O + 4h^+ \rightarrow O_2 + 4H^+ \tag{7}$$

In-situ diffuse reflectance infrared Fourier transform (DRIFT) spectroscopy measurements were taken to monitor the interactions between the active sites and the intermediates. As shown in Fig. 4e and Supplementary Fig. 38, the characteristic stretching of O−O at ~892 cm$^{-1}$ appears under photocatalytic reaction, verifying the occurrence of two-step single-electron route[66,67]. The O-O bond serves as the key intermediate for $H_2O_2$ production in 2e$^-$ ORR. The intensity of the O-O peak follows the order of TaptBtt > TapbBtt > TpaBtt, matching well with the EPR and NBT results. The intensity of the peaks at 1012 cm$^{-1}$ and 1086 cm$^{-1}$, corresponding to the bending modes of C-H and C-C of the center ring on the three COFs, gradually increases. Besides, the C-S-C (951 cm$^{-1}$), belonging to the thiophene unit[68], is also gradually enhanced. The vibration of the C-C and C-S-C peaks indicates that occurrence of photoinduced electron transfer occurs between the thiophene of the Btt motif and the benzene ring fragments of Tapt motif. More importantly, vibrations were observed for C=N (1623−1627 cm$^{-1}$) and C=NH$^+$ (1567 cm$^{-1}$) for three COFs after photoexcitation, indicating that protonation occurs on imines (Supplementary Fig. 39). The peak at 1041 cm$^{-1}$, ascribed to the C-OH intermediate[63], confirms the presence of adsorbed OH* on TaptBtt. In terms of TaptBtt, water could react directly with two holes to form $H_2O_2$ upon exposure to light, while oxygen reacts with protons and electrons to form hydrogen peroxide via •$O_2^-$ (Fig. 4f). OOH*, an important intermediate state for the 2e$^-$ ORR, requires electrons on the surface of the materials. The benzene ring near the imide is the electron-absorbing unit for TapbBtt and TaptBtt. Therefore, we can reasonably infer that the active site is located near the imine bond of the electron-acceptor fragment through in-situ spectra.

## Theoretical analysis of intermediates for the overall process of $H_2O_2$ generation

Density functional theory (DFT) calculations were conducted to investigate the elementary step of $H_2O_2$ formation on the three COFs. The key to selectivity of 2e$^-$ ORR to $H_2O_2$ relied on the generation of OOH* and consequent hydrogenation[69–71]. The adsorption of $O_2$ and H$^+$ is a precondition for the formation of OOH* on the surface of photocatalysts. Therefore, the photocatalytic reduction of $O_2$ to $H_2O_2$ can be divided into four steps (Eqs. (S1−S4)). The optimized structures of three COFs with adsorbed intermediates are carefully given in Fig. 5a and Supplementary Fig. 40. $O_2$ adsorption falls into the electron-acceptor motif of COFs so that it facilitates access to electrons and protons for selectively forming the OOH* intermediate. This is consistent with the stretching of O-O characteristic peaks in the in-situ DRIFT spectra. Hirshfeld analysis reveals that the average charge of these carbon (C) atoms close to imine linkage is −−0.122 eV for TpaBtt (the thiophene ring in Btt) and increases to −0.119 eV for TaptBtt (the benzene ring of Tapt nears imine bond) (Supplementary Fig. 41). The C1 atom of TaptBtt has a positive value of 0.041 eV, which indicates a strong ability to extract electrons (i.e., Lewis acidity), beneficial for in-plane charge transfer[72]. The Hirshfeld charge (−0.09 eV) of $O_2$ on TaptBtt decreases in relative to that of TpaBtt (−0.143 eV) and TapbBtt (−0.109 eV) (Supplementary Fig. 42). The pH value of pure water in the reactive system was 5.60, and it then decreased to 5.12, 5.32, and 5.20 after TpaBtt, TapbBtt and TaptBtt were added, respectively. Thus, the imine bonds in COFs undergo protonation, and the extent of protonation is determined by the quantity and hydrophilicity of nitrogen sites that are available in the framework[73,74]. The protonation of imine bonds provides a favorable hydrogen source for the $O_2$*/H$^+$ step during the formation of the OOH* intermediate.

Correspondingly, the Gibbs free energy (ΔG) for each step involved in 2e$^-$ ORR was calculated, as shown in Fig. 5b. TpaBtt exhibited strong adsorption of the $O_2$* intermediate with a Δ$G_1$ value of −2.33 eV, terrifically restricting the desorption of the OOH* intermediate. The rate-limiting step (OOH* → $H_2O_2$) has an extremely high energy barrier with a value of 1.14 eV. The OOH* formed on the active

sites cannot be released in a timely manner from the surface of the catalyst, which affects the continuous utilization of the site. Although TapbBtt weakened the adsorption of $O_2$* intermediate with a Δ$G_1$ value of 0.28 eV, it exhibited a relatively high free energy (0.67 eV) during protonation to form the $O_2$*/H* intermediate. Optimally, TaptBtt simultaneously modulates the binding strength of $O_2$* and OOH* intermediates, thus promoting the 2e$^-$ ORR with a lower energy barrier of 0.57 eV. The real active sites of forming OOH* are located on the C atom of electron-acceptor fragments nearest the imine linkage (Supplementary Fig. 43) in TaptBtt. This also demonstrates that the electrons on the benzotrithiophene motif of TaptBtt are transported to the benzene ring of Tapt, promoting the photocatalytic oxygen reduction. These results are consistent with the in-situ DRIFT spectra. Figure 5c–e displays the charge difference density between the pivotal intermediate OOH* and active sites near imine bond of COFs. Compared with TpaBtt and TapbBtt, the charge redistribution of TaptBtt interacting with OOH* is more noteworthy, indicating that the line-region combination of imine linkages and linkers (Tapt and Btt) via a dual-donor-acceptor mechanism still determines the selective formation of OOH* with favorable binding ability in active sites.

For 2e$^-$ WOR pathway occurring on TaptBtt, the OH* generation is the largest vital step (Eqs. (S5 and S6)). The C atom (Site 4) in triazine units in TaptBtt shows the smallest Δ$G_1$ for the formation of OH* compared to other probable sites (Supplementary Fig. 44), indicating that the 2e$^-$ WOR occurs in the triazine units. The H$^+$ produced by water oxidation can be utilized by 2e$^-$ ORR and hydrogenation is accelerated effectively for TaptBtt, leading to notable activity towards $H_2O_2$ selectivity. It was confirmed that the local electronic properties changed significantly in the cooperation units of Btt and Tapt, thus regulating the binding strength of $O_2$* and OOH* intermediates. The $O_2$ reduction process occurs on the benzene ring while the water oxidation reaction takes place on the triazine units of TaptBtt. These functional motifs in TaptBtt with spatially separated redox species endow the selective formation of intermediates for highly efficient $H_2O_2$ production via synchronous 2e$^-$ WOR and 2e$^-$ ORR pathways.

Based on the above analysis, the overall process of $H_2O_2$ generation over three COFs is summarized below. Initially, 2e$^-$ ORR plays an important role to generating $H_2O_2$ in the three COFs due to the feasibility of thermodynamics. The partial contribution of 2e$^-$ WOR in TaptBtt system is also important for periodic cycles without a sacrificial agent, while the 4e$^-$ WOR has a little contribution to TpaBtt and TapbBtt. In the ORR process, i) COFs differentially adsorb dissolved oxygen under dark reaction and then undergo photoinduced in-plane electron transfer; ii) the formed $O_2$* on electron-acceptor fragment utilizes the transferred electrons and the protons from the adjacent imine bond to generate the OOH* intermediate; iii) protons in water are hydrogenated with OOH* to selectively produce $H_2O_2$. For the 2e$^-$ WOR process of TaptBtt, two $H_2O$ molecules are adsorbed onto the Tapt unit to form an intermolecular hydrogen bond[75], which is subsequently attacked by photoinduced holes to directly produce $H_2O_2$. The remaining two protons are finally utilized by ORR process, forming a complete cycle of $H_2O_2$ synthesis in Fig. 4f.

## Discussion

In summary, we rationally engineered benzotrithiophene-based covalent organic frameworks by regulating the electron distribution and transfer directionality for hydrogen peroxide photosynthesis. TaptBtt displayed attractive activity for $H_2O_2$ production with a yield rate of 2111 μM h$^{-1}$ (21.11 μmol h$^{-1}$ and 1407 μmol g$^{-1}$ h$^{-1}$) and a solar-to-chemical conversion efficiency of 0.296 %. The codirectional charge transfer and larger energy difference of the line-region combination between linkages and linkers in terms of dual-donor-acceptor structures in the periodic framework promote a satisfying energy band, favorable intermediate interaction, optimal reactive pathways and finally high-yield synthesis of $H_2O_2$. The collected $H_2O_2$ solution can be

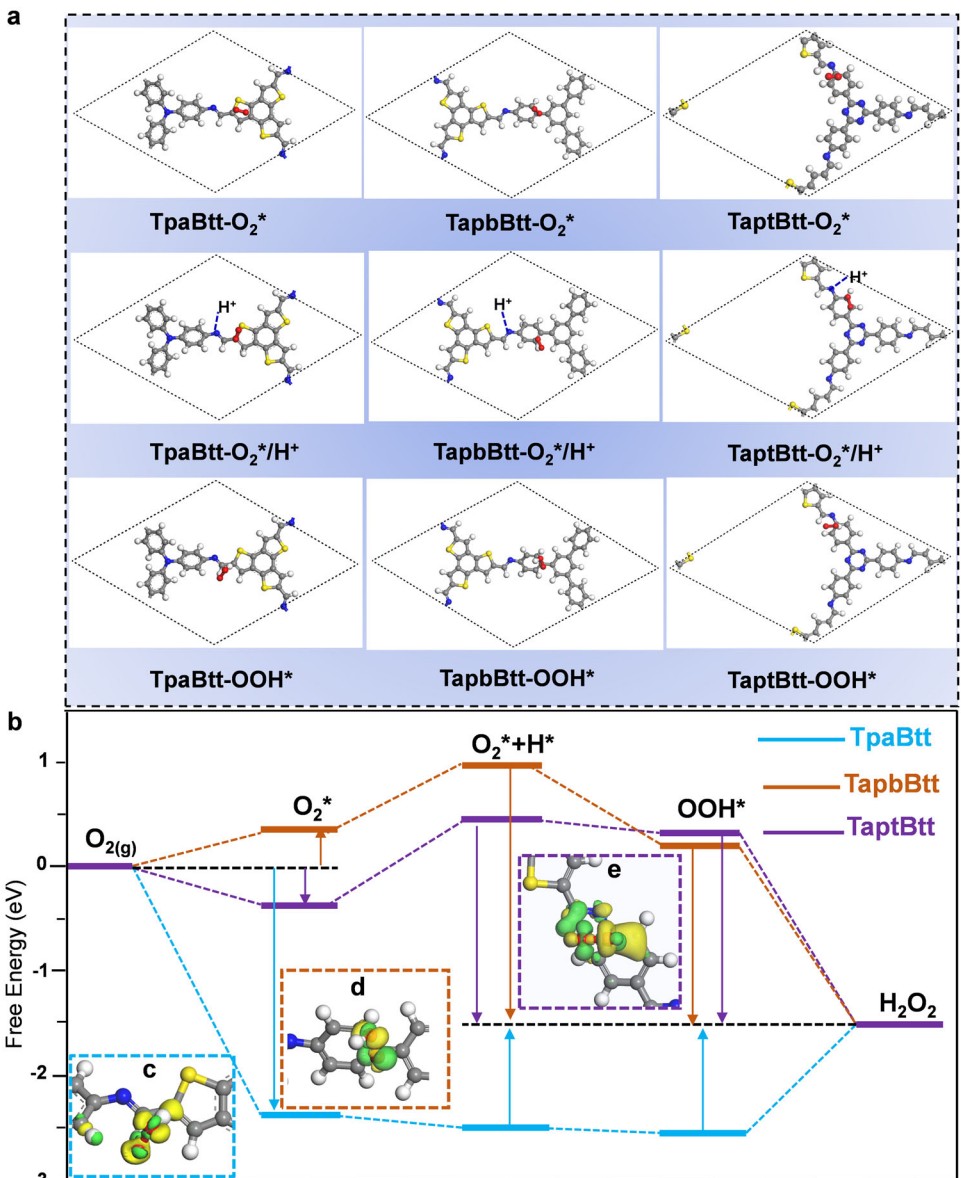

**Fig. 5 | Theoretical knowledge on the intermediates produced during $H_2O_2$ evolution with various free energies for Btt-based COFs. a** Different adsorption site a of $O_2$, $O_2/H^+$ and OOH* on TpaBtt, TapbBtt and TaptBtt, respectively. **b** Free energy profiles for photocatalytic $H_2O_2$ evolution reactions over three COFs. **c–e** Charge difference density between OOH* and adsorption sites on TpaBtt, TapbBtt and TaptBtt, respectively. Yellow represents the electron accumulation area, and the green represents the electron dissipation area. The white, gray, blue, yellow and red spheres refer to hydrogen, carbon, nitrogen sulfur and oxygen, respectively.

utilized for pollutant removal. DFT calculations revealed that Btt binding to different functional motifs (Tpa, Tapb and Tapt) could regulate electron distribution on C atoms near imine bonds, which can facilitate the in-plane charge transfer and mediate the binding strength of $O_2$* and OOH* intermediates during $2e^-$ two-step redox reaction. In addition, partial water oxidation promoted the protonation of TaptBtt, reducing the barrier for the formation of intermediates $O_2$*/H* and OOH*. This study provides insight into the push-pull effects between intramolecular motifs and linkage chemistry in polymers based on COF-based platforms towards highly efficient energy conversion and artificial photosynthesis.

## Methods
### Synthesis of TpaBtt
4,4,4-Triaminotriphenylamine (Tpa, 58.07 mg, 0.2 mmol) and benzo [1,2-b:3,4-b':5,6-b"]trithiophene-2,5,8-tricarbaldehyde (Btt, 66.1 mg, 0.2 mmol) were placed into a 10 mL Pyrex tube, and dissolved into o-dichlorobenzene (o-DCB, 3 mL) and n-butanol (3 mL) mixed solution ($v/v = 1:1$). After the above mixture was sonicated for 10 min, acetic acid aqueous solution (0.3 mL, 6 M) was added, and then the system sonicated again for 2 min. The tube was degassed by three freeze-pump-thaw cycles and then was sealed off and heated at 120 °C for 3 days. The powder collected was washed with N,N-dimethylformamide, methanol and ethanol several times, and dried at 120 °C under vacuum for 2 h to obtain TpaBtt sample in 87% isolated yield.

### Synthesis of TapbBtt
1,3,5-Tris(4-aminophenyl)benzene (Tapb, 70.2 mg, 0.2 mmol) and benzo[1,2-b:3,4-b':5,6-b"]trithiophene-2,5,8-tricarbaldehyde (Btt, 66.1 mg, 0.2 mmol) were placed into a 10 mL Pyrex tube, and dissolved into 1,2-dichlorobenzene (o-DCB, 3 mL) and n-butanol (3 mL) mixed solution ($v/v = 1:1$). After the above mixture was sonicated for 10 min, acetic acid aqueous solution (0.3 mL, 9 M) was added, and then the system was sonicated again for 2 min. The tube was degassed by three

freeze-pump-thaw cycles and then sealed off and heated at 120 °C for 3 days. The powder collected was washed with tetrahydrofuran and acetone several times, and dried at 80 °C under vacuum for 24 h to obtain TapbBtt sample in 82% isolated yield.

## Synthesis of TaptBtt

2,4,6-Tris(4-aminophenyl)−1,3,5-triazine (Tapt, 70.8 mg, 0.2 mmol) and benzo[1,2-b:3,4-b':5,6-b"]trithiophene-2,5,8-tricarbaldehyde (Btt, 66.1 mg, 0.2 mmol) were placed into a 10 mL Pyrex tube, and dissolved into mesitylene (3 mL) and 1,4-dioxane (3 mL) mixed solution ($v/v$ = 1:1). After the above mixture was sonicated for 10 min, acetic acid aqueous solution (0.5 mL, 6 M) was added, and then the system was sonicated again for 2 min. The tube was degassed by three freeze-pump-thaw cycles and was sealed off and heated at 120 °C for 3 days. The powder collected was washed with tetrahydrofuran and acetone several times, and dried at 80 °C under vacuum for 24 h to obtain TaptBtt sample in 85% isolated yield.

## Data availability

The data that support the findings of this study are available within the article and its Supplementary Information. Source data are provided with this paper.

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

## Acknowledgements

The authors gratefully acknowledge the financial support provided by the National Natural Science Foundation of China (No. 22178091 (H.W.), 72088101 (X.C.), 51708195 (H.W.), 51739004 (X.Y.) and 52202367 (X.W.), the Provincial Natural Science Foundation of Hunan (No. 2023JJ10012 (H.W.)), the Science and Technology Innovation Program of Hunan Province (No. 2022RC1120 (H.W.)), the National Key Research and Development Program of China (No. 2021YFC1910400 (L.T.)), the Nature Science Foundation of Jiangsu Province (No. BK20200711 (X.W.)), the Key Laboratory of Advanced Functional Composites Technology (No. 6142906210508 (X.W.)), the Fundamental Research Funds for the Central Universities (No. 531118010675 (H.W.)), and the Ministry of Education Singapore under its Academic Research Funds (No. RG3/21, RG85/22, and MOET2EP10120-0003 (Y.Z.)).

## Author contributions

C.Q. and H.W. came up with the original ideal and performed photo-catalytic research; X.W. helped with calculation part; B.S. and S.W. performed an experiment on isotopes; X.C., M.L., Y.M., S.W., C.F., and J.L. helped with the date interpretations; X.Y., Y.Z., and H.W. supervised the project; C.Q., X.W., L.T., and H.W. wrote the manuscript; all authors commented on the manuscript.

## Competing interests

The authors declare no competing interests.
