## [Peer Review File · Nature Communications]

Dual donor-acceptor covalent organic frameworks for hydrogen peroxide photosynthesisREVIEWER COMMENTS

Reviewer #1 (Remarks to the Author):

This work reports the synthesis of benzotrithiophene-based COFs with dual-donor-acceptor structures for photocatalytic generation of H₂O₂ from water and oxygen. They claimed that, by changing different monomers with various electron-acceptor capability, the photocatalytic performances can be enhanced, especially in the TaptBtt. Authors ascribed the enhanced performance to three reasons: satisfied energy band; facilitated in-plane charge transfer; favorable intermediate interaction. The present study contains several incorrect characterizations and lacks of critical evidences to support their claims. Therefore, the manuscript is not recommended for publication in Nat. Commun. Furthermore, there are some problems in this paper, several comments to be considered:

1. Unlike the TapbBtt and TpaBtt, the optimized structure of TaptBtt in the DFT calculation is in a different form. Please explain it.
2. Solid-state ¹³C NMR spectra should be provided. In addition, ³³S solid-state NMR is recommended.
3. The author pointed out that “the crystallinity of TaptBtt is positively correlated with the yield of H₂O₂ (Fig. 2f)”. This seems to contradict the result of “the crystallinity follows the order of TaptBtt > TaptBtt-1 > TaptBtt-2 > TaptBtt-3 (Fig. 2e and Fig. S11).” In addition, why is the TaptBtt the highest crystallinity.
4. The possible active sites are only differentiated by DFT calculations, and no operando spectroscopic characterizations are conducted to support the computational results. DFT calculations can sometimes lead to groundless conclusions.
5. The author claimed that “the VB values of TpaBtt and TapbBtt do not satisfy the oxidative potential of water oxidation (2e⁻ WOR)”. However, the results in Figure S18 showed that TpaBtt and TapbBtt could generate H₂O₂ in Ar atmosphere. It is puzzling.
6. The pore size distributions shown in Figure S9 indicate that there are two different pores in the COFs. Please explain it.

Reviewer #2 (Remarks to the Author):

The authors designed and synthesized three imine-linked COFs using the Btt building blocks. They found TaptBtt displayed high selectivity for photo-generating H₂O₂ owing to the homodromous charge transfer and large energy difference of line-region combination between linkages and linkers. This work proposed an interesting concept for the analysis of good photocatalytic performance. However, the manuscript is very hard to read. The logic is messed up. The key of this work is the claim of the homodromous charge transfer and large energy difference of line-region combination between linkages

and linkers in double donor-acceptor structures of periodic frameworks. The manuscript should answer the question of "how do the push-pull effects between intramolecular motif and linkage chemistry in polymer affords highly catalytic efficiency?" The D-A direction of the imine linkage and the direction of electron transfer between the molecular motifs should play a key role in photocatalytic performance according to the proposed rule by the authors. However, I could not see direct and clarified evidence for this. The authors do try their best to prove the reason that the TaptBtt has the best performance. Better wettability and faster transfer of electrons in the interface et al., for example. The direct evidence is missing. The Femtosecond time-resolved transient absorption (fs-TA) spectroscopy may give some support to this concept. However, no detailed analysis was given. I only found that the description of "The lower negative band of TapbBtt was observed, indicating that the exciton relaxation dynamics was different from other two COFs." I could not see a very clear explanation that TaptBtt has the best performance due to the homodromous charge transfer like the author claimed as the highlight of this work.

I am afraid that the realization can not reach Nature Communications' quality, and the claims are not supported. Thus, I cannot recommend this work for publication.

Besides, the following problems should be paid attention to.

1. The authors claimed intermolecular donor-acceptor existed in these COFs. If this description is correct after condensation of the two building blocks? "Intramolecular" may be more suitable. Please carefully check that.
2. In Figure 1b, Tpa motif was defined as an electron donor, while the authors described it as an electron acceptor in Page 7 and D short for donor was used as the abbreviation. Thus, Tpa motif as an electron donor was connected to the "anionic" nitrogen atom. This will lead to an opposite conclusion to the manuscript. Please carefully address this problem.
3. In page 7, the authors claimed the required excitation energy for TaptBtt transition is smaller than that of other two COFs. However, the band gap energy (E_g) of TpaBtt, TapbBtt, and TaptBtt was counted as 1.95, 2.32 and 2.29 eV, severally. TpaBtt exhibited a narrower bandgap.
4. As displayed in Figure 2e, the crystallinity follows the order (Fig. 2e and Fig. S11) of TaptBtt > TaptBtt-1 > TaptBtt-2 > TaptBtt-3. The authors claimed that the crystallinity of TaptBtt is positively correlated with the yield of H₂O₂. According to Figure 2f, TaptBtt exhibited the lowest yield of H₂O₂.
5. Page 6, "lowest occupied molecular orbitals (LUMO)" is wrong.

Reviewer #3 (Remarks to the Author):

The manuscript by Qin and co-workers is interesting but there are some issues that need addressing before publication:

1. Did the authors attempt to solve the structures of their COFs from the obtained PXRD patterns?
2. When analysing the ability of TaptBtt to drive oxygen reduction and the 2-electron oxidation of water

on page 13 the values used for the potentials of both these solution half-reactions appear to be the standard values tabulated for pH 0. However, the H₂O₂ synthesis experiments are not performed at pH 0 but pH ~ 5 (see page 20). In their analysis the authors should use the values shifted to pH 5 using the Nernst equation, which will be shifted to more negative values, rather than the pH 0 ones.

3. Ref. 25 is cited at the end of the sentence with the values of the oxygen reduction and the 2-electron oxidation of water potentials on page 13 but I think this is a mistake. Ref. 25: "Patterson AL. The Scherrer formula for X-Ray particle size determination. Phys. Rev. 56, 978-982 (1939)" appears to be about something completely different.

4. The authors should be careful to use their valence band edge measured for a COF in vacuum to analyse the ability of the material to drive half-reactions when immersed in water. The high dielectric constant of water relative to that of the COF will shift the valence band edge in water to more negative values (and the conduction band edge to more positive values). Taking this into account probably won't qualitatively change the picture if one also takes the pH shifts discussed above into account, because these make the water oxidation reaction easier, compensating for the fact that the band edges will shift to more positive values in water.

5. Following on from the above, TpaBtt and TapbBtt cannot oxidise water but still produce H₂O₂. Do they produce H₂O₂ via a different mechanism not involving water oxidation, as suggested on page 16 and page 18, or would this suggest an issue with the XPS measurements? If the mechanism doesn't involve water oxidation what happens to these holes then? It's unclear to me what else can be oxidised in the system when using pure water other than the COF.

6. On page 17 the authors write: "There are two pathways for WOR and ORR to generate H₂O₂ from water and air via 2e⁻ redox, corresponding to respective Eq.1-3 and Eq. 4-6. It can be checked whether one-step (Eq. 3 and 6) or two-step (Eq. 1-2 and Eq. 4-5) occurs through the *O₂⁻ and OH intermediates". What do this equation numbers refer to? To the equations on page S9 of the supporting information? If so, this needs to be clearer, and it might make sense to move/duplicate them in the main text. Also if it is the equations in the supporting information should it not be equations 1-4 and 5-6 in the first part of the sentence?

7. The authors might want to have a look at Fig. 4F. It's not clear to me what all the species are, e.g., the molecules at 9 o'clock which I assume should be water but don't look like water, and the meaning of the text below the arrows. The curved arrows flanking the straight arrows in the middle also look odd to me as *OH and *OOH are not entering the cycle at that point, although obviously in both cases H₂O₂ gets produced, also stoichiometry wise 2 *OHs are required to make H₂O₂. Finally, it might be better to describe this as two combined cycles rather than one cycle.

8. The authors should add all DFT optimised geometries for all species to the supporting information, preferably as a separate ZIP archive of one machine readable file (e.g. XTL or CIF) per structure.

Reviewer #4 (Remarks to the Author):

The manuscript by Chencheng Qin et al reports three COFs as photocatalysts for hydrogen peroxide formation. TaptBtt COF shows a high efficiency of photocatalytic H₂O₂ evolution because two reaction pathways of water oxidation and oxygen reduction can happen simultaneously, while the other two COF only work for ORR. Solid evidence for the reaction pathways has been shown in this paper, such as electron/hole scavenger control experiments, in-situ diffuse reflectance infrared Fourier transform spectroscopy measurement and DFT calculations for the absorption and free energy. However, a few questions must be resolved before the acceptance as a scientific manuscript to be published in Nature Communications.

1. The authors should provide simulated structural models for these COF and the refinement with experimental PXRD data.
2. Isomeric structures of materials can lead to different photocatalytic activities, such as J. Am. Chem. Soc. 2022, 144, 30, 13953–13960, I was just wondering if any isomers for the benzo[1,2-b:3,4-b':5,6-b''] trithiophene-2,5,8-tricarbaldehyde (Btt) monomer? It would be better if authors can character the Btt monomer using NMR to verify the structure of Btt.
3. In the methods section, the authors claim all COFs were synthesized in a 10 mL Pyrex tube, but the amount of solvent is 12 mL (6 mL o-DCB + 6 mL n-BuOH) which is more than the volume of the reactors. The authors should check all details in the paper carefully.
4. Most Schiff-base reactions need acetic acid or other acids to act as a catalyst for the protonation in carbinolamine, however, no acid was used in the COFs synthesis on page 24. So, I was just wondering if it's a mistake or if no acid was requested in synthesis.
5. Most photocatalytic experiments in this work were performed for only 90 mins which is a quite short time, even the long-term measurement is only seven hours. Authors should perform a long run to show the stability of photocatalysts, such as more than 40 hours of photocatalytic measurement for TaptBtt.
6. The stability measurement of TaptBtt for H₂O₂ generation has been presented in Fig. 3c, however, no detail was shown in the paper, such as hours for each cycle, the light source for this reaction, and amounts of catalyst and water.
7. Authors should use the same unit for all photocatalysis illumination, rather than different units as the absolute amount (such as Fig. 2h and 3b) and concentration (such as Fig. 2c, 2d, 2e, 3c, and 4a) of H₂O₂ products. Also, the amount of solution should be clarified in captions and contents if the authors prefer concentration as the unit for the H₂O₂ evolution.
8. Authors claim that 'the crystallinity of TaptBtt is positively correlated with the yield of H₂O₂ (Fig. 2f).' However, it looks opposite from authors' opinion in Fig. 2e and f, as the crystallinity follows the order of TaptBtt > TaptBtt-1 > TaptBtt-2 > TaptBtt-3 (Fig. 2e), but the activity of H₂O₂ production with the order of TaptBtt < TaptBtt-1 < TaptBtt-2 < TaptBtt-3 (Fig. 2f). Authors should be careful rephrase this opinion. Also, authors can discuss this opinion with some previous literature, such as Chem, 2019, 5(6), 1632-1647 and Nature 604, 72–79 (2022), which also show the correlations between the crystallinity and photocatalytic activity.

9. Authors claim that “the hollow “sea urchin’ shape around raised burrs” of TaptBtt can improve the exposure for O₂ that benefit the photo reactions. However, the surface area of TaptBtt is smaller than TapbBtt, so could authors brief comment on TEM and BET measurements here?

10. In-situ diffuse reflectance infrared Fourier transform spectroscopy was performed to analyse the reaction pathway for COFs. The peak at 1041 cm⁻¹ was attributed to the C-OH intermediate, which is the evidence for WOR for TaptBtt in Fig. 4e-f. However, peaks at a similar position also appeared for TpaBtt and TapbBtt in Fig. S25, although the intensity of peaks is weaker. Authors should also explain these peaks for TpaBtt and TapbBtt.

Some minor changes also need to be done as

1. The light sources for the experiments need to be described in the captions and contents even if it was shown in SI.
2. What’s the wavelength for TA decay profiles analysis in Fig. 3i?
3. The average electron transmission numbers participated in ORR of TaptBtt is 1.72 on page 16 and 1.71 in Fig. 4b, which should be the same in both places.
4. Authors should mention equations with the same name in the draft, rather than Eq. 1-3 on page 17 and Eq. S1-4 on page 19.
5. Authors only discussed the thiophene-based COFs for photocatalysis in the introduction, but some other COFs for photocatalytic H₂O₂ production also should be mentioned and discussed, such as *Angew. Chem. Int. Ed.* 2022, 61, e2022004; *J. Am. Chem. Soc.* 2022, 144, 22, 9902–9909; *CCS Chemistry*, <https://doi.org/10.31635/ccschem.022.202101578>; *Chemical Engineering Journal*, 2022, 449, 137802; *Chem. Mater.* 2022, 34, 11, 5232–5240; *Angew. Chem. Int. Ed.* 2022, 61, e2022023. Also, these references should be included in Supplementary table 3 for comparison.

Response to Reviewers' Comments (NCOMMS-22-36081-T)

Reviewer #1 (Remarks to the Author)

This work reports the synthesis of benzotrithiophene-based COFs with dual-donor-acceptor structures for photocatalytic generation of H₂O₂ from water and oxygen. They claimed that, by changing different monomers with various electron-acceptor capability, the photocatalytic performances can be enhanced, especially in the TaptBtt. Authors ascribed the enhanced performance to three reasons: satisfied energy band; facilitated in-plane charge transfer; favourable intermediate interaction. The present study contains several incorrect characterizations and lacks of critical evidences to support their claims. Therefore, the manuscript is not recommended for publication in *Nat. Commun.* Furthermore, there are some problems in this paper, several comments to be considered:

Our Specific Response: We appreciate reviewer's valuable time and efforts in reviewing our manuscript. In this work, we successfully synthesized three kinds of benzotrithiophene-based COFs, termed as TpaBtt, TapbBtt and TaptBtt. These COF catalysts with spatially independent redox centers could be highly efficient for photocatalytic production of hydrogen peroxide from water and oxygen (derived from natural air) without any sacrificial agents. Several highlights have been shown in a broad area as follows: i) The yields (2094 $\mu\text{M h}^{-1}$) of TaptBtt without any sacrificial agents at ambient condition exceeds that of all previously reported COFs via synchronous 2e⁻ water oxidation reaction and 2e⁻ oxygen reduction reaction. ii) A concept of "atomic spot-molecular area" via homodromous dual-donor-acceptor way with large energy difference in periodic framework was unveiled for enhanced photocatalytic performance. iii) It was demonstrated that the active sites of three COFs (2e⁻ ORR) were concentrated on the electron-acceptor fragments near the imine bond. It can optimally reduce the Gibbs free energy of O₂* and OOH* intermediates during the H₂O₂ generation. iv) The structure-performance correlation of COFs between the degree of crystallinity and the photocatalytic H₂O₂ activity was constructed and proven. We believe that this paper would be of particular interest and benefit to the readers of *Nat. Commun.*, as it provides the latest results and discussions across a broad area of research.

In order to consolidate the observed results and proposed concept, we have addressed the reviewers' questions accordingly. In particular, the reanalysis of femtosecond time-resolved transient absorption (fs-TA) spectroscopy solidly provides supports for the conclusion a periodic and unhindered way of charge transport due to the D-A direction of the imine linkage same to the direction of electron transfer between the molecular motifs. In addition, the protonation of linkage in three COFs further offers direct evidence that TaptBtt has a larger energy difference of line-region combination between linkages and linkers due to homodromous charge transfer.

- It is known that the HOMO and LUMO as a cluster are the counterparts of VB and CB levels. It can be seen from the band structure that the VB value moving range (0.7 eV) of the three COFs

is greater than that of CB (0.36 eV), consistent with the narrower band gaps of TpaBtt. According to molecular orbital theory, the up-shifted VB is originated from the effect of electron structure (*J. Mater. Chem. A* 2020, 8, 13299-13310). This indicates that the Tpa unit in three COF has the strongest supplying capacity and can generate more electrons upon excited state, which is in line with the corresponding DFT calculation results in **Fig. S1**. However, the direction of electron donation is opposite to the orientation of electron transfer of imine bond, where TpaBtt shows worse catalytic performance than that of TapbBtt and TaptBtt. For example, TpaBtt has a faster recombination rate of electron and hole leading to the highest fluorescence (PL results in **Fig. 3h**), and the lower carrier lifetime derived from fs-TA results in **Fig. 3i**. This is intrinsically ascribed to the short-range push-pull effect between Tpa and Btt (energy cancellation due to electron heterotransfer between intramolecular moieties and linked chemical molecules). TaptBtt could constitute the long-range push-pull effect between Btt-C=N-Tapt (energy superposition due to electron isotransfer between intramolecular motifs and interlocking chemical molecules). This effect promotes forward electron transfer and suppresses backward charge recombination.

Supplementary Figures 1. Calculated HOMO-LUMO distribution of TpaBtt, TapbBtt and TaptBtt.

Fig. 3 **h** PL spectra of TpaBtt, TapbBtt and TaptBtt. **i** Corresponding kinetics of characteristic fs-TA absorption bands observed at 540 nm for the spectra of TpaBtt, TapbBtt and TaptBtt, respectively.

● The fs-TA spectra of the three COFs form a wide negative feature at 575nm, assigning to ground state bleaching (GSB) and stimulated emission (ES), while the positive absorption band at 650nm is excited state absorption (ESA). The dynamics of excited state relaxation is mainly determined by the magnitude of intramolecular charge-transfer (ICT) in molecules (*J. Phys. Chem. A* 2007, 111, 5806–5812). As a consequence, the peak shift at progressively increasing time delays could explain the charge-transfer character of these push-pull units in COFs (*J. Phys. Chem. C* 2016, 120, 13922-13930). As shown in **Fig. S22**, compared to that of TpaBtt, the ESA and SE peaks for TaptBtt have obvious red shift amplitudes (black arrow). This is the result of the electron-deficient N-bridging of imines to the electron-acceptor unit, further demonstrating that COFs can achieve efficient photogenerated charge transfer using the push-pull mechanism from energy difference. Consequently, a global target analysis has been used in three COFs, where the initial Franck-Condon (FC) state splits into excited states and rapidly reaches bound excitonic state (BE) and charge separation state (CS) through internal transformation. The exciton under BE state is trapped, localizing on a single edge of the COF, and the electron and hole under CS state reside on separate motif edges either by intra- or interlayer charge transfer that increase their exciton radius, reduce their coulombic, and prolong their persistence in the excited (*J. Phys. Chem. Lett.* 2022, 13, 1398-1405). The fitting dynamics show the variation of two decay time constants in **Fig 4i**, where the short lifetime corresponds to the ascending component of TA and the intermediate recombination lifetime of the exciton trapped in the BE state, and the other component is the separation of the exciton into the SC state with a longer recombination lifetime. The value of τ_1 and τ_2 is 13.8 ps and 1925.2 ps for TpaBtt, and 15.7 ps and 2283.8 ps for TaptBtt. The τ_2 lifetime in TaptBtt is much longer-lived than that of TpaBtt, being accountable for greater charge separation capability derived from the homodromous charge transfer.

Supplementary Figure 22. Transient absorption spectra of TpaBtt (a), TapbBtt (b) and TaptBtt (c).

Fig. 4i Corresponding kinetics of characteristic fs-TA absorption bands observed at 540 nm for the spectra of TpaBtt, TapbBtt and TaptBtt, respectively.

- It has recently been reported that imine bonds in COFs reverse the direction of charge transfer after protonation in **Fig. S23**, achieving improved COFs photocatalytic performance (<https://doi.org/10.1002/anie.202216073>). Inspired by this conclusion, we think whether the charge reversal caused by protonation is directly related to the energy difference of line-region combination between linkages and linkers. Therefore, the homogeneous charge transfer in dual-donor-acceptor structure can be demonstrated by the imine protonation. All the three COFs were

protonated by ascorbic acid, and the FTIR spectra were used to prove the characteristic group. As shown in **Fig S24**, a new peak appeared at around 1800 cm^{-1} (broad), assigning to the $\text{C}=\text{NH}^+$ bond (*Nat. Commun.* 2022, 13, 6317). Moreover, the peak from the stretching mode of imine bonds attenuated. Subsequently, the photocatalytic H_2O_2 evolution performance of all COFs was examined in **Fig. 25**. It was found that a value of the H_2O_2 concentration was achieved by TpaBtt-AC ($175\text{ }\mu\text{M}$), nearly 3 time higher than that of pristine TpaBtt, followed by that of TapbBtt-AC ($216\text{ }\mu\text{M}$), nearly 1.8 times higher than that of pristine TapbBtt. However, H_2O_2 production in TaptBtt was essentially unchanged, and even slightly decreased. We further probed the difference in the separation and recombination of carriers for the three COFs by photocurrent (i-T) and electrochemical impedance (EIS). Obviously, TpaBtt-AC and TapbBtt-AC showed a higher current density than that of corresponding unprotonated COFs in **Fig. S26a-b**, while TaptBtt-AC exhibited a opposite trend in **Fig. S26c**. A similar phenomenon was also observed in the EIS test in **Fig. S26d-f**. Considering the above results, it was concluded that the performance of TpaBtt can be improved through protonation of imine bonds. It intrinsically ascribed that the protonation of imine bonds leads to the inversion of charge transfer orientation in an intramolecular way. In terms of homogenous charge transfer for TaptBtt, protonation of imine bond is difficult to overcome the larger energy difference of line-region combination between linkages and linkers for achieving the reversal of charge transfer orientation. This result directly proves that TaptBtt indeed has greater energy transfer between motifs than the other two COFs.

Supplementary Figure 23. Scheme of electron transfer path for both pristine COFs and protonated COFs treated by ascorbic acid (AC).

Supplementary Figure 24. Comparison of the FTIR spectra of COFs before and after protonation.

Supplementary Figure 25. Photocatalytic H_2O_2 performance under visible light irradiation for pristine COFs and protonated COFs. Conditions: water (50 mL), catalyst (15 mg), 300 W Xe lamp; $\lambda > 420 \text{ nm}$.

Supplementary Figure 26. (a-c) Transient photocurrent responses under visible light irradiation, and (d-e) Electrochemical impedance spectroscopy (EIS) of pristine COFs and protonated COFs.

1. Unlike the TpaBtt and TapbBtt, the optimized structure of TaptBtt in the DFT calculation is in a different form. Please explain it.

Response: Thanks for the valuable suggestion. TaptBtt is one of the COFs that has been reported, and its cell data (*cif file*) were obtained according to the initial article (*Adv. Funct. Mater.* 2022, 32, 2112553). The mirror image and symmetry of the COFs hexagon affect the display of the cell data. As seen from the supercell three models of COFs (**Fig. S37**), the difference in display was only observed. Therefore, the display difference of the optimized structure has no impact on the final calculation results.

Supplementary Figure 37. Optimal structural model of three COFs.

2. Solid-state ^{13}C NMR spectra should be provided. In addition, ^{33}S solid-state NMR is recommended.

Response: Thanks for the valuable suggestion. The solid-state ^{13}C NMR spectra of three COFs were provided in our revised manuscript (**Fig. S5**). In addition, we identified the structure of Btt monomer by the ^1H -NMR measurements (**Fig. S7**). The chemical structure of three COFs was verified by solid-state ^{13}C cross-polarization magic angle spinning (CP/MAS) nuclear magnetic resonance (NMR) spectroscopy at molecular level. As showed in **Fig. S5**, the lower-field signal at ~ 157 ppm can be assigned to the carbon of imine bond for TpaBtt and TapbBtt (*Nat. Commun.* 2022, 13, 2878). The characteristic carbon signal of imine group for TpatBtt was observed at about ~ 151 ppm, and the relatively high peak intensity is due to its coincidence with the position of benzene ring carbon bonded with nitrogen (*Appl. Catal B* 2022, 310, 121335).

Supplementary Figure 5. Solid state ^{13}C CP/MAS NMR spectra of TpaBtt, TapbBtt and TaptBtt.

Supplementary Figure 7. ^1H NMR spectrum of benzo[1,2-b:3,4-b':5,6-b'']trithiophene-2,5,8-tricarbaldehyde (Btt) monomer, 400 MHz, room temperature.

3. The author pointed out that “the crystallinity of TaptBtt is positively correlated with the yield of H_2O_2 (Fig. 2f)”. This seems to contradict the result of “the crystallinity follows the order of TaptBtt > TaptBtt-1 > TaptBtt-2 > TaptBtt-3 (Fig. 2e and Fig. S11).” In addition, why is the TaptBtt the highest crystallinity.

Response: Thanks for the question. By comparing this data with the original data, we are sorry to find that the error in labelling caused the problem. The revised **Fig. 2e** has been placed in the manuscript. The difference of crystallinity for TaptBtt, TaptBtt-1, TaptBtt-2, and TaptBtt-3 was achieved by adjusting the mixing ration of organic solvent during the synthesis process. The solvent chosen for the condensation reactions is crucial, as it governs the solubility of reactants (*Chem. Soc. Rev.* 2013, 42, 548-568). Also, it is key to obtaining the COF materials with structural regularity via effecting the reaction rate. In this work, the solubility of the two monomers (2,4,6-tris(4-aminophenyl)-1,3,5-triazine and benzo[1,2-b:3,4-b':5,6-b'']trithiophene-2,5,8-tricarbaldehyde) of the composition TaptBtt is similar to that of the mesitylene and dioxane, and the solubility of the monomers can be controlled effectively when mixed at a ratio of 1:1 (v/v).

However, a bias towards one of others will accelerate the solubility of the monomer in this organic solvent (mesitylene or dioxane). Therefore, a high concentration of monomer causes a rapid reaction, and results in a plethora of amorphous powders with uncontrolled shapes and a lack of molecular ordering. Thus, TaptBtt exhibited a highest crystallinity when a mixture mesitylene and dioxane at a ratio of 1/1(v/v) was employed.

Fig. 2e PXRD patterns of various degree of crystallinity in TaptBtt. **2f** Effects of TaptBtt crystallinity for H₂O₂ synthesis in water. Conditions: water (50 mL), catalyst (15 mg), 300 W Xe lamp; $\lambda > 420$ nm.

4. The possible active sites are only differentiated by DFT calculations, and no operando spectroscopic characterizations are conducted to support the computational results. DFT calculations can sometimes lead to groundless conclusions.

Response: Thanks for the question. For non-metallic materials, DFT calculation is a strong tool reported in most of literature to find possible active sites via the lower adsorption energy (*J. Am. Chem. Soc.* 2020, 142, 8104–8108; *Energy Environ. Sci.* 2022, 15, 830–842; *Adv. Mater.* 2020, 32, 1904433 and *Adv. Mater.* 2020, 32, 1904433). *In-situ* DRIFT spectroscopy is an operando spectroscopic characterization, which is suitable for COFs. *In-situ* DRIFT spectral measurement of three COFs for H₂O₂ photocatalysis under a continuous steam-saturated O₂ flow was carried out according to the reviewer's suggestion (**Fig. 4e and Fig. S35-S36**). The results of DFT calculation are basically consistent with those of *In-situ* DRIFT spectra.

- *In-situ* diffuse reflectance infrared Fourier transform (DRIFT) spectroscopy measurement was taken to monitor the interactions between the active sites and the intermediates. As shown in **Fig. 4e and Fig. S35**, the characteristic stretching of O-O at approximately 892 cm⁻¹ appears under photocatalytic reaction, verifying the occurrence of two-step single-electron route (*ACS Mater. Lett.* 2020, 2, 1008–1024; *Adv. Mater.* 2022, 34, 2110266). The O-O bond serves as the key intermediate for H₂O₂ production in 2e⁻ ORR. The intensity of the O-O peak follows the order of TaptBtt > TapbBtt > TpaBtt, matching well with both EPR and NBT results. It can be observed that

the intensity of the peaks at 1012 cm^{-1} and 1086 cm^{-1} , corresponding to the bending mode of C-H and C-C of center ring on three COFs, gradually increases. Besides, the C-S-C (951 cm^{-1}), belonging to the thiophene unit (*Adv. Mater.* 2020, 32, 2001259), is also gradually enhanced. The vibration of both C-C and C-S-C peak indicates the occurrence of photoinduced electrons transfer between the thiophene of Btt motif and the benzene ring fragments of Tapt motif. More importantly, there were vibrations for C=N (1623-1627 cm^{-1}) and C=NH⁺ (1567 cm^{-1}) for three COFs after photoexcitation, indicating that the subaqueous molecules adsorbed on the imine bond (**Fig. S36**). The simultaneous vibration of thiophene ring, benzene ring and O-O indicated that the active site occurs on the thiophene or benzene ring fragment. The OOH*, an important intermediate state for 2e⁻ ORR, requires electrons on the surface of the materials. The benzene ring near the imide is the electron-absorbing unit for TapbBtt and TaptBtt. Therefore, it can reasonably infer that the active site is located near the imine bond of the electron-acceptor fragment through *In-situ* DRIFT spectra, which is consistent with the DFT results.

Supplementary Figure 35. *In-situ* DRIFT spectra of TpaBtt (a) and TapbBtt (b).

Fig. 4e *In-situ* DRIFT spectra of TaptBtt

Supplementary Figure 36. *In-situ* DRIFT spectra of TaptBtt (a), TapbBtt (b) and TpatBtt (c).

5. The author claimed that “the VB values of TpaBtt and TapbBtt do not satisfy the oxidative potential of water oxidation ($2e^-$ WOR)”. However, the results in Figure S18 showed that TpaBtt and TapbBtt could generate H_2O_2 in Ar atmosphere. It is puzzling.

Response: Thanks for your kind suggestion. This question was also proposed by **Reviewer #3**. At the same time, **Reviewer #3** suggests that it is unscientific to judge whether water oxidation occurs only by the value of valence band. Both the pH of the actual aqueous solution and the high permittivity of water relative to the high permittivity of COF make the valence band value move more negative. Therefore, we retested the H_2O_2 production by TpaBtt and TapbBtt under the conditions of pure Ar and Ar with potassium bromate ($KBrO_3$). Furthermore, isotope labeling experiments of these two COFs were conducted. The experimental results are as follows:

- In the previous experiments, samples were collected every 15 minutes, which may cause oxygen from the air to enter the water. Therefore, in the re-done experiment, we only measured the concentration of H_2O_2 after 1h reaction. Before the reaction, the dissolved oxygen in water was eliminated via vacuum pump and Ar was filled to the water, and the reaction was kept under the atmosphere of Ar. As shown in **Fig. S27**, very weak H_2O_2 can still be detected under Ar. However, after adding the electron sacrificial agent ($KBrO_3$), the H_2O_2 concentration was

undetectable. This result implies that a four-electron water oxidation process may have occurred. However, the results of both RRDE (**Fig. S29a**) and H_2^{18}O isotope labelling experiments (**Fig. S30**) did not detect the production of O_2 . Therefore, we conducted a follow-up decomposition experiment on H_2O_2 generated by COFs using H_2^{18}O .

Isotope labelling experiments of these COFs were conducted. The catalysts (10 mg) and H_2^{18}O (97%, 1 mL) were put into hermetic device mainly composed of quartz tube and sealing components (the air was pumped away with a vacuum pump). The O_2 was bubbled into the suspension in the dark for 30 min. After 6 h irradiation, the formed H_2O_2 was decomposed by MnO_2 under Ar atmosphere. The O_2 generated by decomposition of photogenerated H_2O_2 was analyzed by GC-MS. Using this method, we can assess the $\text{H}_2^{18}\text{O}_2$ generated by two-electron water oxidation or four-electron water oxidation. As shown **Fig. S31**, the presence of $^{18}\text{O}_2$ was detected in all three COFs, indicating that water oxidation occurred in all three COFs, due to $^{18}\text{O}_2$ only deriving from H_2^{18}O . However, we can clearly see that there is a significant difference in the proportion of the two types of $^{16}\text{O}_2$ and $^{18}\text{O}_2$. The ratio of $^{18}\text{O}_2$ and $^{16}\text{O}_2$ is 1:4 (close to the four-electron water oxidation process) for TpaBtt and TapbBtt, while the ratio is 1:1.2 (close to the two-electron water oxidation process) for TaptBtt (*Adv. Mater.* 2020, 32, 1904433; *Adv. Mater.* 2022, 34, 2107480 and *Angew. Chem. Int. Ed.* 2022, 61, e202200413). In addition, only TaptBtt's H_2O_2 concentration showed a downward trend after adding the hole sacrifice (**Fig. 4a** and **Fig. S27**). It can be inferred that TaptBtt can directly use holes to produce hydrogen peroxide, which is consistent with the results of isotopes (the ratio of $^{18}\text{O}_2$ and $^{16}\text{O}_2$ closed to 1:1). For TpaBtt and TapbBtt, we reasonably conclude that they can undergo four-electron water oxidation to produce O_2 , and this part of O_2 is weak and may be adsorbed on the surface of COFs, then being directly used to produce H_2O_2 under the radiation. This also explains the absence of O_2 in the RRDE (**Fig. S29a**) and oxygen-producing isotopes (**Fig. S30**). Therefore, these results give solid supports that the H_2O_2 photosynthesis undergoes $2e^-$ ORR and $4e^-$ ORR for TpaBtt and TapbBtt, while TaptBtt has both $2e^-$ ORR and $2e^-$ WOR dual processes with higher atomic efficiency.

Supplementary Figure 27. Amount of H_2O_2 produced on TpaBtt (a) and Tapbtt (b) in CH_3OH (10% v/v, as the hole acceptor) with KBrO_3 (0.01 M, as the electron acceptor). Conditions: 50 mL H_2O ; 15 mg catalyst; 300 W Xe lamp; $\lambda > 420$ nm.

Supplementary Figure 30. H_2^{18}O isotope labelling experiments to test whether water is used to produce oxygen under light conditions: (a) TpaBtt, (b) TapbBtt and (c) TaptBtt. The relative intensity of $^{16}\text{O}_2$ and $^{18}\text{O}_2$ in the headspace of reactive vessels after photoirradiation was measured by GC-MS.

Supplementary Figure 31. Relative intensities of $^{16}\text{O}_2$ and $^{18}\text{O}_2$ in the gas products decomposed from H_2O_2 generated using different COFs (MnO_2 was used to decompose H_2O_2 .) The relative intensity of $^{16}\text{O}_2$ and $^{18}\text{O}_2$ in the headspace of reactive vessels after photoirradiation was measured by GC-MS.

6. The pore size distributions shown in Figure S9 indicate that there are two different pores in the COFs. Please explain it.

Response: Thanks for the valuable suggestion. Through literature research, we found that the imine COFs also exhibit different pore sizes in these papers (*J. Am. Chem. Soc.* 2022, 144, 19813–19824; *J. Am. Chem. Soc.* 2017, 139, 12911–12914; *Appl. Catal. B* 2022, 310, 121335). We deduce that this may be due to the following reasons. Firstly, the three COFs are microcrystalline materials seen from PXRD pattern, and not all the particles (or domains) are crystalline. There may be some amorphous polymers around the ordered hexagonal COFs. When N_2 adsorption-desorption experiments are conducted, this amorphous polymerization is compacted together (*Nat. Commun.* 2018, 9, 2998; *Angew. Chem. Int. Ed.* 2019, 58, 4906-4910). Secondly, COFs are inherently defective, which result in an unconcentrated distribution of pores.

Reviewer #2 (Remarks to the Author):

The authors designed and synthesized three imine-linked COFs using the Btt building blocks. They found TaptBtt displayed high selectivity for photo-generating H₂O₂ owing to the homodromous charge transfer and large energy difference of line-region combination between linkages and linkers. This work proposed an interesting concept for the analysis of good photocatalytic performance. However, the manuscript is very hard to read. The logic is messed up. The key of this work is the claim of the homodromous charge transfer and large energy difference of line-region combination between linkages and linkers in double donor-acceptor structures of periodic frameworks. The manuscript should answer the question of "how do the push-pull effects between intramolecular motif and linkage chemistry in polymer affords highly catalytic efficiency?" The D-A direction of the imine linkage and the direction of electron transfer between the molecular motifs should play a key role in photocatalytic performance according to the proposed rule by the authors. However, I could not see direct and clarified evidence for this. The authors do try their best to prove the reason that the TaptBtt has the best performance. Better wettability and faster transfer of electrons in the interface et al., for example. The direct evidence is missing. The Femtosecond time-resolved transient absorption (fs-TA) spectroscopy may give some support to this concept. However, no detailed analysis was given. I only found that the description of "The lower negative band of TapbBtt was observed, indicating that the exciton relaxation dynamics was different from other two COFs." I could not see a very clear explanation that TaptBtt has the best performance due to the homodromous charge transfer like the author claimed as the highlight of this work.

I am afraid that the realization can not reach Nature Communications' quality, and the claims are not supported. Thus, I cannot recommend this work for publication.

Our Specific Response: We appreciate reviewer's valuable time and efforts in reviewing our manuscript, and acknowledge that the concepts we presented were interesting. In this work, we successfully synthesized three kinds of benzotrithiophene-based COFs, termed as TpaBtt, TapbBtt and TaptBtt. These COF catalysts with spatially independent redox centers could be highly efficient for photocatalytic production of hydrogen peroxide from water and oxygen (derived from natural air) without any sacrificial agents. Several highlights have been shown in a broad area as follows: i) The yields (2094 $\mu\text{M h}^{-1}$) of TaptBtt without any sacrificial agents at ambient condition exceeds that of all previously reported COFs via synchronous $2e^-$ water oxidation reaction and $2e^-$ oxygen reduction reaction. ii) A concept of "atomic spot-molecular area" via homodromous dual-donor-acceptor way with large energy difference in periodic framework was unveiled for enhanced photocatalytic performance. iii) It was demonstrated that the active sites of three COFs ($2e^-$ ORR) were concentrated on the electron-acceptor fragments near the imine bond. It can optimally reduce the Gibbs free energy of O_2^* and OOH^* intermediates during the H_2O_2 generation. iv) The structure-performance correlation of COFs between the degree of

crystallinity and the photocatalytic H₂O₂ activity was constructed and proven. Therefore, we believe that this paper would be of particular interest and benefit to the readers of *Nat. Commun.*

In order to consolidate the observed results and proposed concept, we have addressed the reviewers' questions accordingly. In particular, "how do the push-pull effects between intramolecular motif and linkage chemistry in polymer affords highly catalytic efficiency" has been explained in detail. The reanalysis of femtosecond time-resolved transient absorption (fs-TA) spectroscopy solidly provides supports for the conclusion that a periodic and unhindered way of charge transport due to the D-A direction of the imine linkage same to the direction of electron transfer between the molecular motifs. In addition, the protonation of linkage in three COFs further offer direct evidence that TaptBtt has a larger energy difference of line-region combination between linkages and linkers due to homodromous charge transfer.

- It is known that the HOMO and LUMO as a cluster are the counterparts of VB and CB levels. It can be seen from the band structure that the VB value moving range (0.7 eV) of the three COFs is greater than that of CB (0.36 eV), consistent with the narrower band gaps of TpaBtt. According to molecular orbital theory, the up-shifted VB originated from the effect of electron structure (*J. Mater. Chem. A* 2020, 8, 13299-13310). This indicates that the Tpa unit in three COF has the strongest supplying capacity and can generate more electrons upon excited state, which is in line with the corresponding DFT calculation results in **Fig. S1**. However, the direction of electron donation is opposite to the orientation of electron transfer of imine bond, and TpaBtt shows worse catalytic performance than that of TapbBtt and TaptBtt. For example, TpaBtt has a faster recombination rate of electron and hole leading to the highest fluorescence (PL results in **Fig. 3h**), and the lower carrier lifetime derived from fs-TA results in **Fig. 3i**. This is intrinsically ascribed to the short-range push-pull effect between Tpa and Btt (energy cancellation due to electron heterotransfer between intramolecular moieties and linked chemical molecules). TaptBtt could constitute the long-range push-pull effect between Btt-C=N-Tapt (energy superposition due to electron isotransfer between intramolecular motifs and interlocking chemical molecules). This effect promotes forward electron transfer and suppresses backward charge recombination.

Supplementary Figures 1. Calculated HOMO-LUMO distribution of TpaBtt, TapbBtt and TaptBtt.

Fig. 3 h PL spectra of TpaBtt, TapbBtt and TaptBtt. **i** Corresponding kinetics of characteristic fs-TA absorption bands observed at 540 nm for the spectra of TpaBtt, TapbBtt and TaptBtt, respectively.

● The fs-TA spectra of the three COFs form a wide negative feature at 575nm, assigning to ground state bleaching (GSB) and stimulated emission (ES), while the positive absorption band at 650nm is excited state absorption (ESA). The dynamics of excited state relaxation is mainly determined by the magnitude of intramolecular charge-transfer (ICT) in molecules (*J. Phys. Chem. A* 2007, 111, 5806–5812). As a consequence, the peak shift at progressively increasing time delays could explain the charge-transfer character of these push-pull units in COFs (*J. Phys. Chem. C* 2016, 120, 13922-13930). As shown in **Fig. S22**, compared to that of TpaBtt, the ESA and SE peaks for TaptBtt have obvious red shift amplitude (black arrow). This is the result of the electron-deficient N-bridging of imines to the electron-acceptor unit, further demonstrating that COFs can achieve

efficient photogenerated charge transfer using the push-pull mechanism from energy difference. Consequently, a global target analysis is used in three COFs, where the initial Franck-Condon (FC) state splits into excited states and rapidly reaches bound excitonic state (BE) and charge separation state (CS) through internal transformation. The exciton under BE state is trapped, localizing on a single edge of the COF, and the electron and hole under CS state reside on separate motif edges either by intra- or interlayer charge transfer that increase their exciton radius, reduce their coulombic, and prolong their persistence in the excited (*J. Phys. Chem. Lett.* 2022, 13, 1398-1405). The fitting dynamics show the variation of two decay time constants in Fig. 4i, where the short lifetime corresponds to the ascending component of TA and the intermediate recombination lifetime of the exciton trapped in the BE state, and the other component is the separation of the exciton into the SC state with a longer recombination lifetime. The value of τ_1 and τ_2 is 13.8 ps and 1925.2 ps for TpaBtt, and 15.7 ps and 2283.8 ps for TaptBtt. The τ_2 lifetime in TaptBtt is much longer-lived than that of TpaBtt, being accountable for greater charge separation capability derived from the homodromous charge transfer.

Supplementary Figure 22. Transient absorption spectra of TpaBtt (a), TapbBtt (b) and TaptBtt (c).

Fig. 4i Corresponding kinetics of characteristic fs-TA absorption bands observed at 540 nm for the spectra of TpaBtt, TapbBtt and TaptBtt, respectively.

- It has recently been reported that imine bonds in COFs reverse the direction of charge transfer after protonation in **Fig. S23**, achieving improved COFs photocatalytic performance (<https://doi.org/10.1002/anie.202216073>). Inspired by this conclusion, we think whether the charge reversal caused by protonation is directly related to the energy difference of line-region combination between linkages and linkers. Therefore, the homogeneous charge transfer in dual-donor-acceptor structure can be demonstrated by the imine protonation. All the three COFs were protonated by ascorbic acid, and the FTIR spectra were used to prove the characteristic group. As shown in **Fig. S24**, a new peak appeared at around 1800 cm^{-1} (broad), assigning to the $\text{C}=\text{NH}^+$ bond (*Nat. Commun.* 2022, 13, 6317). Moreover, the peak from the stretching mode of imine bonds attenuated. Subsequently, the photocatalytic H_2O_2 evolution performance of all COFs was examined in **Fig. 25**. It was found that a value of the H_2O_2 concentration was achieved by TpaBtt-AC ($175\ \mu\text{M}$), nearly 3 times higher than that of pristine TpaBtt, followed by that of TapbBtt-AC ($216\ \mu\text{M}$), nearly 1.8 times higher than that of pristine TapbBtt. However, H_2O_2 production in TaptBtt was essentially unchanged, and even slightly decreased. We further probed the difference in the separation and recombination of carriers for the three COFs by photocurrent (i-T) and electrochemical impedance (EIS). Obviously, TpaBtt-AC and TapbBtt-AC showed a higher current density than that of corresponding unprotonated COFs in **Fig. S26a-b**, while TaptBtt-AC exhibited an opposite trend in **Fig. S26c**. A similar phenomenon was also observed in the EIS test in **Fig. S26d-f**. Considering the above results, it was concluded that the performance of TpaBtt can be improved through protonation of imine bonds. It is intrinsically ascribed that the protonation of imine bonds leads to the inversion of charge transfer orientation in an intramolecular way. In terms of homogeneous charge transfer TaptBtt, protonation of imine bond is difficult to overcome the larger energy difference of line-region combination between linkages and linkers for achieving

the reversal of charge transfer orientation. This result directly proves that TaptBtt does have greater energy transfer between motif than the other two COFs.

Supplementary Figure 23. Scheme of electron transfer path for both pristine COFs and protonated COFs treated by ascorbic acid (AC).

Supplementary Figure 24. Comparison of the FTIR spectra of COFs before and after protonation.

Supplementary Figure 25. Photocatalytic H_2O_2 performance under visible light irradiation for pristine COFs and protonated COFs. Conditions: water (50 mL), catalyst (15 mg), 300 W Xe lamp; $\lambda > 420$ nm.

Supplementary Figure 26. (a-c) Transient photocurrent responses under visible light irradiation, and (d-e) Electrochemical impedance spectroscopy (EIS) of pristine COFs and protonated COFs.

Besides, the following problems should be paid attention to.

1. The authors claimed intermolecular donor-acceptor existed in these COFs. If this description is correct after condensation of the two building blocks? "Intramolecular" may be more suitable. Please carefully check that.

Response: Thanks for the valuable suggestion. COF materials are porous polymer materials formed by organic structural units through covalent bonds. Therefore, it is very appropriate to

change “intermolecular” to “intramolecular” as proposed by the reviewer. The revised manuscript has been replaced with an “intramolecular” description.

- It also provides a paradigm to solve the common problem that how the push-pull effects between intramolecular motif and linkage chemistry in polymer affords highly catalytic efficiency.
- It indicates the intramolecular donor-acceptor (D-A) structure between the two functional motifs in COFs are profitably constructed.
- Hence, the imine linkage of Btt-based COFs is not only endowed with photo-reactivity, but also synchronously regulate both the charge transfer directionality and the energy difference of intramolecular donor-acceptor in these COFs, further affecting the utilization of charge carrier.
- The photocatalytic yield of $\cdot\text{O}_2^-$ by TaptBtt was 6.02×10^{-5} M, markedly higher than that of TapbBtt (3.44×10^{-5} M) (**Fig. 4c**), suggesting that the larger energy difference of intermolecular donor-acceptor in TapbBtt promoted the generation of $\cdot\text{O}_2^-$ intermediate.

2. In Figure 1b, Tpa motif was defined as an electron donor, while the authors described it as an electron acceptor in Page 7 and D short for donor was used as the abbreviation. Thus, Tpa motif as an electron donor was connected to the "anionic" nitrogen atom. This will lead to an opposite conclusion to the manuscript. Please carefully address this problem.

Response: Thanks for the advice. Through careful examination, we found that it was our writing mistakes to cause this problem. The density functional theory (DFT) demonstrated that HOMO orbitals are mainly distributed on the donor-unit Tpa motif, while LUMO orbitals are mainly distributed on the acceptor unit BTT motif. This is consistent with previously reported results (*J. Catal.* 2021, 402, 52–60). The description has been corrected in the revised manuscript.

- As shown in **Fig. 1b**, the electron- donor Tpa motif (D) is connected to the “anionic” nitrogen atom, and the electron- acceptor Btt motif (A) is coupled with the “cationic” carbon atom in the imine bond of TpaBtt.

3. In page 7, the authors claimed the required excitation energy for TaptBtt transition is smaller than that of other two COFs. However, the band gap energy (Eg) of TpaBtt, TapbBtt, and TaptBtt was counted as 1.95, 2.32 and 2.29 eV, severally. TpaBtt exhibited a narrower bandgap.

Response: Thank you very much for pointing out this problem. We found that the relationship between the gap of HOMO-LUMO and optical band gap is closed related, while these are still different through literature search. First of all, the gap of LUMO-HOMO describes the transition from an occupied orbital to a non-occupied orbital, and the orbitals are ordered. For excitation energy, the transition from one electronic state (ground state) to another (excited state) is dominated by a large number of orbital transitions. The difference between these two is the binding energy of the associated exciton. Secondly, the lowest excited state is theoretically dominated by the HOMO-LUMO orbital, but there are still other orbital transitions possible, and

this part of the transition can also affect the optical band gap of the semiconductor. Therefore, the orbital energy difference is not completely equal to the excitation energy and the optical band gap difference. Taking the above two points into consideration, we revised the ambiguous statement.

- Based on the transition probability (**Table S1**) analysis of molecular orbitals (MO), most of electrons were contributed by the LUMO (MO 147, 163 and 163 for respective TpaBtt, TapbBtt and TaptBtt) in transition process, indicating that the electronic configuration of LUMO nearly represents photogenerated electron composition. Moreover, the transition energy of some molecular orbitals in TaptBtt is lower than that of the other two COFs.

4. As displayed in Figure 2e, the crystallinity follows the order (Fig. 2e and Fig. S11) of TaptBtt > TaptBtt-1 > TaptBtt-2 > TaptBtt-3. The authors claimed that the crystallinity of TaptBtt is positively correlated with the yield of H₂O₂. According to Figure 2f, TaptBtt exhibited the lowest yield of H₂O₂.

Response: Thanks for the reviewer's advice. By comparing the data with the original data, we are sorry to find that the error in labelling caused the problem. The revised **Fig. 2e** has been placed in the manuscript, and the relationship between crystallinity and photocatalytic activity has been discussed accordingly (*Chem 2019, 5, 1632-1647; Nature 2022, 604, 72–79*).

- To demonstrate the influence of ordered degree in pure COFs for the production of H₂O₂, TaptBtt with different crystallinity was prepared via adjusting the ratios of mesitylene and 1,4-dioxane in mixture, corresponding to 3:3, 2:4, 4:2, and 6:0 for respective TaptBtt, TaptBtt-1, TaptBtt-2, TaptBtt-3. The crystallinity follows the order (**Fig. 2e**) of TaptBtt > TaptBtt-1 > TaptBtt-2 > TaptBtt-3. The crystallinity of TaptBtt is positively correlated with the yield of H₂O₂ (**Fig. 2f**), revealing that the long-ranged ordered structure of TaptBtt may assist the H₂O₂ photosynthesis. A higher crystalline TapbBtt, as a result of its π columns, allows for quick exciton migration and hole transport along the π -conjugated direction, and greatly retards backward reverse recombination of charge. In contrast, the lower crystallinity in TaptBtt-1,2,3 cannot effectively prevent backward charge recombination, resulting in dissipation of the photoexcited states (*Chem 2019, 5, 1632-1647; Nature 2022, 604, 72–79*).

Fig. 2e PXR D patterns of various degree of crystallinity in TaptBtt. **2f** Effects of TaptBtt crystallinity for H₂O₂ synthesis in water. Conditions: water (50 ml), catalyst (15 mg), 300 W Xe lamp; $\lambda > 420$ nm.

5. Page 6, "lowest occupied molecular orbitals (LUMO)" is wrong.

Response: Thanks for the reviewer's advice. We are sorry that this error occurred in our submitted manuscript, and "lowest occupied molecular orbitals (LUMO)" has been replaced by "lowest unoccupied molecular orbitals (LUMO)" in the revised manuscript.

Reviewer #3 (Remarks to the Author):

The manuscript by Qin and co-workers is interesting but there are some issues that need addressing before publication:

Response: The reviewer's valuable time and effort on reviewing our manuscript are greatly appreciated. We have made revisions according to the reviewer's suggestions.

1. Did the authors attempt to solve the structures of their COFs from the obtained PXRD patterns?

Response: According to the reviewer's suggestion, simulated COF structural models and experimental PXRD refinement data are provided in the supporting information. As shown in **Fig. S2**, the calculated the PXRD profiles for AA stacking mode were found to match well with experimental profiles of TpaBtt, TapbBtt and TaptBtt, respectively. The optimized PXRD displayed TpaBtt, TapbBtt and TaptBtt the (100) reflection at $2\theta = 5.43^\circ$, 4.84° and 4.76° . Pawley refinements of the AA stacking model based on the experimental profiles gave unit cells with parameters ($a=b=57.62 \text{ \AA}$, $c=7.06$, $\alpha=\beta=90^\circ$, $\gamma=120^\circ$) for TpaBtt (residuals $R_p=3.33\%$, $R_{wp}=4.71\%$), ($a=b=63.97 \text{ \AA}$, $c=7.00$, $\alpha=\beta=90^\circ$, $\gamma=120^\circ$) for TapbBtt (residuals $R_p=3.71\%$, $R_{wp}=5.18\%$), and ($a=b=65.39 \text{ \AA}$, $c=6.98$, $\alpha=\beta=90^\circ$, $\gamma=120^\circ$) for TaptBtt (residuals $R_p=4.10\%$, $R_{wp}=7.67\%$).

Supplementary Figure 2. Experimental, refined and simulated PXRD patterns of TpaBtt (a), TapbBtt (b) and TaptBtt (c). Insert represented the simulated eclipsed (AA) and staggered (AB) stacking models of the corresponding imine 2D COF.

2. When analysing the ability of TaptBtt to drive oxygen reduction and the 2-electron oxidation of water on page 13 the values used for the potentials of both these solution half-reactions appear to be the standard values tabulated for pH 0. However, the H₂O₂ synthesis experiments are not performed at pH 0 but pH ~ 5 (see page 20). In their analysis the authors should use the values shifted to pH 5 using the Nernst equation, which will be shifted to more negative values, rather than the pH 0 ones.

Response: Thanks for the reviewer's advice. As reported (*Nat. Commun.* 2017, 8, 701; *ACS. Catal.* 2014, 4, 3749), the standard potential values for two-electron oxidation of water and two-electron reduction of O₂ are as follow:

According to Nernst's equation ($E = E^\circ - 0.059 \cdot \text{pH}$), the potential value of $E_{(\text{H}_2\text{O}_2/\text{H}_2\text{O})}$ and $E_{(\text{O}_2/\text{H}_2\text{O}_2)}$ are respectively 1.47 eV and 0.41 eV when pH=5. We have marked the corresponding pH values in the description of the standard values of these half reactions. Besides, we redescribed the relationship between band structure of COFs and thermodynamics in our revised manuscript.

● As shown in **Fig. 3f**, the band structure of TaptBtt was thermodynamically sufficient for the synchronous synthesis of H₂O₂ from H₂O oxidation ($E_{\text{H}_2\text{O}_2/\text{H}_2\text{O}} = +1.76 \text{ eV}$ and $E_{\text{O}_2/\text{H}_2\text{O}} = 1.23 \text{ eV}$ vs pH=0) and O₂ reduction ($E_{\text{O}_2/\text{H}_2\text{O}_2} = +0.70 \text{ eV}$ vs pH=0). However, in the actual reaction the pH of our solution is about 5. Thus, according to the Nernst equation, the above values are offset to a certain extent ($E_{\text{H}_2\text{O}_2/\text{H}_2\text{O}} = +1.47 \text{ eV}$; $E_{\text{O}_2/\text{H}_2\text{O}} = 0.94 \text{ eV}$; $E_{\text{O}_2/\text{H}_2\text{O}_2} = +0.41 \text{ eV}$ vs pH=5).

3. Ref. 25 is cited at the end of the sentence with the values of the oxygen reduction and the 2-electron oxidation of water potentials on page 13 but I think this is a mistake. Ref. 25: "Patterson AL. The Scherrer formula for X-Ray particle size determination. *Phys. Rev.* 56, 978-982 (1939)" appears to be about something completely different.

Response: We are sorry that this error occurred in our submitted manuscript, and we have replaced it with the corresponding reference (*Nat. Commun.* 2017, 8, 701; *ACS Catal.* 2014, 4, 3749) in the revised manuscript.

4. The authors should be careful to use their valence band edge measured for a COF in vacuum to analyse the ability of the material to drive half-reactions when immersed in water. The high dielectric constant of water relative to that of the COF will shift the valence band edge in water to more negative values (and the conduction band edge to more positive values). Taking this into account probably won't qualitatively change the picture if one also takes the pH shifts discussed

above into account, because these make the water oxidation reaction easier, compensating for the fact that the band edges will shift to more positive values in water.

Response: Thanks for the reviewer's advice. We have also noticed the question whether it is appropriate to use the valence band edge of COFs measured in vacuum to analyze the ability of the materials to drive half-reactions when immersed in water. According to the reviewer's opinion and analysis, we deleted this inappropriate sentence in the revised manuscript. Therefore, three kinds of COF oxygen production experiments and H₂O₂ decomposition isotope experiments were performed. The test results and analysis are as follows:

The detailed processes of isotope labelling experiments were conducted according to literature method (*Adv. Mater.* 2022, 34, 2107480). Catalysts (10 mg) and H₂¹⁸O (97%, 1 mL) were put into hermetic device mainly composed of quartz tube and sealing components (the air was pumped away with a vacuum pump). After O₂ was bubbled into the suspension in the dark for 30 min, the suspension was stirred in dark for 30 min to reach the absorption-desorption equilibrium. Prior to the photocatalytic experiment, the O₂ atmosphere was detected by GC-MS as a control. After 6 h irradiation, the gas products in the headspace of the reaction vessel were analyzed by Gas Chromatography-Mass Spectrometer (GC-MS-QP2020 NX, SHIMADZU). Meanwhile, the formed H₂O₂ was decomposed by MnO₂ under Ar atmosphere. The O₂ generated by decomposition of photogenerated H₂O₂ was analyzed by GC-MS. Using this method, we can assess the H₂¹⁸O₂ generated by two/four-electron water oxidation.

As shown in the **Fig. S30** and **Fig. S31**, none of the three COFs was detected ¹⁸O₂ production in the first stage, while all did in the second stage. This demonstrated that three COFs can use water to produce H₂O₂. However, we can clearly see that the ratio of the two oxygen (¹⁸O₂ and ¹⁶O₂) is significantly different after H₂O₂ decomposition in second step. The ratio of ¹⁸O₂ and ¹⁶O₂ is 1:4.8 (close to the four-electron water oxidation process, **Eq.1**) for TpaBtt and TapbBtt, while the ratio is 1:1.2 (close to the two-electron water oxidation process, **Eq. 2**) for TaptBtt (*Adv. Mater.* 2020, 32, 1904433; *Adv. Mater.* 2022, 34, 2107480 and *Angew. Chem. Int. Ed.* 2022, 61, e202200413). In addition, there is a difference trend of H₂O₂ concentration after adding sacrificial agent for TpaBtt, TapbBtt and TaptBtt, and H₂O₂ can still be detected under the condition of Ar for three COFs (**Fig. 4a** and **Fig. S27**). It can be inferred that TaptBtt can directly use holes to produce H₂O₂, while TpaBtt and TapbBtt can indirectly use holes to produce O₂ and then evolve into H₂O₂ (*Energy Environ. Sci.* 2022, 15, 830-842; *Adv. Mater.* 2020, 32, 1904433; *Angew. Chem. Int. Ed.* 2022, 61, e202200413). It is consistent with the results of isotope studies (the ratio of ¹⁸O₂ and ¹⁶O₂ closed to 1:1 for TaptBtt). For TpaBtt and TapbBtt, we reasonably conclude that they can undergo four-electron water oxidation to produce O₂, and this part of O₂ is weak and may be adsorbed on the surface of COFs, then being directly used to produce H₂O₂ under the radiation. This also explains the absence of O₂ in the RRDE (**Fig. S29a**) and oxygen-producing isotopes (**Fig. S30**). Therefore, these results give solid supports that the H₂O₂ photosynthesis undergoes 2e⁻

ORR and $4e^-$ ORR for TpaBtt and TapbBtt, while TaptBtt has both $2e^-$ ORR and $2e^-$ WOR dual processes with higher atomic efficiency.

Supplementary Figure 30. H_2^{18}O isotope labelling experiments to test whether water is used to produce oxygen under light conditions: (a) TpaBtt, (b) TapbBtt and (c) TaptBtt. The relative intensity of $^{16}\text{O}_2$ and $^{18}\text{O}_2$ in the headspace of reactive vessels after photoirradiation was measured by GC-MS.

Supplementary Figure 31. Relative intensities of $^{16}\text{O}_2$ and $^{18}\text{O}_2$ in the gas products decomposed from H_2O_2 generated using different COFs (MnO_2 was used to decompose H_2O_2 .) The relative intensity of $^{16}\text{O}_2$ and $^{18}\text{O}_2$ in the headspace of reactive vessels after photoirradiation was measured by GC-MS.

5. Following on from the above, TpaBtt and TapbBtt cannot oxidise water but still produce H_2O_2 . Do they produce H_2O_2 via a different mechanism not involving water oxidation, as suggested on page 16 and page 18, or would this suggest an issue with the XPS measurements? If the mechanism doesn't involve water oxidation what happens to these holes then? It's unclear to me what else can be oxidised in the system when using pure water other than the COF.

Response: Thanks for the reviewer's advice. This question was also proposed by **Reviewer #1**. Therefore, we retested the experiment of hydrogen peroxide production by TpaBtt and TapbBtt

under the conditions of Ar and Ar with potassium bromate (KBrO_3). Furthermore, we have also retested the XPS measurements, and the results are the same as before.

In the previous experiment, samples were collected every 15 minutes, which may cause oxygen from the air to enter the water. Therefore, in the re-done experiment, we only measured the concentration of H_2O_2 after 1h reaction. Before the reaction, the dissolved oxygen in water was eliminated via vacuum pump and Ar was filled to the water, and the reaction was kept under the atmosphere of Ar. As shown in **Fig. S27**, very weak H_2O_2 can still be detected under Ar. However, after adding the electron sacrificial agent (KBrO_3), the H_2O_2 concentration was undetectable. This result implies that a four-electron water oxidation process may have occurred. However, the results of both RRDE (**Fig. S29a**) and H_2^{18}O isotope labelling experiments (**Fig. S30**) did not detect the production of O_2 . Therefore, we conducted a follow-up decomposition experiment on H_2O_2 generated by COFs using H_2^{18}O . This part of the data analysis has been answered in the fourth question above.

Supplementary Figure 27. Amount of H_2O_2 produced on TpaBtt (a) and TapbBtt (b) in CH_3OH (10% v/v, as the hole acceptor) with KBrO_3 (0.01 M, as the electron acceptor). Conditions: 50 mL H_2O ; 15 mg catalyst; 300 W Xe lamp; $\lambda > 420$ nm.

6. On page 17 the authors write: “There are two pathways for WOR and ORR to generate H_2O_2 from water and air via $2e^-$ redox, corresponding to respective Eq.1-3 and Eq. 4-6. It can be checked whether one-step (Eq. 3 and 6) or two-step (Eq. 1-2 and Eq. 4-5) occurs through the $^*\text{O}_2^-$ and OH intermediates”. What do these equation numbers refer to? To the equations on page S9 of the supporting information? If so, this needs to be clearer, and it might make sense to move/duplicate them in the main text. Also if it is the equations in the supporting information should it not be equations 1-4 and 5-6 in the first part of the sentence?

Response: Thanks for the reviewer’s suggestions. We have added related equations 1-6 to the revised manuscript, and the equations in the supporting information refer to the calculation process of DFT intermediate states. To better understand the formation pathway of H_2O_2 , it

makes a distinction between the reaction pathway and the calculation process of DFT intermediate states.

Eq. 1-6

Eq. S1-S6

7. The authors might want to have a look at Fig. 4F. It's not clear to me what all the species are, e.g., the molecules at 9 o'clock which I assume should be water but don't look like water, and the meaning of the text below the arrows. The curved arrows flanking the straight arrows in the middle also look odd to me as *OH and *OOH are not entering the cycle at that point, although obviously in both cases H₂O₂ gets produced, also stoichiometry wise 2 *OHs are required to make H₂O₂. Finally, it might be better to describe this as two combined cycles rather than one cycle.

Response: Thanks for the reviewer's advice. In the revised manuscript, we have redrawn Fig. 4f according to the corresponding comments, and clearly marked the various substances. The mechanism of H₂O₂ produced by TaptBtt was described by two combined cycles: 1) the left side is the two-electron water oxidation pathway, the intermediate state *OH formed on the site of triazine unit via water molecules and photogenic holes, and then the two intermediate *OH states generate H₂O₂; 2) the right side is two-electron oxygen reduction, the O₂ adsorbed by the benzene ring to form an intermediate state *OOH with electrons and protons, and then to produce H₂O₂ with an electron and proton.

Fig. 4f Mechanism of TaptBtt for photocatalytic H₂O₂ formation. The white, grey, blue, yellow and red sphere referred to hydrogen, carbon, nitrogen, sulfur and oxygen, severally.

8. The authors should add all DFT optimised geometries for all species to the supporting information, preferably as a separate ZIP archive of one machine readable file (e.g. XTL or CIF) per structure.

Response: Thanks for the reviewer's advice. All DFT optimized geometries for all species, as a separate ZIP archive of one machine readable file (e.g. XTL or CIF) per structure, have been added to the supporting information in our revised manuscript.

Reviewer #4 (Remarks to the Author):

The manuscript by Chencheng Qin et al reports three COFs as photocatalysts for hydrogen peroxide formation. TaptBtt COF shows a high efficiency of photocatalytic H₂O₂ evolution because two reaction pathways of water oxidation and oxygen reduction can happen simultaneously, while the other two COF only work for ORR. Solid evidence for the reaction pathways has been shown in this paper, such as electron/hole scavenger control experiments, in-situ diffuse reflectance infrared Fourier transform spectroscopy measurement and DFT calculations for the absorption and free energy. However, a few questions must be resolved before the acceptance as a scientific manuscript to be published in Nature Communications.

Response: We thank the reviewer for the helpful comments to our work, and have made revisions according to the reviewer's suggestions.

1. The authors should provide simulated structural models for these COF and the refinement with experimental PXRD data.

Response: According to the reviewer's suggestion, we used Material Studio software to simulate the crystal structure, and the Pawley refinement demonstrated the good fit of the eclipse stacking model (AA stacking) for three COFs. The optimized PXRD displayed TpaBtt, TapbBtt and TaptBtt the (100) reflection at $2\theta = 5.43^\circ$, 4.84° and 4.76° . Pawley refines of the AA stacking model based on the experimental profiles gave unit cell with parameters ($a=b=57.62 \text{ \AA}$, $c=7.06$, $\alpha=\beta=90^\circ$, $\gamma=120^\circ$) for TpaBtt (residuals $R_p=3.33\%$, $R_{wp}=4.71\%$), ($a=b=63.97 \text{ \AA}$, $c=7.00$, $\alpha=\beta=90^\circ$, $\gamma=120^\circ$) for TapbBtt (residuals $R_p=3.71\%$, $R_{wp}=5.18\%$), and ($a=b=65.39 \text{ \AA}$, $c=6.98$, $\alpha=\beta=90^\circ$, $\gamma=120^\circ$) for TaptBtt (residuals $R_p=4.10\%$, $R_{wp}=7.67\%$).

Supplementary Figure 2. Experimental, refined and simulated PXRD patterns of TpaBtt (a), TapbBtt (b) and TaptBtt (c). Insert represented the simulated eclipsed (AA) and staggered (AB) stacking models of the corresponding imine 2D COF.

2. Isomeric structures of materials can lead to different photocatalytic activities, such as *J. Am. Chem. Soc.* 2022, 144, 30, 13953–13960, I was just wondering if any isomers for the benzo[1,2-b:3,4-b':5,6-b'']trithiophene-2,5,8-tricarbaldehyde (Btt) monomer? It would be better if authors can character the Btt monomer using NMR to verify the structure of Btt.

Response: Thanks for the valuable suggestion. There are isomers for the benzo[1,2-b:3,4-b':5,6-b'']trithiophene-2,5,8-tricarbaldehyde (Btt) monomer (**Fig. S6**, *J. Mater. Chem. A* 2017, 5, 8317-8324). According to the reviewer's advice, we identified the structure of Btt monomer by the $^1\text{H-NMR}$ measurement in **Fig. S7**.

Supplementary Figure 6. Molecular structure of benzotrithiophene isomeric core C-1 and C-2.

Supplementary Figure 7. $^1\text{H-NMR}$ spectrum of benzo[1,2-b:3,4-b':5,6-b'']trithiophene-2,5,8-tricarbaldehyde (Btt) monomer, 400 MHz, room temperature.

3. In the methods section, the authors claim all COFs were synthesized in a 10 mL Pyrex tube, but the amount of solvent is 12 mL (6 mL *o*-DCB + 6 mL *n*-BuOH) which is more than the volume of the reactors. The authors should check all details in the paper carefully.

Response: Thanks for the reviewer's advice, and we are sorry that this error occurred in our submitted manuscript. We have corrected it in the revised manuscript, and checked all details in the paper carefully.

- Tpa (58.07 mg, 0.2 mmol) and Btt (66.1 mg, 0.2 mmol) were put into 10 mL Pyrex tube, and dissolved into *o*-dichlorobenzene (*o*-DCB, 3 mL) and *n*-butanol (3 mL) mixed solution (v/v = 1:1).
- Tapb (70.2 mg, 0.2 mmol) and Btt (66.1mg, 0.2 mmol) were put into 10 mL Pyrex tube, and dissolved into *o*-DCB (3 mL) and *n*-butanol (3 mL) mixed solution (v/v = 1:1).
- Tapt (70.8 mg, 0.2 mmol) and Btt (66.1mg, 0.2 mmol) were put into 10 mL Pyrex tube, and dissolved into mesitylene (3 mL) and 1,4-dioxane (3 mL) mixed solution (v/v = 1:1).

4. Most Schiff-base reactions need acetic acid or other acids to act as a catalyst for the protonation in carbinolamine, however, no acid was used in the COFs synthesis on page 24. So, I was just wondering if it's a mistake or if no acid was requested in synthesis.

Response: Thanks for the reviewer's advice. Schiff-base reactions need acetic acid to act as a catalyst for the protonation in carbinolamine in our system. In supporting information on page 2, the synthesis of TaptBtt with different crystallinity requires acetic acid: "TaptBtt-1: After the above mixture was sonicated for 10 min, acetic acid aqueous solution (0.5 mL, 6 M) was added, which was then sonicated again for 2 min." However, we regret that the concentration and content of acetic acid required for the synthesis of the three COFs were missing from the main text of the manuscript. We have added them in the revised manuscript.

- After the above mixture was sonicated for 10 min, 0.3ml of 6M acetic acid aqueous solution was added, and then sonicated 2 min over again.
- After the above mixture was sonicated for 10 min, 0.3 ml of 9M acetic acid aqueous solution was added, and then sonicated 2 min over again.
- After the above mixture was sonicated for 10 min, 0.5ml of 6M acetic acid aqueous solution was added, and then sonicated 2 min over again.

5. Most photocatalytic experiments in this work were performed for only 90 mins which is a quite short time, even the long-term measurement is only seven hours. Authors should perform a long run to show the stability of photocatalysts, such as more than 40 hours of photocatalytic measurement for TaptBtt.

Response: Thanks for the valuable suggestion. To be practically useful, the long-term photostability of catalysts is essential. We therefore tested the photostability of TaptBtt using a continuous approach (96h) in pure water. As shown in **Fig. S16**, the photocatalytic H₂O₂ production rate of TaptBtt reaches to 580 μmol (5800 μM), which is higher than 330 μmol for SonoCOF-F2 under same conditions (*J. Am. Chem. Soc.* 2022, 144, 9902-9909). In addition, although the formation rate of H₂O₂ began to slow down after 48h, the total amount continued to rise.

Supplementary Figure 16. Long-term photocatalytic H₂O₂ production of TaptBtt: 100 ml of pure water and 80 mg of TaptBtt; 300W Xe lamp; $\lambda > 420$ nm.

6. The stability measurement of TaptBtt for H₂O₂ generation has been presented in Fig. 3c, however, no detail was shown in the paper, such as hours for each cycle, the light source for this reaction, and amounts of catalyst and water.

Response: Thanks for the reviewer's suggestion. For the stability measurement of TaptBtt for H₂O₂ generation in pure water, the conditions for each cycle were that: 50 mL of pure water and 15 mg of TaptBtt for 90 min; 300 W Xe lamp; $\lambda > 420$ nm. Meanwhile, the corresponding reaction conditions are added in the captions in Fig. 3c of the revised manuscript.

7. Authors should use the same unit for all photocatalysis illumination, rather than different units as the absolute amount (such as Fig. 2h and 3b) and concentration (such as Fig. 2c, 2d, 2e, 3c, and 4a) of H₂O₂ products. Also, the amount of solution should be clarified in captions and contents if the authors prefer concentration as the unit for the H₂O₂ evolution.

Response: Thanks for the valuable suggestion. According to the reviewer's suggestion, all photocatalysis illumination use the concentration as unit and the amount of solution in figure captions of the revised manuscript (Fig. 2c, 2d, 2f, 3b, 3c, and 4a). For Fig. 2h, considering that the amount of catalyst, solution volume and illumination time are different in previous literature, we still decided to use $\mu\text{mol h}^{-1}\text{g}^{-1}$ as the comparison unit, according to the reference (*J. Am. Chem. Soc.* 2020, 142, 20107–20116).

8. Authors claim that 'the crystallinity of TaptBtt is positively correlated with the yield of H₂O₂ (Fig. 2f).' However, it looks opposite from authors' opinion in Fig. 2e and f, as the crystallinity follows the order of TaptBtt > TaptBtt-1 > TaptBtt-2 > TaptBtt-3 (Fig. 2e), but the activity of H₂O₂ production with the order of TaptBtt < TaptBtt-1 < TaptBtt-2 < TaptBtt-3 (Fig. 2f). Authors should be careful rephrase this opinion. Also, authors can discuss this opinion with some previous

literature, such as *Chem*, 2019, 5(6), 1632-1647 and *Nature* 604, 72–79 (2022), which also show the correlations between the crystallinity and photocatalytic activity.

Response: Thanks for the reviewer’s advice. By comparing the data with the original data, we are sorry to find that the error in labelling caused the problem. The revised **Fig. 2e** has been placed in the manuscript, and the relationship between crystallinity and photocatalytic activity has been discussed accordingly (*Chem* 2019, 5, 1632-1647; *Nature* 2022, 604, 72–79)

● To demonstrate the influence of ordered degree in pure COFs for the production of H₂O₂, TaptBtt with different crystallinity was prepared via adjusting the ratios of mesitylene and 1,4-dioxane in mixture, corresponding to 3:3, 2:4, 4:2, and 6:0 for respective TaptBtt, TaptBtt-1, TaptBtt-2, TaptBtt-3. The crystallinity follows the order (**Fig. 2e**) of TaptBtt > TaptBtt-1 > TaptBtt-2 > TaptBtt-3. The crystallinity of TaptBtt is positively correlated with the yield of H₂O₂ (**Fig. 2f**), revealing that the long-ranged ordered structure of TaptBtt may assist the H₂O₂ photosynthesis. A higher crystalline TaptBtt, as a result of its π columns, allow for quick exciton migration and hole transport along the π -conjugated direction, greatly retards backward reverse recombination of charge. In contrast, the lower crystallinity in TaptBtt-1,2,3 cannot effectively prevent backward charge recombination, might resulting in dissipation of the photoexcited states (*Chem* 2019, 5, 1632-1647; *Nature* 2022, 604, 72–79).

Fig. 2e PXR D patterns of various degree of crystallinity in TaptBtt. **2f** Effects of TaptBtt crystallinity for H₂O₂ synthesis in water. Conditions: water (50 ml), catalyst (15 mg), 300 W Xe lamp; $\lambda > 420$ nm.

9. Authors claim that “the hollow “sea urchin’ shape around raised burrs” of TaptBtt can improve the exposure for O₂ that benefit the photo reactions. However, the surface area of TaptBtt is smaller than TapbBtt, so could authors brief comment on TEM and BET measurements here?

Response: Thanks for the valuable suggestion. We have briefly commented on TEM and BET measurements in the revised manuscript. The SEM and TEM images show that TaptBtt presents larger raised burrs and thinner layers. However, TapbBtt is more likely to form dendritic

aggregates, with tiny folds clearly visible on the surface of the branches. This tiny fold may be responsible for TapbBtt with large surface area ($1492.4 \text{ m}^2 \text{ g}^{-1}$), compared with TpaBtt and TaptBtt (850.9 and $994.9 \text{ m}^2 \text{ g}^{-1}$) in **Table S2**. In addition, the data of dominant pore size distribution exhibit that TaptBtt has a larger pore diameter (1.51 nm) than that of TapbBtt (1.45 nm) and TpaBtt (1.20 nm) in **Fig. S13**. Therefore, we conclude that the larger raised burr and larger pore size improve the exposure for O_2 , benefiting for the photo reactions.

10. In-situ diffuse reflectance infrared Fourier transform spectroscopy was performed to analyse the reaction pathway for COFs. The peak at 1041 cm^{-1} was attributed to the C-OH intermediate, which is the evidence for WOR for TaptBtt in Fig. 4e-f. However, peaks at a similar position also appeared for TpaBtt and TapbBtt in, although the intensity of peaks is weaker. Authors should also explain these peaks for TpaBtt and TapbBtt.

Response: Thanks for the reviewer's advice. This weak peak may be due to the adsorption of water molecules on the surface of TpaBtt and TapbBtt, and the dissociation of water molecules can lead to the formation of C-OH. In addition, the generation of hydrogen peroxide could react with the electron and can also produce C-OH.

Some minor changes also need to be done as

1. The light sources for the experiments need to be described in the captions and contents even if it was shown in SI.

Response: Thanks for the reviewer's suggestions. The light sources for the experiments were described in the figure captions (**Fig. 2c, 2d, 2f, 2h, 2g, 3c, and 4a**) in the revised manuscript, and the light sources of the experiments were also added in the figure captions of the Supporting information (**Fig. S14, S16, S25, S27, and S34**).

2. What's the wavelength for TA decay profiles analysis in Fig. 3i?

Response: Thanks for the valuable suggestion. The wavelength for TA decay profiles analysis is 540 nm for three COFs. The corresponding date was added in caption of **Fig. 3i**.

3. The average electron transmission numbers participated in ORR of TaptBtt is 1.72 on page 16 and 1.71 in Fig. 4b, which should be the same in both places.

Response: Thanks for the reviewer's advice, and we are sorry that this error occurred in our submitted manuscript. In **Fig. 4b** and on page 1, the average electron transmission numbers participated in ORR of TaptBtt is 1.71 in the revised manuscript.

● As seen from **Fig. 4b** and **Fig. S28**, the average electron transmission numbers participated in ORR were 1.62, 1.57 and 1.71 for TpaBtt, TapbBtt and TaptBtt, respectively.

4. Authors should mention equations with the same name in the draft, rather than Eq. 1-3 on page 17 and Eq. S1-4 on page 19.

Response: Thanks for the reviewer's advice. The **Eq. 1-3** on page 17 mainly describes the path of H₂O₂ produced by COFs, focusing on the free radical process in solution. The **Eq. S1-4** on page 19 is using DFT to calculate the active site and Gibbs free energy of H₂O₂ produced by COFs, taking into account the oxygen, free radicals and H₂O₂ in the adsorbed state. There is a slight difference between the two sets. According to **Reviewer #2**, **Eq. 1-6** were exhibited in the revised manuscript, and **Eq. S1-56** were displaced in the supporting information. The **Eq. 1-6** and **Eq. S1-6** were respectively described below:

Eq. 1-6

Eq. S1-6

5. Authors only discussed the thiophene-based COFs for photocatalysis in the introduction, but some other COFs for photocatalytic H₂O₂ production also should be mentioned and discussed, such as *Angew. Chem. Int. Ed.* 2022, 61, e2022004; *J. Am. Chem. Soc.* 2022, 144, 22, 9902–9909; *CCS Chemistry*, <https://doi.org/10.31635/ccschem.022.202101578>; *Chemical Engineering Journal*, 2022, 449, 137802; *Chem. Mater.* 2022, 34, 11, 5232–5240; *Angew. Chem. Int. Ed.* 2022, 61, e2022023. Also, these references should be included in Supplementary table 3 for comparison.

Response: Thanks for the valuable suggestion. The corresponding references have been cited to the revised manuscript. **Supplementary table 3** only compares the H₂O₂ production under the condition without sacrificial agent, while the references (*Chem. Mater.* 2022, 34, 5232–5240 and *Angew. Chem. Int. Ed.* 2022, 61, e2022023) have no data under the same condition. Therefore, they are not included for comparison.

● These references, including *CCS Chemistry* (DOI: 10.31635/ccschem.022.202101578), *Chem. Eng. J.*, 2022, 449, 137802; *Chem. Mater.*, 2022, 34, 11, 5232–5240 and *Angew. Chem. Int. Ed.*, 2022, 61, e2022023, were cited in the introduction: Covalent organic frameworks (COFs), a

type of crystalline and non-metallic polymers, become feasible and promising platforms in field of artificial photosynthesis due to porous structure, photoelectric properties, and photochemical stability¹⁰⁻¹². The structural tunability and regularity of COFs, easily realized by practical building blocks with abundant topologies and dimensionalities, endow broad optical absorption, favorable mass transfer and fast charge carrier mobility¹³⁻¹⁵. However, using COFs as photocatalysts for H₂O₂ production is still rare. Voort et al. reported two COFs based on a (diarylamino) benzene linker for generating H₂O₂, and subsequently vinyl, fluorinated and Ti-based COFs were also used to catalyze H₂O₂ production¹⁶⁻²⁰. But most of these reports require sacrificial agents or oxygen bubbling during the reaction.

- The reference (*J. Am. Chem. Soc.* 2022, 144, 9902-9909) has been cited on page 11: As shown in **Fig. S16**, the photocatalytic H₂O₂ production rate of TaptBtt reach to 580 μmol (5800 μM), which is higher than 330 μmol for SonoCOF-F2 under same conditions⁴⁶.
- The reference (*Angew. Chem. Int. Ed.* 2022, 61, e202200413) has been cited on page 12: The presence of sacrificial reagents or buffers normally limits the direct utilization of H₂O₂ for environmental implications⁴⁷.

REVIEWER COMMENTS

Reviewer #1 (Remarks to the Author):

This work designed benzotrithiophene-based covalent organic frameworks with homodromous dual donor-acceptor structure for photocatalytic production of hydrogen peroxide from water and oxygen, which can effectively improve charge separation efficiency, leading to enhanced hydrogen peroxide production rate of up to 2094 $\mu\text{M h}^{-1}$ with apparent quantum efficiency of 4.5% at 450 nm. While the work is of some interest, the innovation of this work is insufficient. Therefore, the manuscript is not recommended for publication in Nature Communication.

Reviewer #2 (Remarks to the Author):

The authors made a lot of effort to do the revision according to my suggestions. I am happy to see that fs-TA spectra measurement is done. The quality of the overall manuscript has been improved. The analysis is still not very satisfactory.

The very highlight of this work is the concept of "atomic spot-molecular area" via homodromous dual-donor-acceptor, as the authors claimed several times in the manuscript. The author should try to verify the concept with direct evidence. The catalytic performance can not be direct evidence to verify this concept. Too many factors, such as the BET surface area, crystallinity, the dispersion of the COFs materials, the conditions of the photocatalysis, et al., can influence the results of the catalytic activity. Even the authors also found that the crystallinity of the TaptBtt could influence the result directly (Figure 2e,f). DFT is better to be used as companion evidence.

In the analysis of the fs-TA spectra, the authors claimed that the redshift amplitudes are the result of the electron-deficient N bridging of imines to the electron-acceptor unit, further demonstrating that COFs can achieve efficient photogenerated charge transfer using the push-pull mechanism from energy difference. Why? Please give an explanation in detail. And for TaptBtt, the redshift is not obvious. How to explain that by using the same logic?

And I could not find the fitting dynamics which show the variation of two decay time constants in Fig 4i. I think it is supposed to be Fig. 3i. That result is positive. However, in my opinion, this evidence is still not sufficient to prove the concept of "atomic spot-molecular area" Very interestingly, the authors gave very good references, which are the protonated COFs. I would highly suggest comparing the fs-TA spectra of the pristine and protonated COFs. The difference may give direct and clear evidence. In addition, the study of the molecular segments of the three COFs as small molecules can also be helpful and also suggested. The author should try to prove the push-pull interaction occurred in the "atomic spot-molecular area," not the integral results.

Besides, the following problems should be paid attention to.

1. For the PXRD of the three COFs, it was always observed a sharp decrease before 5 degree. Please give an explanation and provide the raw data.
2. I am surprised that the author put Figure S7 in SI. It is unacceptable for a chemist. No signals are from the compound itself. It looks like the signals of the deuterated DMSO.
3. Please give the .cif files of the final structures of the three COFs with AA and AB stacking modes. The distance between the layers looks unreasonably large. Please explain this. I would suggest to double check the models. The models are very different from the reported Adv. Funct. Mater. 2022, 32, 2112553.
4. Please cite Adv. Funct. Mater. 2022, 32, 2112553.

Reviewer #3 (Remarks to the Author):

- 1) Firstly, and I should have picked this up in my original review, but I am not sure about the word "Homodromous" in the article title. It's a word I have never come across and so might not be obvious word to any chemist. Worse, I looked it up in some dictionaries and while the Oxford English Dictionary gives "Running in the same direction: opposed to heterodromous" besides a botany based definition "Turning in the same direction, as two generating spirals of a phyllotaxis (e.g. on the main stem and on a branch)", Merriam Webster only gives the botany based definition "having the genetic spiral following the same direction in both stem and branches", while the word doesn't appear in other online dictionaries. Hence, I think it's best to avoid the word homodromous in the title and the text, replace it with a better word or perhaps just leave it out all together as I don't think any extra adjective is required.
- 2) The authors have added some text about previous work on page 4 which reads "vinyl, fluorinated and Ti-based COFs". While the vinyl and fluorinated materials are probably COFs, Ti-based COFs sounds more like a MOF than a COF.
- 3) When discussing the HOMO and LUMO of the COFs on page 6/7 the authors should clarify to the reader that (i) these are predictions rather than measurements and (ii) that the calculations were done on isolated fragments of the COFs than rather than the fully periodic COF.
- 4) The authors have added a discussion about transition probability analysis (page 7, supplementary information table 1) but it's unclear to me what this analysis involves as little detail of how this analysis is performed is given. The information in table 1 as it stands makes little sense. In theory one could calculate excited states of the fragments with TD-DFT and then analyse the character of these excited states in terms of a projection of the excited state on excitations between orbitals but TD-DFT (or other method that could have been used CI/S, CC2 etc.) is not mentioned. Alternatively, one could crudely approximate excited states just in terms of orbital energy differences but in that case there would be no percentage contributions as every excitation is approximated to be identical to an excitation of a single

electron between one occupied and unoccupied orbital.

5) It would make sense to show in Fig. 3F the 2e⁻ water oxidation and reduction potentials at pH 5 rather than pH 0.

6) The isotope measurements are a great addition, but the discussion needs to be clearer. A first stage and second stage are mentioned but not explained. For me it would make more sense to discuss the isotope experiments in terms of in the dark, in the light, and/before and after addition of MnO₂.

7) On page 21 the authors write “For TpaBtt and TapbBtt, it was reasonably concluded that they can undergo four-electron water oxidation to produce O₂, and this part of O₂ is weak and may be adsorbed on the surface of COFs, then being directly used to produce H₂O₂ under the radiation. However, this four-electron process has a little contribution to H₂O₂ production for TpaBtt and TapbBtt. This also explains the absence of O₂ in the RRDE (Supplementary Fig. 29a) and oxygen-producing isotopes (Supplementary Fig. 30).”. While I agree with the analysis in terms of TaptBtt performing 2e⁻ water oxidation and the other materials the thermodynamic easier 4e⁻ water oxidation, I have absolutely no idea what the authors mean in the paragraph above with “this part of O₂ is weak” or “this four-electron process has a little contribution to H₂O₂ production”. What does it mean that O₂ is weak and while the 4 e⁻ oxidation of water obviously produces no H₂O₂ as many holes are consumed by 4e⁻ water oxidation as electrons consumed in water reduction to H₂O₂ and if the water oxidation is slow for TpaBtt and TapbBtt the H₂O₂ production must also be slow.

Reviewer #4 (Remarks to the Author):

I am satisfied with the revisions made by the authors in response to my comments and am happy to approve the manuscript for publication in its current form.

Reviewer #1 (Remarks to the Author):

This work designed benzotrithiophene-based covalent organic frameworks with homodromous dual donor-acceptor structure for photocatalytic production of hydrogen peroxide from water and oxygen, which can effectively improve charge separation efficiency, leading to enhanced hydrogen peroxide production rate of up to 2094 $\mu\text{M h}^{-1}$ with apparent quantum efficiency of 4.5% at 450 nm. While the work is of some interest, the innovation of this work is insufficient. Therefore, the manuscript is not recommended for publication in Nature Communication.

Our Specific Response: We appreciate your comments about innovative aspect of our work. Please understand that we have carefully addressed the comments about the additional characterization and experimental evidences you provided during the first round of review. We have additionally simulated the photocatalytic reaction process by operando spectroscopic characterizations according to the suggestions by reviewer #2, and supported our conclusions by reversing the charge transfer direction through protonated COFs. According to the insightful comments and advice from the reviewers, further elaborations on the innovation of this work have been provided.

Several enlightening highlights have been shown in a broad area as follows: i) a concept of “atomic spot-molecular area” in dual-donor-acceptor way proposed and verified in periodic frameworks for explaining the behavior of favorable charge transfer during the photocatalytic process; ii) thiophene-based COFs, such as trithiophene-based COFs, are rarely used to the artificial H_2O_2 photosynthesis; iii) a high H_2O_2 production yield of 2094 $\mu\text{M h}^{-1}$ for TaptBtt without any sacrificial agents at ambient condition exceeds most of previously reported COFs via synchronous 2e^- water oxidation reaction and 2e^- oxygen reduction reaction; iv) a systemic study was uniquely carried out for practical hydrogen peroxide production, involving the integration with the regulation of charge carrier separation, the identification of real active sites, the verification of key intermediates, and macroscopically structure-performance correlations. The detailed elaborations are as follows:

1. Novelty of theoretical concept for explaining the photoinduced charge transfer

Previous studies have mainly focused on the donor-acceptor (D-A) properties of the two building blocks in COFs (*J. Am. Chem. Soc.* 2023, 145, 8364–8374; *Angew. Chem. Int. Ed.* 2021, 60, 19797–19803; *ACS Catal.* 2020, 10, 15, 8717–8726; *J. Am. Chem. Soc.* 2015, 137, 7817–7827 and *J. Am. Chem. Soc.* 2014, 136, 9806–9809), while the intrinsic D-A characteristics of linkage (e.g. imine bond) at the atomic scale have been less studied. The effects of dual D-A integration between building blocks and linkages in periodic frameworks for charge transfer and subsequent photocatalysis remain an uncover field.

In this work, considering that the linkage chemistry of COFs serves as the connection and transport bridge, the D-A structure within linkage from the perspective of atomic scale should be explored, i.e., carbon of imine bonds as donor unit (D) and nitrogen as acceptor unit (A). The

electron-donor Tpa motif is connected to the “anionic” nitrogen atom, and the electron-acceptor Btt motif is coupled with the “cationic” carbon atom in the imine linkage of TpaBtt. This makes the D-A direction of the imine linkage opposite to the direction of electron transfer between the molecular motifs. In contrast, both TapbBtt and TaptBtt are in the same direction (D-A-type imine COFs), forming periodic and unhindered ways of charge transfer. The required energy for the D-A-type imine COFs to twist the same angle in excited state is reduced, resulting in an increase in the energy difference of the groups connected near the imine bond. Hence, the imine linkage of Btt-based COFs is endowed with photo-reactivity, and can synchronously regulate both the directionality of charge transfer and the energy difference of intramolecular donor-acceptor in these COFs, further affecting the utilization of charge carrier in surface/interface. This combination of line-region between linkages and linkers (termed as a uniport concept of “atom spot-molecular area”) via dual-donor-acceptor way in periodic framework is proposed to vary the catalytic reaction pathways and photosynthetic performance.

In order to consolidate the observed results and proposed concept, we explained in detail “how the push-pull effects between intramolecular motif and linkage chemistry in polymer affords high catalytic efficiency” in terms of pure COFs, molecular fragment, and protonated COFs. A series of characterization techniques, including the Femtosecond time-resolved transient absorption (fs-TA) spectroscopy, fluorescence spectroscopy, photoelectrochemical tests, in-situ X-band electron paramagnetic resonance (EPR), etc., directly verified the unhindered ways of charge transfer in the periodic units, since the D-A direction of the imine linkage is same to the direction of electron transfer between the D-A building organic motifs. In addition, the positive variation on the protonation of linkage in three COFs to the photoelectronic properties and the performance of hydrogen peroxide formation further offers evidence. The following is a detailed description of the above phenomenon.

1.1 Study of pure COFs

It is known that the HOMO and LUMO are a cluster, which are the counterparts of VB and CB levels. It can be seen from the band structure that the VB moving range (0.7 eV) of the three COFs is greater than that of CB (0.36 eV), consistent with the narrower band gaps of TpaBtt. According to molecular orbital theory, the up-shifted VB is originated from the effect of electron structure (*J. Mater. Chem. A.*, 2020, 8, 13299-13310). This indicates that the Tpa unit in three COF has the strongest supplying capacity and can generate more electrons upon excited state, which is line with the corresponding DFT calculation results in **Fig. S1**. However, the direction of electron donation is opposite to the orientation of electron transfer of imine bond, hindering the separation of charge carrier. Thus, the TpaBtt shows worse photoelectronic performance than that of TapbBtt and TaptBtt. For example, TpaBtt has a faster recombination rate of electron and hole leading to the highest fluorescence (PL results in **Fig. 3h**), and the lower carrier lifetime derived from fs-TA results in **Fig. 3i**. This is intrinsically ascribed to the short-range push-pull effect between Tpa and Btt (energy cancellation due to the opposite direction of electron transfer

between intramolecular moieties and imine linkage). TaptBtt could constitute the long-range push-pull effect between Btt-C=N-Tapt (energy superposition due to electron can be smoothly transfer between intramolecular motifs and interlocking chemical molecules). This effect promotes forward electron transfer and subsequently suppresses backward charge recombination.

Supplementary Figure 1. Calculated HOMO-LUMO distribution of TpaBtt, TapbBtt and TaptBtt.

Fig. 3. **h** PL spectra of TpaBtt, TapbBtt and TaptBtt. **i** Corresponding kinetics of characteristic fs-TA absorption bands observed at 540 nm for the spectra of TpaBtt, TapbBtt and TaptBtt, respectively.

Furthermore, the fs-TA spectra of the three COFs form a wide negative feature at 575 nm, assigning to ground state bleaching (GSB) and stimulated emission (ES), while the positive absorption band at 650 nm is excited state absorption (ESA). The dynamics of excited state relaxation is mainly determined by the magnitude of intramolecular charge-transfer (ICT) in molecules (*J. Phys. Chem. A* 2007, 111, 5806–5812). As a consequence, the peak shift at progressively increasing time delays could explain the charge-transfer character of these push-pull units in COFs (*J. Phys. Chem. C* 2016, 120, 13922-13930). As shown in **Fig. S23**, compared with TpaBtt, the ESA and SE peaks for TaptBtt have obvious red shift amplitude (black arrow). This is the result of the electron-deficient N-bridging of imines to the electron-acceptor unit, further demonstrating that COFs can achieve efficient photogenerated charge transfer using the push-pull mechanism from energy difference. Consequently, a global target analysis is used in three COFs, and the initial Franck-Condon (FC) state splits into excited states and rapidly reaches bound excitonic state (BE) and charge separation state (CS) through internal transformation. The exciton under BE state is trapped, localizing on a single edge of the COF, and the electron and hole under CS state reside on separate motif edges either by intra- or interlayer charge transfer that increase their exciton radius, reduce their coulombic, and prolong their persistence in the excited (*J. Phys. Chem. Lett.* 2022, 13, 1398-1405). The fitting dynamics show the variation of two decay time constants in **Fig 4i**, where the short lifetime corresponds to the ascending component of TA and the intermediate recombination lifetime of the exciton trapped in the BE state, and the other component is the separation of the exciton into the SC state with a longer recombination lifetime. The values of τ_1 and τ_2 are 13.8 ps and 1925.2 ps for TpaBtt, and 15.7 ps and 2283.8 ps for TaptBtt. The τ_2 lifetime in TaptBtt is much longer-lived than that of TpaBtt, being accountable for greater capability of charge carrier separation derived from the same direction of charge transfer.

Supplementary Figure 23. Transient absorption spectra of TpaBtt (a), TapbBtt (b) and TaptBtt (c).

Fig. 4i Corresponding kinetics of characteristic fs-TA absorption bands observed at 540 nm for the spectra of TpaBtt, TapbBtt and TaptBtt, respectively.

1.2 Study of molecular fragments in COFs

The molecular fragments (the same as the component of unit cell) of three COFs were also studied to prove the universality of our proposed concept in aperiodic segments. The molecular fragments of three kinds of COFs were successfully synthesized using benzo[1,2-b:3,4-b':5,6-b'']-trithiophene-2-tricarbaldehyde and different monomers, including 4,4,4-triaminotriphenylamine, 1,3,5-tris(4-aminophenyl) benzene, and 2,4,6-tris(4-aminophenyl)-1,3,5-triazine under the reflux of trichloromethane and ethanol solution for 24 h. The obtained model compounds were respectively named as TapBtt-Fragment, TapbBtt-Fragment and TaptBtt-Fragment.

We firstly probed the difference in the separation and recombination of charge carriers of these model compounds by photocurrent measurement and electrochemical impedance (EIS) spectra. According to **Fig. S2**, TaptBtt-fragment had the higher photocurrent density with a lesser charge transference resistance, indicating that the TaptBtt-fragment showed the upper charge separation capability and the more available surface photogenerated carriers for solid-liquid interfacial reaction. Subsequently, as demonstrated by time-resolved fluorescence spectroscopy (**Fig. S3**) under the excited wavelength of 350 nm, the fluorescence lifetime of TaptBtt-fragment (1.40 ns) was obviously higher than that of TapbBtt-fragment (1.19 ns) and TpaBtt-fragment (0.74 ns). TaptBtt-fragment was able to suppress the e^- - h^+ recombination more effective than that of TapbBtt-fragment and TpaBtt-fragment (*Angew. Chem. Int. Ed.* 2023, 62, e202216073). To get insight into the charge transfer pathway, the X-band electron paramagnetic resonance (EPR) experiments were also conducted. In **Fig. S4**, the relative intensity (0.31) of EPR signal for the TpaBtt-fragments in dark and light conditions is significantly higher than that of TapbBtt-fragments (0.29) and TaptBtt-fragments (0.23). The less relative strength for TaptBtt-fragments is probably related to the ability of ground state charge transfer from the acceptor unit to the donor unit (*Nat. Commun.* 2019, 10, 5538; *Nat. Commun.* 2018, 9, 1182; *Adv. Sustainable syst.* 2022, 6, 2100264),

which is dominated by the integrated interaction of the directivity of double D-A structure and the energy difference in these molecular fragments. Integrating photoelectrochemical measurement and PL lifetime with EPR results, TaptBtt-fragment owns more efficient charge transfer due to the favorable push-pull effects between intramolecular motif and imine bond. Therefore, it can be concluded that the push-pull interaction of “atomic spot-molecular area” also occurs in aperiodic segments.

Supplementary Figure 2. (a) Transient photocurrent response under visible light irradiation, and (b) Electrochemical impedance spectroscopy (EIS) of fragments.

Supplementary Figure 3. Fluorescence lifetime decay of COF fragments.

Supplementary Figure 4. EPR spectra of all COF fragments under dark and light.

1.3 Comparison study of protonated and unprotonated COFs

It has recently been reported (*Angew. Chem. Int. Ed.* 2023, 62, e2022160) that the imine linkage in COFs could reverse the direction of charge transfer after protonation (**Fig. S24**), achieving improved COFs photocatalytic performance. Inspired by this conclusion, we think that the charge reversal caused by protonation is directly related to the energy difference of line-region combination between linkages and linkers. Therefore, the direction of charge transfer in dual-donor-acceptor structure can be demonstrated by the imine protonation. All of three COFs were protonated by ascorbic acid, and the FITR spectra were used to prove the characteristic group. As shown in **Fig. S25**, a new peak appeared at around 1800 cm^{-1} (broad), assigning to the $\text{C}=\text{NH}^+$ bond (*Nat. Commun.* 2022, 13, 6317). Moreover, the peak from the stretching mode of imine bonds attenuated. Subsequently, the photocatalytic H_2O_2 evolution performance of all COFs was examined in **Fig. S26**. It was found that a value of $175\text{ }\mu\text{M}$ for the H_2O_2 concentration was achieved by TpaBtt-AC, nearly 3.0 time higher than that of pristine TpaBtt, followed by that of TapbBtt-AC ($216\text{ }\mu\text{M}$), nearly 1.8 times higher than that of pristine TapbBtt. However, H_2O_2 production in TaptBtt was slightly decreased. We further probed the difference in the separation and recombination of charge carriers of the three COFs by photocurrent measurement and electrochemical impedance spectra. Obviously, both TpaBtt-AC and TapbBtt-AC showed a higher current density than the corresponding unprotonated COFs (**Fig. S27a-b**), while TaptBtt-AC exhibited a opposite trend in **Fig. S27c**. Similar trend was also observed in the result of EIS (**Fig. S27d-f**). Considering the above results, we conclude that the performance of TpaBtt can be improved through protonation of imine bonds. The protonation of imine bonds leads to the direction inversion of charge transfer in an intramolecular way. In terms of TaptBtt, the protonation of imine bond is difficult to overcome the larger energy difference of line-region combination between linkages and linkers for achieving the direction reversal of charge transfer.

This result directly proves that TaptBtt indeed has a greater energy difference between building block motif than that of the other two COFs.

As shown in the fs-TA spectrum in **Fig. S28**, after being excited by a pump pulse with a wavelength of 400 nm (consistent with the test conditions of pure COF), the TpaBtt-AC shows obvious negative signal at 500-650 nm compared with pure TpaBtt, which belongs to ground state bleaching. This signal represents the process of donor molecules in the material leaving the ground state to become excited states. The signal of TpaBtt-AC becomes more sustained compared with TpaBtt, indicating that the TpaBtt-AC has more excited states formed under photoexcitation. It enables the charge form/transfer in a manner of more ordered way to a certain extent (*Angew. Chem. Int. Ed.* 2023, 62 e202218688). On the contrary, TaptBtt significantly weakens after protonation, which is consistent with the photoelectrochemical results and the photocatalytic performance. Subsequently, we explore the different spectral features revealed by spectral cross sections at different times and the observed dynamic lines in **Fig. S29**. The TpaBtt-AC showed obvious blue-shift after 0.3 ps around at 550 nm. However, this blue-shift becomes less obvious after protonation, indicating that the short-range push-pull effect is weakened, further demonstrating that the excited electrons of TpaBtt-AC can be efficiently transferred to the receptor unit, rather than being concentrated on the imine bond. For TaptBtt and TaptBtt-AC, the trend of spectral cross section is the opposite of TpaBtt and TpaBtt-AC.

Supplementary Figure 24. Scheme of electron transfer path for both unprotonated COFs and protonated COFs treated by ascorbic acid (AC).

Supplementary Figure 25. Comparison of the FTIR spectra for COFs before and after protonation.

Supplementary Figure 26. Photocatalytic H₂O₂ performance under visible light irradiation for unprotonated and protonated COFs. Conditions: water (50 ml), catalyst (15 mg), 300 W Xe lamp; $\lambda > 420$ nm.

Supplementary Figure 27. (a-c) Transient photocurrent response under visible light irradiation, and (d-e) electrochemical impedance spectroscopy (EIS) of unprotonated and protonated COFs.

Supplementary Figure 28. 2D contour plots of TA spectra for unprotonated and protonated COFs.

Supplementary Figure 29. Spectral signals of TpaBtt-AC and TaptBtt-AC on the ps timescales compared with unprotonated COFs.

2. Uniqueness and importance of photocatalytic H₂O₂ generation based on benzotrithiophene-COFs

The photocatalytic applications of COFs were mainly focused on photocatalytic hydrogen production, water splitting, and organic transformations. The COFs for producing H₂O₂ were reported in 2020 (*J. Am. Chem. Soc.* 2020, 142, 20107–20116). Before the submission of this work to *Nature Communications*, there were only nine relevant articles studying the applications of COFs in H₂O₂ production (*Angew. Chem. Int. Ed.* 2022, 61, e202200413; *Angew. Chem. Int. Ed.* 2022, 61, e202202328; *Environ. Sci.: Nano* 2022, 9, 2464; *J. Am. Chem. Soc.* 2020, 142, 20107–20116; *J. Am. Chem. Soc.* 2022, 144, 9902–9909; *Chem. Eng. J.* 2022, 449, 137802; *Chem. Mater.* 2022, 34, 5232–5240; *Adv. Funct. Mater.* 2021, 2106120; *CCS Chem.* 2022, 4, 3751–3761). Most of them focused on the photosynthetic H₂O₂ with the aid of sacrificial agent system. Little knowledge was shown in pure water and oxygen (derived from natural air) without any sacrificial agents at ambient condition. The reaction path of COFs to trigger the full reaction (2e⁻ ORR and 2e⁻ WOR) is still unclear (*Angew. Chem. Int. Ed.* 2022, 61, e202200413).

Moreover, the thiophene-based COFs, such as trithiophene-based COFs, are rarely used to the artificial H₂O₂ photosynthesis. Thiophene is a stable π -aromatic five-membered ring compound, whose aromatic rings allow the introduction of electroactive and solubilizing groups at each alfa or beta site, with excellent conductivity and adjustable electron density. Functional thiophene motif can be not only pre-assembled to easily construct the donor-acceptor COFs, but also differ the transport direction and dissipation of excited charge carrier in the presence of

polarized imine linkage. It also provides a paradigm to solve the common problem that how the push-pull effects between intramolecular motif and linkage chemistry in polymers affords highly catalytic efficiency. Considering the $n-\pi^*$ transition of the long pairs on sulfur and suitable molecular orbital occupancy, precisely placing independent oxidation and reduction centers in an ordered manner based on thiophene-COFs may be more accessible for photocatalytic H_2O_2 production. Thus, this paper provides a pioneering research basis for green and low-energy synthesis of hydrogen peroxide based on trithiophene-based COFs.

In terms of photocatalysts, we successfully synthesized three kinds of benzotrithiophene-based COFs, termed as TpaBtt, TapbBtt and TaptBtt. These catalysts with spatially independent redox centers could be highly efficient for photocatalytic production of hydrogen peroxide from water and oxygen (derived from natural air) without any sacrificial agents. The yields ($2094 \mu\text{M h}^{-1}$) of TaptBtt without any sacrificial agents at ambient condition exceeds most of reported COFs via synchronous $2e^-$ water oxidation reaction and $2e^-$ oxygen reduction reaction. Pure H_2O_2 can be obtained during the process without sacrificial agent, which can be directly used in Fenton reaction for degradation of sulfamethoxazole after separation and purification with catalysts, and the degradation efficiency can reach 72% within 5 min. The whole process of synthesizing hydrogen peroxide for TaptBtt was revealed. In terms of ORR process, oxygen molecules are selectively adsorbed on the unit near the imine bond, and then carbon atoms located in the electron acceptor fragment nearest the imine linkage use the imine bond as a bridge to obtain electrons to form the $^*\text{OOH}$ intermediate state. For WOR process, two H_2O molecules are adsorbed onto the Tapt unit to form intermolecular hydrogen bond, which is subsequently attacked by photoinduced holes to directly produce H_2O_2 , and the protons in the whole reaction process can be recycled by both processes. Importantly, the energy difference between the imine linkage and its connected unit promotes the efficient separation of photoinduced charge carriers.

To sum up, the new theoretical concept with real and simple H_2O_2 photosynthesis reported in this work represents promising inspiration about how to design efficient COF-based photocatalysts based on an integration of linkage chemistry and linker hybridization. We believe that the importance of this work warrants its publication in Nature Communications.

Reviewer #2 (Remarks to the Author):

The authors made a lot of effort to do the revision according to my suggestions. I am happy to see that fs-TA spectra measurement is done. The quality of the overall manuscript has been improved. The analysis is still not very satisfactory.

The very highlight of this work is the concept of "atomic spot-molecular area" via homodromous dual-donor-acceptor, as the authors claimed several times in the manuscript. The author should try to verify the concept with direct evidence. The catalytic performance can not be direct evidence to verify this concept. Too many factors, such as the BET surface area, crystallinity, the dispersion of the COFs materials, the conditions of the photocatalysis, et al., can influence the results of the catalytic activity. Even the authors also found that the crystallinity of the TaptBtt could influence the result directly (Figure 2e,f). DFT is better to be used as companion evidence.

Response: Thanks for your kind suggestion. We fully agree that the improvement of catalytic performance is a result of the combined effects of various factors. Therefore, we revised the statement in our revised manuscript.

The concept of a double donor-acceptor "atomic spot-molecular area" is to explain the primary driving force for the efficient separation of photogenerated charge carriers during the photocatalytic process. To support our conclusion, we conducted a range of characterizations, including photoluminescence spectra, photocurrent measurement, electrochemical impedance spectra, and femtosecond transient absorption spectroscopy (fs-TA). In line with the study of the synthesized COFs, both the molecular fragment of COFs and the protonated COFs were also prepared. Additionally, we employed density functional theory (DFT) to confirm the charge redistribution of active sites before and after the actual photocatalytic reaction process.

Added in Page 25 of revised manuscript, "Hirshfeld charge analysis uncovers that the average charge of these carbon (C) atoms close to imine linkage is about -0.122 eV for TpaBtt (the thiophene ring in Btt) and increases to -0.119 eV for TaptBtt (the benzene ring of Tapt near imine bond) (Fig. S41). The C1 atom of TaptBtt has a positive value of 0.041 eV, which indicates a strong ability to extract electrons (i.e., Lewis acidity) beneficial for the in-plane charge transfer."

Added in Page 26 of revised manuscript, "Compared with TpaBtt and TapbBtt, charge redistribution of TaptBtt interacted with OOH* is significantly more noteworthy, indicating that the line-region combination of imine linkages and linkers (Tapt and Btt) via dual-donor-acceptor way still determinates the selective formation of OOH* with favorable binding ability in active sites (Fig. R1)."

Supplementary Figure 41. Hirshfeld charge of the different carbon atoms at benzene ring for TpaBtt, TapbBtt and TaptBtt. The white, grey, blue, yellow, and red spheres refer to hydrogen, carbon, nitrogen, sulfur and oxygen, respectively.

Fig R1. Charge difference density between OOH* and adsorption sites on TpaBtt, TapbBtt and TaptBtt, respectively. The yellow represents the electron accumulation area, and the green represents the electron dissipation area. The white, grey, blue, yellow and red spheres refer to hydrogen, carbon, nitrogen, sulfur, and oxygen, severally.

In the analysis of the fs-TA spectra, the authors claimed that the redshift amplitudes are the result of the electron-deficient N bridging of imines to the electron-acceptor unit, further demonstrating that COFs can achieve efficient photogenerated charge transfer using the push-pull mechanism from energy difference. Why? Please give an explanation in detail. And for TaptBtt, the redshift is not obvious. How to explain that by using the same logic?

Response: We really appreciate the question. The dynamics of excited state relaxation is mainly determined by the magnitude of intramolecular charge-transfer (ICT) in molecules (*J. Phys. Chem. A* 2007, 111, 5806-5812; *J. Phys. Chem. C* 2016, 120, 13922-13930; *J. Phys. Chem. A* 2015, 119, 1964-1972). When a bridging unit is inserted between two push-pull units, the push-pull effect can increase the transition dipole moment of monomer structure, thereby enhancing the coupling between functional primitives and a larger fraction of excited delocalization (*J. Phys. Chem. C* 2016, 120, 13922-13930). This feature can be intuitively confirmed by the peak shift on the fs-TA spectrum. That is, the peak shift under the gradually increasing time delay can explain the characteristics of charge transfer between the donor and acceptor units. Inspired by this, the peak shift at progressively increasing time delays could explain the charge-transfer character of the D-A units via imine linkage in COFs.

Compared with the other two COFs, the redshift of TaptBtt before 600 ps is obvious. Although the displacement intensity is not as obvious as that of TapbBtt after 600 ps, the signal amplitude of optical density (ΔOD) is much stronger than that of TapbBtt. Previous studies have shown that the signal amplitude of optical density (ΔOD) is proportional to the number of excitons (*J. Am. Chem. Soc.* 2020, 142, 14957-14965). It can be inferred that TaptBtt shows better push-

pull performance and more excitons under the combined action of peak shift and optical density signals.

And I could not find the fitting dynamics which show the variation of two decay time constants in Fig 4i. I think it is supposed to be Fig. 3i. That result is positive. However, in my opinion, this evidence is still not sufficient to prove the concept of "atomic spot-molecular area" Very interestingly, the authors gave very good references, which are the protonated COFs. I would highly suggest comparing the fs-TA spectra of the pristine and protonated COFs. The difference may give direct and clear evidence. In addition, the study of the molecular segments of the three COFs as small molecules can also be helpful and also suggested. The author should try to prove the push-pull interaction occurred in the "atomic spot-molecular area," not the integral results.

Response: According to the reviewer's suggestion, we successfully synthesized molecular segments of the three COFs, and further demonstrated that push-pull interactions occur in the "atomic point-molecular region" by using the corresponding properties of these small molecules. In addition, we also characterized TA of the three materials after imine protonation.

1. Synthesis and characterization of three kinds of model compounds (Added in Page 5 of revised Supplementary Information)

1.1 Synthesized procedure of TpaBtt-fragment

A mixture of 4,4,4'-triaminodiphenylamine (0.05 mmol, 14.52 mg) and benzo[1,2-b:3,4-b':5,6-b'']trithiophene-2-tricarbaldehyde (0.2 mmol, 54.88 mg) was dissolved in ethanol (4 mL) and trichloromethane (4 mL), and the mixture was heated to reflux for 24 h. After cooling to room temperature, the precipitate was collected by filtration, washed with anhydrous ethanol, and dried under vacuum to give a red solid. ^{13}C NMR (δ , CDCl_3): 150.79, 145.31, 130.47, 123.56.

1.2 Synthesized procedure of TapBtt-fragments

A mixture of 1,3,5-tris(4-aminophenyl) benzene (0.05 mmol, 17.55 mg) and benzo[1,2-b:3,4-b':5,6-b'']trithiophene-2-tricarbaldehyde (0.2 mmol, 54.88 mg) was dissolved in ethanol (4 mL) and trichloromethane (4 mL), and the mixture was heated to reflux for 24 h. After cooling to room temperature, the precipitate was collected by filtration, washed with anhydrous ethanol, and dried under vacuum to give a red solid. ^{13}C NMR (δ , CDCl_3): 150.44, 148.70, 140.39, 129.31, 125.58, 122.67, 116.01.

1.3 Synthesized procedure of TaptBtt-fragments

A mixture of 2,4,6-tris(4-aminophenyl)-1,3,5-triazine (0.05 mmol, 17.70 mg) and benzo[1,2-b:3,4-b':5,6-b'']trithiophene-2-tricarbaldehyde (0.2 mmol, 54.88 mg) was dissolved in ethanol (4 mL) and trichloromethane (4 mL), and the mixture was heated to reflux for 24 h. After cooling to room temperature, the precipitate was collected by filtration, washed with anhydrous ethanol, and dried under vacuum to give a red solid. ^{13}C NMR (δ , CDCl_3): 169.35, 151.85, 141.93, 132.69, 129.45, 122.44, 115.32.

1.3 Solid state ^{13}C NMR of three kinds of model compounds

Fig R2. ^{13}C NMR (125 MHz, DMSO-d_6) spectra of model molecules. The NMR results showed that the molecular fragments of the three COFs had peaks assigned to the carbon of imine bond at 151 ppm.

2. Studies of three kinds of model compounds

The molecular fragments of three kinds of COF were successfully synthesized using benzo[1,2-b:3,4-b':5,6-b'']-trithiophene-2-tricarbaldehyde and different monomers, including 4,4,4-triaminotriphenylamine, 1,3,5-tris(4-aminophenyl) benzene, and 2,4,6-tris(4-aminophenyl)-1,3,5-triazine under the reflux of trichloromethane and ethanol solution for 24 h. The obtained model compounds were respectively named as TapBtt-fragment, TpaBtt-fragment, and TaptBtt-fragment. We firstly probed the difference in the separation and recombination of charge carriers of these model compounds by photocurrent measurement and electrochemical impedance (EIS) spectra. According to **Fig. S2**, TaptBtt-fragment had higher photocurrent density with a lesser charge transference resistance, indicating that the TaptBtt-fragment showed the upper charge separation capability and more available surface photogenerated carriers for solid-liquid interfacial reaction. Subsequently, as demonstrated by time-resolved fluorescence spectroscopy (**Fig. S3**) under the excited wavelength of 350 nm, the fluorescence lifetime of TaptBtt-fragment (1.40 ns) was obviously higher than that of TapbBtt-fragment (1.19 ns) and TpaBtt-fragment (0.74 ns). TaptBtt-fragment was able to suppress the e^-h^+ recombination more effective than that of TapbBtt-fragment and TpaBtt-fragment (*Angew. Chem. Int. Ed.* 2023, 62, e202216073). To get insight into the charge transfer pathway, the X-band electron paramagnetic resonance (EPR) experiments were also conducted. In **Fig. S4**, the relative intensity (0.31) of EPR signal for the TpaBtt-fragment in dark and light condition is significantly higher than that of TapbBtt-fragment (0.29) and TaptBtt-fragment (0.23). The less relative strength for TaptBtt-fragment is probably related to the ability of ground state charge transfer from the acceptor unit to the donor unit (*Nat. Commun.* 2019, 10, 5538; *Nat. Commun.* 2018, 9, 1182; *Adv. Sustainable syst.* 2022, 6, 2100264), which is dominated by the combined interaction of the directivity of double D-A structure and the energy difference in these molecular fragments. Integrating photoelectrochemical measurement and PL lifetime with EPR results, TaptBtt-fragment owns more efficient charge transfer due to the

favorable push-pull effects between intramolecular motif and imine bond. Therefore, it can be concluded that the push-pull interaction of “atomic spot-molecular area” occurs in both aperiodic segments and COFs.

Added in Page 8 of revised manuscript and Page S14-S15 of Supplementary Information, “Molecular segments of the three COFs were used to verify that the push-pull interactions occur in the “atomic point-molecular region”. According to the results of both photoelectrochemistry and time-resolved fluorescence spectroscopy (Supplementary Fig. 2-3), TaptBtt-fragment shows the upper charge separation capability and more available suppressed e^-h^+ recombination. To get insight into the electron transfer direction for fragments, in-situ X-band electron paramagnetic resonance (EPR) experiments were then conducted (Supplementary Fig. 4). The relative intensity (0.31) of EPR signal for the TpaBtt-fragment in dark and light condition is significantly higher than that of TapbBtt-fragment (0.29) and TaptBtt-fragment (0.23). The less relative strength for TaptBtt-fragment is probably related to the ability of ground state charge transfer from the acceptor unit to the donor unit, which is dominated by the integrated interaction of the directivity of double D-A structure and the energy difference in these molecular fragments. Integrating photoelectrochemical measurement and PL lifetime with EPR results, TaptBtt-fragment owns more efficient charge transfer due to the favorable push-pull effects between intramolecular motif and imine bond. Therefore, it can be concluded that the push-pull interaction of “atomic spot-molecular area” occurs in both aperiodic segments and COFs”.

Supplementary Figure 2. (a) Transient photocurrent responses under visible light irradiation, and (b) electrochemical impedance spectroscopy (EIS) of fragments.

Supplementary Figure 3. Fluorescence lifetime decay of COF fragments.

Supplementary Figure 4. EPR spectra of all COF fragments under dark and light.

3. Comparison study of protonated and unprotonated COFs

Added in Page S31-S32 of revised Supplementary Information, “As shown in the fs-TA spectrum in Fig. S28, after being excited by a pump pulse with a wavelength of 400 nm (consistent with the test conditions of pure COF), the TpaBtt-AC shows obvious negative signal at 500-650 compared with pure TpaBtt, which belongs to ground state bleaching. This signal represents the process of donor molecules in the material leaving the ground state to become excited states. The signal of TpaBtt-AC becomes more sustained compared with TpaBtt, indicating that the TpaBtt-AC

has more excited states formed under photoexcitation. It enables the charge form/transfer in a manner of more ordered way to a certain extent (*Angew. Chem. Int. Ed.* 2023, 62, e202218688). On the contrary, TaptBtt significantly weakens after protonation, which is consistent with the photoelectrochemical results and the photocatalytic performance. Subsequently, we explore the different spectral features revealed by spectral cross sections at different times and the observed dynamic lines in **Fig. S29**. TpaBtt-AC showed obvious blue-shift after 0.3 ps around at 550 nm. However, this blue-shift becomes less obvious after protonation, indicating that the short-range push-pull effect is weakened, further demonstrating that the excited electrons of TpaBtt-AC can be efficiently transferred to the receptor unit, rather than being concentrated on the imine bond. For TaptBtt and TaptBtt-AC, the trend of spectral cross section is the opposite of TpaBtt and TpaBtt-AC”.

Supplementary Figure 28. 2D contour plots of TA spectra for unprotonated and protonated COFs.

Supplementary Figure 29. Spectra signals of TpaBtt-AC and TaptBtt-AC on the ps timescales compared with pure COFs.

Besides, the following problems should be paid attention to.

1. For the PXRD of the three COFs, it was always observed a sharp decrease before 5 degree. Please give an explanation and provide the raw data.

Response: Thanks for the valuable suggestion of the reviewer. The sharp decrease before 5 degrees for PXRD of the three COFs is probably due to the scattering diffraction of X-rays in the air. Similar results had been reported in the literature (*J. Am. Chem. Soc.* 2022, 144, 17097–17109; *Angew. Chem. Int. Ed.* 2023, 62, e202300224). The corresponding raw data have been provided along with the attachment.

2. I am surprised that the author put Figure S7 in SI. It is unacceptable for a chemist. No signals are from the compound itself. It looks like the signals of the deuterated DMSO.

Response: Thank you for pointing out this problem. The benzo[1,2-b:3,4-b':5,6-b'']trithiophene-2,5,8-tricarbaldehyde (Btt) was prepared according to a modified literature method (*J. Am. Chem. Soc.* 2018, 140, 11618–11622; *J. Am. Chem. Soc.* 2023, 145, 8364–8374). After many attempts, we could not measure the relevant solid-state $H^1/^{13}C$ NMR data due to the bad solubility of the product in any common solvents. Their structure can be indirectly proven through the NMR characterization of its Schiff-based condensation product with 4-tert-Butyl benzenamine according to the previously reported literature (*J. Am. Chem. Soc.* 2018, 140, 11618–11622).

Added in Page S2-S4 of revised Supplementary Information:

1,3,5-Trichloro-2,4,6-tris(dichloromethyl)benzene was prepared according to a modified literature method (*Org. Lett.* 2009, 11, 3230). To a 100 mL polytetrafluoroethylene lined reactor was added 1,3,5-trichlorobenzene (3.0 g, 16.5 mmol), aluminium chloride (2.6 g, 19.5 mmol) and 60 mL trichloromethane, and then the mixture was stirred at 125 °C. After 6 h and 18 h of reaction, it was cooled down to room temperature and the lining was opened to release the gas. It was cooled down to room temperature and poured into ice water after 3 d. The mixture was stirred for 1 h and extracted with dichloromethane, and solvents were evaporated under reduced pressure. The crude product was chromatographed on silica gel with petroleum ether to get a near white solid (5.56 g, 78.3%) as product. $^1\text{H NMR}$ (400 MHz, CDCl_3): δ 7.75 (s, 2H), 7.63 (s, 1H) ppm.

1,3,5-Trichloro-2,4,6-tris(dichloromethyl)benzene was prepared according to a modified literature method (*Org. Lett.* 2009, 11, 3230). To a 25 mL flask was added 1,3,5-trichloro-2,4,6-tris(dichloromethyl)benzene (1.32 g, 3.07 mmol), FeSO_4 (60 mg, 0.39 mmol) and 8 mL concentrated sulfuric acid, and the mixture was stirred at 125 °C overnight. After cooling down to room temperature, the reddish-brown oily liquid was poured into ice, which was stood for a few minutes. Subsequently, the mixture was extracted with CH_2Cl_2 and solvents were evaporated under reduced pressure. The crude product was chromatographed on silica gel with petroleum ether to get a white solid (0.44 g, 54.0%). $^1\text{H NMR}$ (400 MHz, CDCl_3): δ 10.42 (s, 3H) ppm.

Benzo[1,2-b:3,4-b':5,6-b'']trithiophene-2,5,8-tricarbaldehyde (BTT) was prepared according to a modified literature method (*J. Am. Chem. Soc.* 2018, 140, 11618–11622; *J. Am. Chem. Soc.* 2023, 145, 8364–8374). To a mixture of 2,4,6-trichlorobenzene-1,3,5-tricarbaldehyde (160 mg, 0.6 mmol), pdithiane-2,5-diol (137 mg, 0.9 mmol) and anhydrous DMF (2.2 mL) was added triethylamine (0.5 mL, 3.6 mmol) dropwise in the ice bath. Then the mixture was stirred at 35 °C overnight. It was poured into ice water and centrifuged after returning to room temperature, washed with water and tetrahydrofuran repeatedly. Finally, a pale brown solid was obtained with 71.6% yield.

(1E,1'E,1''E)-1,1',1''-(Benzo[1,2-b:3,4-b':5,6-b'']trithiophene-2,5,8-triyl)tris(N-(4(tertbutyl) phenyl) methanimine) (the model compound) was prepared according to a modified literature method (*J. Am. Chem. Soc.* 2018, 140, 11618–11622). To a 10 mL flask was added benzo[1,2-b:3,4-b':5,6-b'']trithiophene-2,5,8-tricarbaldehyde (50 mg, 0.15 mmol), and 4-tert-butylaniline (86 μ L, 0.54 mmol), 6M HOAc (0.15 mL) and 1,4-dioxane (4.5 mL) were charged into a 10 mL Shlenk reaction tube. After sonication for 5 min and degassed by three freeze-pump-thaw cycles, the reaction system was heated at 120 °C for 5 h. After cooling to room temperature, the precipitate was filtered and recrystallized in 1,4-dioxane to obtain yellow needle-like solid (66.1 mg, 61%). ^1H NMR (400 MHz, CDCl_3): δ 9.96 (s, 3H), 8.25 (s, 3H, thiophene), (1.77 (s, 27H, CH_3) ppm. ^{13}C NMR (100 MHz, CDCl_3): δ 151.28, 148.02, 143.91, 138.25, 135.70, 131.93, 126.04, 122.52, 35.0, 31.8 ppm.”

Figure R3 Solid state ^1H NMR and ^{13}C NMR spectra of the target compound.

3. Please give the .cif files of the final structures of the three COFs with AA and AB stacking modes. The distance between the layers looks unreasonably large. Please explain this. I would suggest to double check the models. The models are very different from the reported *Adv. Funct. Mater.* 2022, 32, 2112553.

Response: According to the reviewer’s suggestion, we used Material Studio software to restimulate the crystal structure, and the Pawley refinement demonstrated the good fit of the eclipse stacking model (AA stacking) for three COFs (**Fig. S5**). The optimized PXRD displayed TpaBtt, TapbBtt and TaptBtt the (100) reflection at $2\theta = 5.43^\circ$, 4.84° and 4.76° . Pawley refines of the AA stacking model based on the experimental profiles gave unit cell with parameters ($a=b=19.1881 \text{ \AA}$, $c=3.5238$, $\alpha=\beta=90^\circ$, $\gamma=120^\circ$) for TpaBtt (residuals $R_p=3.01\%$, $R_{wp}=4.05\%$), ($a=b=21.350 \text{ \AA}$, $c=3.505$, $\alpha=\beta=90^\circ$, $\gamma=120^\circ$) for TapbBtt (residuals $R_p=2.75\%$, $R_{wp}=4.55\%$), and ($a=b=21.755 \text{ \AA}$, $c=3.493$, $\alpha=\beta=90^\circ$, $\gamma=120^\circ$) for TaptBtt (residuals $R_p=3.99\%$, $R_{wp}=8.63\%$). The corresponding cif files of the three COFs are provided.

Supplementary Figure 5. Experimental, refined and simulated PXRD patterns of TpaBtt (a), TapbBtt (b) and TaptBtt (c). Insert represented the simulated eclipsed (AA) and staggered (AB) stacking models of the corresponding imine 2D COF.

4. Please cite Adv. Funct. Mater. 2022, 32, 2112553.

Response: Thanks for the valuable suggestion of the reviewer. The corresponding reference has been cited to the revised manuscript.

- The reference (Adv. Funct. Mater. 2022, 32, 2112553) has been cited on Page 9 in our revised manuscript: The crystalline construction of the COFs was confirmed by powder X-ray diffraction (PXRD) in Fig. 2a. The Pawley refinement demonstrated the good fit of the eclipse stacking model (AA stacking) for three COFs (Adv. Funct. Mater. 2022, 32, 2112553). The optimized PXRD displayed the (100) reflection of TpaBtt, TapbBtt and TaptBtt at $2\theta = 5.43^\circ$, 4.84° and 4.76° , respectively (Fig. S5).

Reviewer #3 (Remarks to the Author):

1) Firstly, and I should have picked this up in my original review, but I am not sure about the word “Homodromous” in the article title. It’s a word I have never come across and so might not be obvious word to any chemist. Worse, I looked it up in some dictionaries and while the Oxford English Dictionary gives “Running in the same direction: opposed to heterodromous” besides a botany based definition “Turning in the same direction, as two generating spirals of a phyllotaxis (e.g. on the main stem and on a branch)”, Merriam Webster only gives the botany based definition “having the genetic spiral following the same direction in both stem and branches”, while the word doesn’t appear in other online dictionaries. Hence, I think it’s best to avoid the word homodromous in the title and the text, replace it with a better word or perhaps just leave it

Response: We thank the reviewer for the helpful comments to our work. In line with the reviewer’s suggestions, the word of “homodromous” was removed in our revised manuscript.

2) The authors have added some text about previous work on page 4 which reads “vinyl, fluorinated and Ti-based COFs”. While the vinyl and fluorinated materials are probably COFs, Ti-based COFs sounds more like a MOF than a COF.

Response: Thanks for the reviewer’s careful review. After the careful examination, Ti-based COF is considered as a metal-loaded COF rather than pure MOF. The literature describes a titanium-based COF with *spn* topology (TiCOF-*spn*) through [6+3] imine condensation of a Ti(IV) complex with six aldehyde groups positioned in a trigonal antiprismatic arrangement and a planar triazine-based amine connector. Its XRD pattern proved that it is a COF. Thanks for the understanding.

3) When discussing the HOMO and LUMO of the COFs on page 6/7 the authors should clarify to the reader that (i) these are predictions rather than measurements and (ii) that the calculations were done on isolated fragments of the COFs than rather than the fully periodic COF.

Response: Thanks for the valuable suggestion. According to the suggestion, the HOMO and LUMO of the COFs on page 6/7 have been rewritten. It should be noted that these results are predictions rather than practical measurements. Secondly, our calculations are performed on periodic unit cells of the three COFs, rather than on isolated fragments of the COFs. **Fig. S1** we provided did not display the full periodic framework, which might make the reviewer misunderstanding. A new **Fig. S1** was replaced in our revised manuscript.

Added in Page 6 of revised supporting information, “The highest occupied molecular orbital (HOMO) and lowest unoccupied molecular orbitals (LUMO) for three COFs were predicted and displayed in **Fig. S1.**”

Supplementary Figure 1. Calculated HOMO-LUMO distribution of TpaBtt, TapbBtt and TaptBtt.

4) The authors have added a discussion about transition probability analysis (page 7, supplementary information table 1) but it's unclear to me what this analysis involves as little detail of how this analysis is performed is given. The information in table 1 as it stands makes little sense. In theory one could calculate excited states of the fragments with TD-DFT and then analyse the character of these excited states in terms of a projection of the excited state on excitations between orbitals but TD-DFT (or other method that could have been used CI/S, CC2 etc.) is not mentioned. Alternatively, one could crudely approximate excited states just in terms of orbital energy differences but in that case there would be no percentage contributions as every excitation is approximated to be identical to an excitation of a single electron between one occupied and unoccupied orbital.

Response: Thanks for your kind suggestion. In the revised manuscript, we added TD-DFT to calculate the excited states of the three COF fragments, and analyzed the characteristics of the excited states according to the oscillator strength.

Added in Page 7 of revised manuscript, “Based on the time dependent-density functional theory (TD-DFT) simulation, we calculated transition energies and probabilities of each excited state for all COFs. The excited states of S_0-S_2 , S_0-S_1 and S_0-S_3 have the strongest oscillator strength on the two fragments for TpaBtt, TapbBtt and TaptBtt, respectively (**Supplementary Table 1**). According to the transition probability analysis of molecular orbitals (MO), most of electrons were contributed by the LUMO (MO 149, 165 and 159 for respective TpaBtt, TapbBtt and TaptBtt) in transition process, indicating that the electronic configuration of LUMO nearly represents photogenerated electron composition.”

Added Supplementary Table 1. Calculated molecular orbital transition (MOT), contribution of transition, oscillator strength (f) excitation energy (E) at different excited states for TpaBtt, TapbBtt and TaptBtt.

Model	MOT	MO contribution of transition (%)	E (eV) [λ (nm)]	f
TpaBtt	HOMO \rightarrow LUMO	35	2.43 [510]	0.0132
	HOMO \rightarrow LUMO+2	41	2.51 [494]	0.0045
	HOMO-1 \rightarrow LUMO	58	2.64 [469]	0.6313
	HOMO-2 \rightarrow LUMO	44	2.85 [435]	0.0071
	HOMO-2 \rightarrow LUMO+1	52	2.87 [432]	0.0063

Model	MOT	MO contribution of transition (%)	E (eV) [λ (nm)]	f
TapbBtt	HOMO \rightarrow LUMO	75	2.31 [536]	1.240
	HOMO-2 \rightarrow LUMO	24	2.52 [491]	0.0019
	HOMO-3 \rightarrow LUMO	60	2.74 [470]	0.0507
	HOMO-4 \rightarrow LUMO	16	2.90 [427]	0.0077
	HOMO \rightarrow LUMO+4	58	3.08 [403]	0.04817

Model	MOT	MO contribution of transition (%)	E (eV) [λ (nm)]	f
TaptBtt	HOMO-2 \rightarrow LUMO	73	2.32 [534]	1.0478
	HOMO \rightarrow LUMO	21	2.06 [602]	0.0092
	HOMO-1 \rightarrow LUMO	21	2.07 [598]	0.0087
	HOMO-2 \rightarrow LUMO	39	2.72 [456]	0.0013
	HOMO-5 \rightarrow LUMO+1	6	2.21 [560]	0.0002

5) It would make sense to show in Fig. 3F the 2e⁻ water oxidation and reduction potentials at pH 5 rather than pH 0.

Response: Thanks for the reviewer's advice. Fig. 3f shows the 2e⁻ water oxidation and reduction potentials at pH 5 in page 14 of the revised manuscript.

Fig. 3f Energy band values of these three COFs. Red and green lines stand for 2e⁻ ORR and 2e⁻ WOR.

6) The isotope measurements are a great addition, but the discussion needs to be clearer. A first stage and second stage are mentioned but not explained. For me it would make more sense to discuss the isotope experiments in terms of in the dark, in the light, and/before and after addition of MnO₂.

Response: Thanks for the valuable suggestion. According to the advice, we rewrote the results of the isotope experiments in our revised manuscript.

Added in Page 21 of revised manuscript, "Subsequently, H₂¹⁸O was used in photocatalytic tests to further identify the two/four-electron water oxidization. As shown in Supplementary Fig. 30, none of the three COFs was detected for ¹⁸O₂ production in the first stage including dark, light and before addition of MnO₂, while all did in the second stage-decomposition of photogenerated H₂O₂ by the MnO₂ (Fig. 4d and Supplementary Fig. 31). However, we can clearly see that the ratio of two types of oxygen (¹⁸O₂ and ¹⁶O₂) is significantly different after H₂O₂ decomposition in the second step. The ratio of ¹⁸O₂ and ¹⁶O₂ is 1:4.8 (close to the four-electron water oxidation process, Eq.7) for TpaBtt and TapbBtt, while the ratio is 1:1.2 (close to the two-electron water oxidation process, Eq. 3) for TaptBtt 47, 60, 61. In addition, there is a difference trend of H₂O₂ concentration after adding a sacrificial agent, and H₂O₂ can still be detected under the atmosphere of Ar for TpaBtt, TapbBtt and TaptBtt (Fig. 4a and Supplementary Fig. 27) ^{43, 47}. For TpaBtt and TapbBtt, it was reasonably concluded that this four-electron process is involved in the synthesis of hydrogen peroxide. The oxygen produced by the four-electron water oxidation is extremely tiny and may

be adsorbed on the surface of COFs, then being directly used to the formation of H₂O₂. Thus, the four-electron process has a little contribution to H₂O₂ production for TpaBtt and TapbBtt. This also explains the absence of O₂ in the RRDE (Supplementary Fig. 29a) and oxygen-producing isotopes (Supplementary Fig. 30). Therefore, these results give solid supports that the H₂O₂ photosynthesis undergoes 2e⁻ ORR and 4e⁻ ORR for TpaBtt and TapbBtt, while TaptBtt has 2e⁻ ORR and 2e⁻ WOR dual processes with higher atomic efficiency.”

7) On page 21 the authors write “For TpaBtt and TapbBtt, it was reasonably concluded that they can undergo four-electron water oxidation to produce O₂, and this part of O₂ is weak and may be adsorbed on the surface of COFs, then being directly used to produce H₂O₂ under the radiation. However, this four-electron process has a little contribution to H₂O₂ production for TpaBtt and TapbBtt. This also explains the absence of O₂ in the RRDE (Supplementary Fig. 29a) and oxygen-producing isotopes (Supplementary Fig. 30).”. While I agree with the analysis in terms of TaptBtt performing 2e⁻ water oxidation and the other materials the thermodynamic easier 4e⁻ water oxidation, I have absolutely no idea what the authors mean in the paragraph above with “this part of O₂ is weak” or “this four-electron process has a little contribution to H₂O₂ production”. What does it mean that O₂ is weak and while the 4 e⁻ oxidation of water obviously produces no H₂O₂ as many holes are consumed by 4e⁻ water oxidation as electrons consumed in water reduction to H₂O₂ and if the water oxidation is slow for TpaBtt and TapbBtt the H₂O₂ production must also be slow.

Response: Thanks for the reviewer’s comments. The previous labelling experiments (the second stage-decomposition of photogenerated H₂O₂ by the MnO₂) proved that there was a four-electron water oxidation process in TpaBtt and TapbBtt, and the generated oxygen participated in the formation of hydrogen peroxide. However, there was no oxygen production detected in the first phase of the isotope including dark, light and before addition of MnO₂. Therefore, we believe that the amount of oxygen produced by the four-electron water oxidation is very little, and is directly adsorbed on the surface of the material to produce hydrogen peroxide. To avoid ambiguity, we have improved this sentence.

Added in Page 21 of revised manuscript, “For TpaBtt and TapbBtt, it was reasonably concluded that this four-electron process is involved in the synthesis of hydrogen peroxide. The oxygen produced by the four-electron water oxidation is tiny and may be adsorbed on the surface of COFs, then being directly used to the formation of H₂O₂.”

Reviewer #4 (Remarks to the Author):

I am satisfied with the revisions made by the authors in response to my comments and am happy to approve the manuscript for publication in its current form.

Response: We thank the reviewer for the recommendation of publication.

REVIEWER COMMENTS

Reviewer #2 (Remarks to the Author):

The authors have addressed most of my concerns. The manuscript can be accepted for publication.

Reviewer #3 (Remarks to the Author):

Sorry with respect to my comment 4 in the last review round, I think the discussion of the data in supplementary table 1 still makes no sense.

In the main text the authors write “The excited states of S0-S2 , S0-S1 and S0-S3 have the strongest oscillator strength on the two fragments for TpaBtt, TapbBtt and TaptBtt, respectively (Supplementary Table 1)”. This description is problematic as the ground state is labelled S0 and the excited states are labelled S1, S2, S3,... There is no such thing as S0-S1 state, S0-S2, S0-S3 states! I suspect the authors got confused with an orbital description, where one describes an excited state in terms of the main orbital from which an electron gets removed and the main orbital in which the excited electron ends up, e.g. HOMO->LUMO. One can say that the S1 state has mostly HOMO->LUMO (mostly, as generally any excited state can be described as a combination of single electron excitations between occupied and unoccupied orbitals) but labelling a state somehow as S0-S1 wrongly mixes up both descriptions.

Supplementary table 1 then doesn't include the Sn excited state labels but gives excited state in terms of their main orbital nature, e.g. HOMO-LUMO, the percentage of which this main single electron excitation contributes to the excited state, the excitation energy and wavelength, and the oscillator strength. If I'm correct and the table gives for each fragment the different singlet excited-states in increasing energies then there should be an additional column, most sensibly the second column after the fragment name, labelling the first excited state as S1, the second excited state as S2, etc. etc. Also, if the main single electron excitation doesn't describe the majority of the excitation, i.e. a contribution less than say 80%, then the authors should probably give not just the main contribution but the first n contribution, the contribution of which sum up to say 80%, e.g. 35% HOMO->LUMO, 30% HOMO->LUMO+1, 15% HOMO-1->LUMO.

No computational details about the TD-DFT calculations underlying the data in supplementary table 1 seems to be given in the main text and supplementary information, other than the mention of time dependent-density functional theory (TD-DFT) in the main text (which btw should be time-dependent density functional theory instead of time dependent-density functional theory). The authors should mention the computational details of the TD-DFT calculations in the method section in the supplementary information and specifically mention the DFT functional and basis-set in the caption of supplementary table 1.

Response Letter for NCOMMS-22-36081B

Reviewer #2 (Remarks to the Author):

The authors have addressed most of my concerns. The manuscript can be accepted for publication.

Response: We thank the reviewer for the time and efforts on our work.

Reviewer #3 (Remarks to the Author):

Sorry with respect to my comment 4 in the last review round, I think the discussion of the data in supplementary table 1 still makes no sense.

In the main text the authors write “The excited states of S0-S2 , S0-S1 and S0-S3 have the strongest oscillator strength on the two fragments for TpaBtt, TapbBtt and TaptBtt, respectively (Supplementary Table 1)”. This description is problematic as the ground state is labelled S0 and the excited states are labelled S1, S2, S3,... There is no such thing as S0-S1 state, S0-S2, S0-S3 states! I suspect the authors got confused with an orbital description, where one describes an excited state in terms of the main orbital from which an electron gets removed and the main orbital in which the excited electron ends up, e.g. HOMO->LUMO. One can say that the S1 state has mostly HOMO->LUMO (mostly, as generally any excited state can be described as a combination of single electron excitations between occupied and unoccupied orbitals) but labelling a state somehow as S0-S1 wrongly mixes up both descriptions.

Response: Thanks for the reviewer’s advice, and we have corrected our descriptions in the revised manuscript accordingly.

Added in Page 7 of Revised Manuscript: “Based on the time-dependent density functional theory (TD-DFT) simulations, we calculated transition energies and probabilities of each excited state for all COFs. The strongest oscillator strength on fragments of TpaBtt, TapbBtt and TaptBtt indicated that the S1 state has mostly HOMO to LUMO transition (Supplementary Table 1).”

Supplementary table 1 then doesn’t include the Sn excited state labels but gives excited state in terms of their main orbital nature, e.g. HOMO-LUMO, the percentage of which this main single electron excitation contributes to the excited state, the excitation energy and wavelength, and the oscillator strength. If I’m correct and the table gives for each fragment the different singlet excited-states in increasing energies then there should be an additional column, most sensibly the second column after the fragment name, labelling the first excited state as S1, the second excited state as S2, etc. etc. Also, if the main single electron excitation doesn’t describe the majority of the excitation, i.e. a contribution less than say 80%, then the authors should probably give not just the main contribution but the first n contribution, the contribution of which sum up to say 80%, e.g. 35% HOMO->LUMO, 30% HOMO->LUMO+1, 15% HOMO-1->LUMO.

Response: Thanks for raising a nice question. We have made the necessary modifications to the description of TD-DFT data. Specifically, we have included a new column as the second column of Supplementary Table 1 to illustrate the different excitation process of each fragment. As recommended by the reviewer, we have provided both the main contribution and the first n contribution in Supplementary Table 1 (*Angew. Chem. Int. Ed.* **2022**, 61, e202202328).

Revised Table 1 in Supplementary Information:

Supplementary Table 1. Calculated molecular orbital transition (MOT), contribution of transition, and excitation energy (E) at different excited states for COFs. Excited states calculated by the CAM-B3LYP functional and the def2-SVP basis set via ORCA software.

Model	Excitation	MOT	MO contribution of transition (%)	E(eV) [λ] (nm)	f
TpaBtt	S0→S1	HOMO-4 →LUMO	1.19	378 [3.28]	1.2955
		HOMO-3 →LUMO	3.35		
		HOMO-1 →LUMO	6.12		
		HOMO →LUMO	79.72		
	S0→S2	HOMO-2 →LUMO	68.25	310 [4.00]	0.0151
		HOMO-2 →LUMO+2	5.59		
		HOMO-1 →LUMO+1	13.25		
		HOMO →LUMO+1	4.68		
	S0→S3	HOMO-9 →LUMO	13.80	304 [4.08]	0.1496
		HOMO-4 →LUMO	8.70		
		HOMO-3 →LUMO	5.82		
		HOMO-2 →LUMO	13.09		
		HOMO-2 →LUMO+1	2.75		
		HOMO-1 →LUMO	54.81		
		HOMO →LUMO+2	10.06		
	S0→S4	HOMO-4 →LUMO	1.29	290 [4.27]	0.0274
		HOMO-3 →LUMO	1.08		
		HOMO-3 →LUMO+6	1.04		
		HOMO-1 →LUMO	5.56		
		HOMO-1 →LUMO+3	3.44		
HOMO →LUMO		1.49			
HOMO →LUMO+3		65.84			
HOMO →LUMO+5		1.98			
HOMO →LUMO+6		1.64			

Model	Excitation	MOT	MO contribution of transition (%)	E(eV) [λ] (nm)	f
TapbBtt	S0→S1	HOMO-9 →LUMO	4.00	354 [3.51]	1.3681
		HOMO-4 →LUMO	3.51		
		HOMO →LUMO	82.86		
	S0→S2	HOMO-2 →LUMO+2	4.34	311 [3.99]	0.01853
		HOMO-1 →LUMO	71.04		
		HOMO →LUMO+2	11.41		
S0→S3	HOMO-10 →LUMO	9.26	304 [4.08]	0.1234	

		HOMO-9→LUMO	12.91		
		HOMO-5→LUMO	6.26		
		HOMO-4→LUMO	5.80		
		HOMO-2→LUMO	41.86		
		HOMO→LUMO	7.05		
	S0→S4	HOMO-9 →LUMO+1	3.33	271 [4.58]	0.0013
		HOMO-3→LUMO	16.49		
		HOMO-3 →LUMO+3	24.96		
		HOMO-2 →LUMO+1	19.61		
		HOMO→LUMO+1	17.21		

Model	Excitation	MOT	MO contribution of transition (%)	E(eV) [λ] (nm)	f
TaptBtt	S0→S1	HOMO-2 →LUMO	12.13	356 [3.49]	1.4997
		HOMO→LUMO	73.35		
	S0→S2	HOMO-1 →LUMO	63.38	314 [3.94]	0.0209
		HOMO-1 →LUMO+2	15.01		
		HOMO→LUMO+3	11.61		
	S0→S3	HOMO-13 →LUMO	4.68	306 [4.05]	0.0829
		HOMO-11 →LUMO	12.62		
		HOMO-2→LUMO	40.19		
		HOMO-2 →LUMO+2	6.38		
		HOMO→LUMO	11.45		
	S0→S4	HOMO-10 →LUMO+1	28.26	278 [4.45]	0.0007
		HOMO-9→LUMO	31.99		
		HOMO-9 →LUMO+2	30.34		

No computational details about the TD-DFT calculations underlying the data in supplementary table 1 seems to be given in the main text and supplementary information, other than the mention of time dependent-density functional theory (TD-DFT) in the main text (which btw should be time-dependent density functional theory instead of time dependent-density functional theory). The authors should mention the computational details of the TD-DFT calculations in the method section in the supplementary information and specifically mention the DFT functional and basis-set in the caption of supplementary table 1.

Response: Thanks for the reviewer's comments. The description of time-dependent density functional theory (TD-DFT) was added in the revised manuscript. The computational details of the TD-DFT calculations were provided in the method section in the Supplementary Information. Specifically, the DFT functionality and basis-set in the caption of Supplementary Table 1 was

provided, including excited states calculated by the CAM-B3LYP functional and the def2-SVP basis set via ORCA software.

Added in Page 13 of Revised Supplementary Information: “All-electron density functional theory (DFT) calculations were carried out by the ORCA quantum chemistry software (Version 5.0.3). Grimme's D3BJ dispersion correction was used to improve calculation accuracy¹⁴. The excited states were calculated with linear response time-dependent DFT (TD-DFT) at the optimized ground state geometry. The excited states calculations were using the CAM-B3LYP functional and the def2-SVP basis set¹⁵⁻¹⁸.”

Added in Page 7 of Revised Manuscript: “Based on the time-dependent density functional theory (TD-DFT) simulations, we calculated transition energies and probabilities of each excited state for all COFs. The strongest oscillator strength on fragments of TpaBtt, TapbBtt and TaptBtt indicated that the S1 state has mostly HOMO to LUMO transition (Supplementary Table 1). Energetic levels of several molecular orbital are equal to or lower than the highest occupied molecular orbital (HOMO), while almost all electrons are contributed by the lowest occupied molecular orbital (LUMO) in most of transition, indicating that the electronic configuration of LUMO nearly represents photogenerated electron composition.”

REVIEWERS' COMMENTS

Reviewer #3 (Remarks to the Author):

I am happy with the changes made by the authors and can now recommend publication of the manuscript.

REVIEWERS' COMMENTS

Reviewer 3:

I am happy with the changes made by the authors and can now recommend publication of the manuscript.

Response: We thank the reviewer for the time and effort on reviewing our work.